# Exploring the Spatial Dynamics of In-Distribution and Out-of-Distribution Data in Logit Space

## Abstract

Out-of-distribution (OOD) data pose a significant challenge to deep learning (DL) classifiers, prompting extensive research into their effective detection methods. Current state-of-the-art OOD detection methods employ a scoring technique designed to assign lower scores to OOD samples compared to in-distribution (ID) ones. Nevertheless, these approaches lack foresight into the configuration of OOD and ID data within the latent space. Instead, they make an implicit assumption about their inherent separation or force a separation post-training by utilizing selected OOD data. As a result, most OOD detection methods result in complicated and hard-to-validate scoring techniques. This study conducts a thorough analysis of the logit embedding landscape, revealing that both ID and OOD data exhibit a distinct trend. Specifically, we demonstrate that OOD data tend to reside near the center of the logit space. In contrast, ID data tend to be situated farther from the center, predominantly in the positive regions of the logit space, thus forming class-wise clusters along the orthogonal axes that span the logit space. This study highlights the critical role of the DL classifier in differentiating between ID and OOD logits.

## 1 Introduction

Deep learning (DL) classification models perform well at generalizing from large datasets, achieving superior classification accuracy compared to many alternatives. They deliver highly accurate predictions when the test data aligns with the training data's distribution. However, current DL models are unable to handle out-of-distribution (OOD) data. This limits their application in critical fields such as biomedicine, finance, and autonomous systems.

For instance, consider the scenario in biomedicine where DL models classify bacteria from genome sequences. In such cases, it is crucial to account for the presence of novel bacteria types, which can be considered as OOD instances. Failure to account for these could lead to misclassifying these novel bacteria as known types, potentially resulting in flawed diagnostics or misleading scientific conclusions (Ren et al., 2019). This example underscores the need for more robust DL models to identify and handle OOD data effectively. Neglecting these novel entities may result in their misclassification as known types (Ren et al., 2019).

Recent OOD detection methods predominantly operate under the assumption that a classifier, when trained on ID data, intrinsically maps the logits of OOD samples to a distinct spatial location within the logit landscape, divergent from those of ID instances. Thus, differentiating OOD instances from ID data typically involves assigning high likelihood values to the logit (or softmax) location of the ID samples (Vyas et al., 2018; Lee et al., 2018; Sun et al., 2022; Gomes et al., 2022; Liu et al., 2020; Komini & Girdzijauskas, 2024).

Nevertheless, these strategies lack foundational awareness regarding the specific locational distribution of OOD samples in the embedding space. Consequently, these techniques attempt complicated and computationally intensive density estimations of the ID logits, categorizing those samples that fall beneath a certain likelihood threshold as OOD. Furthermore, the scalability of these methods in relation to the number of ID classes is limited, as they require robust performance in density mapping across all ID classes.

An inaccuracy in mapping even a single ID class may lead to a substantial error rate, where the methods could erroneously classify ID instances as OOD. Instead, our study demonstrates that a well-trained DL classifier, incorporating nonlinearities that suppress negative values (e.g., ReLU), systematically maps ID data into well-defined, class-specific clusters with a predictable structure. These ID clusters are situated along orthogonal axes within the positively constrained logit space and are notably separated from the logit space's center. Additionally, we reveal that OOD data are not arbitrarily scattered in the logit space but rather centrally positioned.

*Although previous research has explored the separation of OODs and IDs in the logit space (Lee et al., 2018; Liu et al., 2020; Choi et al., 2024; Katz-Samuels et al., 2022b), this work demonstrates a configuration of OODs and IDs logits.* The noted positioning of OOD and ID logits lays the groundwork for the possible creation of a binary classifier (OOD from ID), which could lead to simpler yet more effective OOD detection models. This study presents the following contributions:
1. An investigation into the spatial allocation of ID data within the logit space.
2. An empirical validation of the observed ID and OOD logit allocation over many models.

## 2 Method

### 2.1 Understanding ID and OOD data in DL models

In exploring ID and OOD data, it is crucial to delineate their distinctions in relation to DL classifiers.

The training dataset is regarded as the optimal empirical representation of ID data. Within this dataset, ID data tend to aggregate into class-specific clusters based on discriminative features corresponding to each class. The exact parameterization of this feature space remains unknown; however, the annotated empirical ID dataset serves as a practical surrogate. Assuming, these discriminative features constitute a multimodal distribution where each mode corresponds to one of the target classes. For ID data classification, DL models are engineered to exploit these discriminative features distribution and the annotated labels to effectively map the ID data into respective class-specific clusters within the logit vector space.

Whenever we encounter data whose features deviate significantly from this defined distribution of discriminative ID features, they are considered OOD. The degree to which the features of OOD data diverge from this distribution determines their shift with the ID features. Consequently, the more distant the OOD features are from the ID distribution, the lesser their resemblance to the ID discriminative features, rendering them less perceptible from the DL model.

More concretely, a DL classifier trained to differentiate cats from dogs will encounter difficulties with photos of horses and wolves. However, because wolves' features are more similar to dogs, wolves represent closer OOD data compared to horses, which are farther from both trained categories.

### 2.2 Analyzing ID and OOD logits in DL models

Assuming that DL models operate as spatial-invariance pattern-matching mechanisms operating over the distribution of discriminative features, these models produce high positive logits when the data features exhibit strong coherence with the feature representations parameterized during training. To do so, DL models rely on convolutions and matrix multiplications, which are composed of dot-product computations between the model weights and the input data. During training, the objective is to maximize the dot-product values between the model weights and the ID training data.

Furthermore, we assume that the feature representations of OOD data reside in regions significantly distant from the distribution of discriminative features associated with ID data. Consequently, these OOD features exhibit a distributional shift relative to the feature representation parameterized by the DL model. As a result, the dot-products for the OOD data with the model weights remain low, leading to their logits being centrally concentrated both before and after training. Simultaneously, the logits associated with ID data are driven towards higher positive values.

To visually represent the empirical distributions of ID and OOD logits before and after the training phase, we employed a binary classification model using a multilayer perceptron (MLP) model (see Appendix A and fig. 1a).

To minimize any initial biases, weights of DL models are initialized using a centered Gaussian distribution (Glorot & Bengio, 2010; He et al., 2015a), while the biases are set to zero. Furthermore, considering that both ID and OOD data originate from different distributions than the model's initial weight distribution, it is reasonable to assume that they are distributionally shifted from the model's initialized weights. This shift exemplifies as reduced co-variability between the model's weight parameters and both OOD and ID data samples (see Corollary 1), consequently leading to negligible covariance.

**Corollary 1.** *Assuming the features of the ID and OOD data, as well as the initialized model weights, originate from three distinct distributions with minimal overlap—it follows that the expected dot-product value between the data (ID and OOD) and the DL model's weights is minimal.*

*Proof.* Given the distribution shift between the data vector $\hat{\boldsymbol{x}}$ and the model's weights $\omega$, their covariance is expected to be minimal, i.e.,

$$|\operatorname{Cov}(\hat{x}, \omega)| < c, \quad \text{where } c \text{ is a sufficiently small positive constant.}$$

Define the covariance between model weights $\omega$ and data $\hat{\boldsymbol{x}}$ as

$$|\operatorname{Cov}(\hat{\boldsymbol{x}}, \omega)| = |\mathbb{E}[\langle \hat{\boldsymbol{x}}, \omega \rangle] - \mathbb{E}[\hat{\boldsymbol{x}}]\mathbb{E}[\omega]| < c.$$

Considering the DL model are initialized from a centralized distribution (Glorot & Bengio, 2010; He et al., 2015a), the initial expectation of the model weights is zero ($\mathbb{E}[\omega] = 0$), it follows that

$$|\mathbb{E}[\langle \hat{\boldsymbol{x}}, \omega \rangle]| < c.$$

$\square$

Because DL models fundamentally rely on the dot-product operations between inputs and weights, this result suggests that the logits of both ID and OOD data are centered around zero in the logit space prior to model training (see fig. 1b).

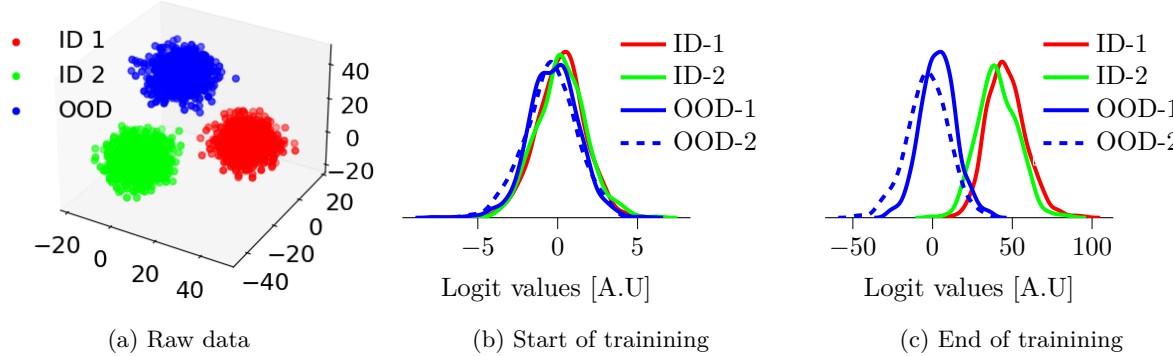

(a) Raw data      (b) Start of trainining      (c) End of trainining

Figure 1: Figure 1a shows raw data sampled from a multimodal Gaussian distribution, utilized as training data for a simple MLP binary classifier depicted in Appendix A. In this figure, red and green points denote ID classes for binary classification, and blue points represent OOD data. Figure 1b and Figure 1c demonstrate kernel density estimations (KDE) across logit cells for both OOD and ID data before and after model training, respectively. In both figures, 'OOD-1' and 'OOD-2' refer to KDEs for OOD data within the first and second logits, while 'ID-1' and 'ID-2' represent KDEs for ID class one data in the first logit cell and ID class two data in the second logit cell, respectively.

## 2.3   Allocation of ID logits towards positive regions

Training a DL classifier involves utilizing the cross-entropy loss, $(i.e., \boldsymbol{H}(\boldsymbol{Y}, \hat{\boldsymbol{Y}}))$, to encourage the prediction $(\hat{\boldsymbol{Y}})$ to closely align with the ground truth $(\boldsymbol{Y})$. When employing one-hot encoding for both $\hat{\boldsymbol{Y}}$ and $\boldsymbol{Y}$, the training objective simplifies to:

$$\boldsymbol{H}(\boldsymbol{Y}, \hat{\boldsymbol{Y}}) = -\sum_i \boldsymbol{Y}(i) \log(\hat{\boldsymbol{Y}}(i)) = \underbrace{-\boldsymbol{Y}(j) \log(\hat{\boldsymbol{Y}}(j))}_{\boldsymbol{Y}(j)=1} - \sum_{i,i \neq j} \underbrace{\boldsymbol{Y}(i) \log(\hat{\boldsymbol{Y}}(i))}_{\boldsymbol{Y}(i)=0} = -\log(\hat{\boldsymbol{Y}}(j)).$$

Eventually, the minimization of the cross-entropy loss $\left(i.e., \min[\boldsymbol{H}(\boldsymbol{Y}, \hat{\boldsymbol{Y}})]\right)$ equivalues to the maximum likelihood estimation (MLE) $\left(i.e., \min[-\log(\hat{\boldsymbol{Y}}(j))]\right)$.

As training progresses, the softmax layer aims to generate a response close to one for the cell corresponding to the correct class $\left(i.e., \hat{\boldsymbol{Y}}(j) \to 1\right)$. Additionally, owing to the inherent property that the softmax output is confined within a simplex $\left(i.e., \hat{\boldsymbol{Y}}(j)^{\uparrow} + \sum_{i,i \neq j} \hat{\boldsymbol{Y}}(i)^{\downarrow} = 1\right)$, the remaining cells are pushed towards values close to zero $\left(i.e., \hat{\boldsymbol{Y}}(i)_{i \neq j} \to 0\right)$. Hence, optimization can be conceptualized as the maximization of the softmax cell corresponding to the correct class and the simultaneous minimization of cells associated with incorrect classes.

This pattern of maximization-minimization is also observed in other classification losses (i.e., Support Vector Machine (Tang, 2015)and Kullback-Leibler divergence (Cui et al., 2024)), which are commonly employed in training DL classification models. This maximization-minimization optimization extends from softmax cells directly to the respective logit cells, as softmax maintains the order of logits. In particular, the logit cell linked to the correct class tries to attain large positive values (see fig. 1c).

However, when suppressing the negative values in an activation layer, the minimization process results in logit values near zero rather than approaching negative values of high magnitudes. Therefore, ID data are projected toward the positive regions of the logit space (see Appendix B for an extended discussion). Given that the logits reach their high positive value for the correct logit cell indicated by the one-hot encoding and approach zero for all other categories, it is evident that the logits for ID samples cluster by class along orthogonal axes within the logit space.

## 2.4   Central allocation of OOD logits

Since DL classifiers (i.e., ResNet, DenseNet, ViT) are trained using maximum likelihood estimation and their architectures rely on discrete convolutions, which itself relies on dot-products, the training process inherently seeks to maximize the dot-product between the ID data discriminative features and the model's parameters (see Appendix B). Furthermore, maximizing the dot-product between two vectors enhances their linear association. Correspondingly, a pronounced linear relationship between two vectors typically results in a higher co-variability between these two vectors. As a result, maximizing the dot-product enables a high degree of covariance and, by extension, cross-correlation. Hence, one can safely assume that there is a positive relationship between the dot-product of two vectors (i.e., $\langle \hat{\boldsymbol{x}}, \omega \rangle$) and their covariance, given that both metrics assess the degree of alignment between the two vectors.

Prior to training, the initial distributions of the model's weights and any given data are disparate. This disparity typically results in both the covariance and the dot-product between the model's weights and any given data being close to zero due to the lack of any established relationship. Thus, an untrained DL model generally steers any input data towards the center of the logit space (see Corollary 1,Figure 1b).

After the onset of training, the aim is to align the model's weights with the discriminative features of the ID data, maximizing the dot-product and increasing their co-variability, culminating in stronger activation of the logit cell that encodes the correct class (see fig. 1c and Theorem 1 in Appendix B). Since the discriminative features of OOD and ID data derive from fundamentally different distributions, they inherently exhibit a certain level of distributional shift. This shift is also reflected in the distribution of the OOD features in relation to the model's parameters since the latter are trained to align with the ID discriminative features. This shift implies that the covariance between OOD and parameters will likely remain minimal, even post-

train. As a result, their expected dot-product tends to yield smaller magnitudes. Therefore, OOD data tend to remain centered within the logit space even after training (see figs. 1b and 1c).

## 3 Related Work

Although the differentiation between OOD and ID logits has been extensively studied (Wang et al., 2022; Liu et al., 2020; Hsu et al., 2020; Lee et al., 2018; Ren et al., 2021), existing methods do not anticipate their actual configuration. Conventional OOD detection methods predominantly classify data by first identifying ID samples and subsequently labeling all other samples as OOD by default. A recent empirical investigation has not only highlighted the transferability of ID training strategies to OOD detection but also identified a tangible correlation between the robustness of ID training protocols and OOD detection efficacy (Wenzel et al., 2022). This study suggests that refining ID training methods could unlock potential pathways for enhancing OOD detection. Another study examines the influence of pre-trained Vision Transformers (ViT) (Vaswani et al., 2023) on ImageNet and reports notable improvements in OOD detection performance (Dosovitskiy et al., 2021). Parallel to these observations, another line of research incorporates outlier data — surrogates for OOD samples — within the training phase. This is achieved through an auxiliary loss term that sharpens the contrast between ID and outlier inputs, potentially strengthening OOD detection (Katz-Samuels et al., 2022a; Hendrycks et al., 2019; Wang et al., 2023; Du et al., 2022; Ming et al., 2022). Complementing these approaches, there has been a significant effort to restrict the classification of ID data into a hyperspherical embedding, which intrinsically helps OOD detection(Ming et al., 2023).

Another line of research assumes an inherent separation between OOD and ID logits and tries to devise scoring techniques using solely ID logits or softmax output. The OOD detection works by telling as OOD anything that is not ID. The earliest work on this front assumes clustering of ID logits into a multimodal Gaussian distribution and then tries to utilize Mahalanobis distance (Lee et al., 2018; Ren et al., 2021). More advanced methods try to upgrade the Mahalonobis distance with geometric information using Fisher Information matrix (Gomes et al., 2022) Other works try to perform a data drive density estimation using energy-based models (Liu et al., 2020).

Another promising research demonstrates the utility of enhanced Hopfield networks in amplifying the distinction between ID and OOD data Hofmann et al. (2024). Similarly, another proactive work tries to increase the separability between OOD and ID using kernel principal component analysis on the OOD and ID embeddings (Fang et al., 2024). Last but not least, Zhang et al. (2024) tries to learn the shape of ID feature space using an online expectation maximization, which enhances the detection of OOD post-train.

## 4 Results

In these experiments, we demonstrate that the OOD logits remain near the center of the logit space both before and after training. In contrast, ID logits consistently gravitate towards clusters around class-specific areas in the positive regions of the logit space. Furthermore, we show that these ID clusters align with the orthogonal axis that spans the logit space embeddings.

**OOD vs ID during training:** In fig. 2, we empirically illustrate the distribution of ID and OOD logits before and after training. Additionally, we present the evolution of these distributions throughout the training process. To do so, we employed Resnet-9 (He et al., 2015b) with CIFAR-100 (Krizhevsky et al., a) as the ID data and CIFAR-10 (Krizhevsky et al., b) as the OOD data. *Additionally, for the correctly classified ID data, we categorize 'ID-in' when the logit values reach their maximum in the cell corresponding to the correct class, as indicated by the one-hot encoding of that class. Conversely, we categorize it as 'ID-out' when this condition is not met.*

We represent the empirical distributions of the logit outputs for both ID and OOD samples via kernel density estimation (KDE) (Bishop, 2006). At the beginning of training, one can notice that the densities for both OOD and ID logits are concentrated near zero (see figs. 2a to 2d). While OOD and ID-out logits maintain their central tendency around zero (see fig. 2c) the ID-in logits exhibit a shift towards higher positive values (see fig. 2a). Analyzing the peak (i.e., mode) of each KDE plot (i.e., ID-in, ID-out, and OOD in fig. 2c), it

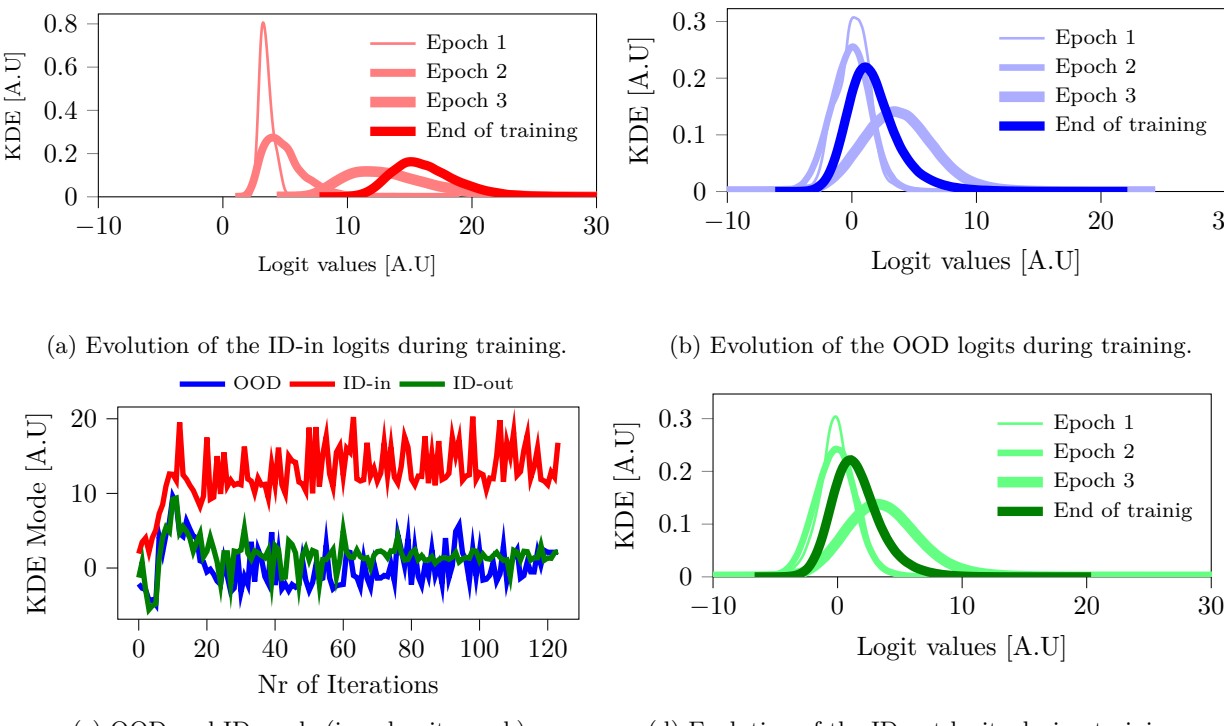

(a) Evolution of the ID-in logits during training.

(b) Evolution of the OOD logits during training.

(c) OOD and ID mode (i.e., density peak).

(d) Evolution of the ID-out logits during training.

Figure 2: Figure 2a presents the density plot across various epochs for the aggregation of ID-in across all logits, while fig. 2b displays the density plot across different epochs for the aggregation of OOD across logits. Similarly, fig. 2d shows the density plot over different epochs for the aggregation of ID-out across all logits. Since the KDE plots are limited to the first three and the final epochs, we included fig. 2c to provide a comprehensive view of the entire trajectory, featuring the peak (i.e, mode) of the density plot for every epoch.

is evident that ID-in trends towards positive values over time as anticipated by Theorem 1 in Appendix B. Furthermore, the ID-out and OOD logits remain centrally positioned, aligning with our analytical predictions. In addition to the density plots in figs. 2a, 2b and 2d, which illustrate the aggregation of ID-in, ID-out from training data along with OOD across all logit cells, see Appendix C for detailed visualization of density plots on individual logit cells for a more in-depth analysis.

To empirically validate the persistence of this configuration, we conduct a series of comprehensive experiments across diverse settings. Furthermore, to rigorously analyze the distributional properties of ID and OOD logits, we employ KDE to visualize their densities. This unified visualization framework enables a direct comparison of ID-in (the maximum logit values for correctly classified ID samples), ID-out (the remaining logit values for correctly classified ID samples), and OOD. By overlaying their KDE curves, we assess the divergence between ID-in and OOD, the separation between ID-in and ID-out, and the degree of overlap between OOD and ID-out distributions.

**Effect of activation function:** To further understand the configuration of ID and OOD logits, we investigate the impact of various activation functions on a ResNet-34 model. Specifically, we empirically demonstrated this impact by utilizing a selection of activation functions known for their inherent suppression of negative values, including Celu (Barron, 2017), Elu (Clevert et al., 2016), Gelu (Hendrycks & Gimpel, 2023), Selu (Klambauer et al., 2017), Silu (Elfwing et al., 2017), Relu (Hein et al., 2019), Leaky-Relu (Maas et al., 2013), and Mish (Misra, 2020). A ResNet-34 model was trained on the SVHN dataset (Goodfellow et al., 2014) (i.e., ID data), utilizing each activation function. Simultaneously, the CIFAR-10 dataset was used as OOD data.

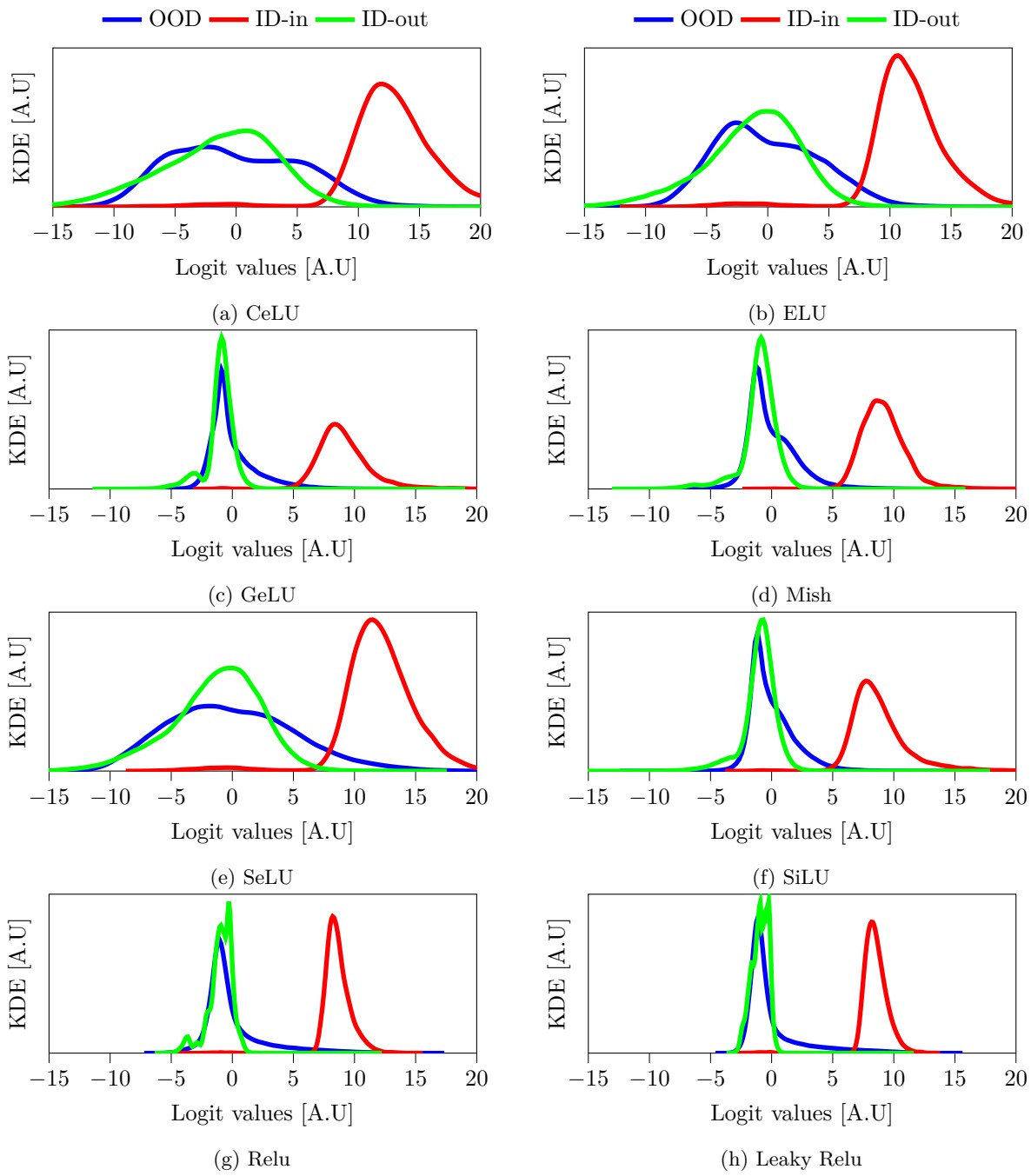

Figure 3: An analysis of the density over logits across eight distinct activation functions that suppress negative values is presented. The ResNet-34 architecture is utilized and trained on the SVHN dataset as the ID data, while the OOD includes CIFAR-10.

A ResNet-34 model was trained on the SVHN dataset (Goodfellow et al., 2014), (i.e., ID data), utilizing each activation function (i.e., Celu, Elu, Gelu, Mish, Selu, Silu, ReLU, Leaky ReLU). The model is trained using stochastic gradient descent (SGD) with a cyclical learning rate starting at $lr = 10^{-3}$ with a cosine annealing operation with a periodicity of 200. Furthermore, the momentum is 0.9 while the weight decay $5 * 10^{-4}$. A batch size of 256 is applied for both test and train data. No regularization is applied to the training process, while the training data are augmented with random flipping and cropping.

One can notice that ID-in logits maintain a tendency towards high positive values across all the activation functions (see fig. 3). On the other hand, ID-out and OOD logits are predominantly centralized around zero (see fig. 3). Consequently, despite the application of varying non-linearities, the relative configuration of ID and OOD logits remains similar.

However, different activation functions yield varying degrees of separation and overlap between ID and OOD logits. ReLU produces the most compact logit distributions with minimal overlap, as it preserves positive activations while suppressing negative ones. In contrast, activation functions with nonlinear modulations (e.g., CeLU, ELU, SELU) induce a broader spread in logits and exhibit greater overlap between ID and OOD samples.

This increased dispersion in ID logits from these nonlinearities may stem from the introduction of spurious discriminative features parameterized by the DL model despite their absence in the training data. These spurious features contribute to higher variability in OOD logits, thereby increasing their overlap with ID distributions.

Thus, unlike ReLU, transcendental activation functions (e.g., CeLU, ELU, SELU) increase computational overhead in training and inference while simultaneously diminishing the distinctiveness of ID and OOD logit distributions, potentially degrading OOD detection performance.

In addition to the density plots depicted in fig. 3, which show the distribution of aggregated ID-in, ID-out, and OOD across all logit cells, figs. 10 to 17 in Appendix D provides a detailed visualization of density plots on individual logit cells using various activation functions.

**Effect of dropout:** Dropout is a pivotal component in enhancing the generalizability of contemporary DL methodologies Srivastava et al. (2014). This technique forces the model to yield accurate classifications, leveraging a randomly selected subset of its parameters by deactivating a proportion of the gradients during the training phase. Although training involves only a subset of parameters, the entire set is employed during inference. However, the influence of dropout on the differentiation of ID and OOD data has been relatively understudied, as the main focus of dropout has been optimizing the accuracy of ID datasets rather than maximizing the discriminative capability between OOD and ID datasets. Instead, our study focuses on investigating this discrimination. It reveals that OOD vs. ID behaves differently when using a partial subset of parameters (i.e., dropout ON) versus the entire set (i.e., dropout OFF).

In our experimentation, we employed a ResNet-34 architecture configured with varying dropout ratios of 20%, 40%, 60%, and 80% applied in each convolution layer. The dataset ensemble comprised $\{D\}$ = SVHN, CIFAR-100, CIFAR-10, Tiny ImageNet (Deng et al., 2009), iSUN (Xu et al., 2015), LSUN (Yu et al., 2016), where the model was trained specifically on CIFAR-10 and utilized $\{D/\text{CIFAR-10}\}$ as the OOD dataset.

One can notice that the anticipated pattern of ID and OOD persist across all levels of dropout utilized. However, an increase in the dropout rate results in an enhanced overlap between ID and OOD predictions (see fig. 4). Moreover, this overlap is more pronounced when the complete set of weights is engaged during inference (i.e., dropout OFF) compared to when only a randomly selected subset is used (i.e., dropout ON) (refer to fig. 4).

Because the overlap between OOD and IID logits increases as the dropout rate rises (see fig. 4), it suggests that applying dropout to the convolutional layers has a detrimental effect on the generalization of the ID discriminative features. Moreover, this empirical observation indicates the unintended introduction of a spurious correlation between the model weights and OOD features.

In addition to the density plots shown in fig. 4, which illustrate the aggregated distribution of ID-in, ID-out, and OOD across all logit cells, a more detailed visualization of density plots on individual logit cells can be found in figs. 18 to 25 of Appendix D.

**Experiments on different classifiers:** The analysis of ID and OOD logits has been expanded across various DL classifier models. Our study examines various iterations of DenseNet (Huang et al., 2018), specifically versions 121, 161, 169, and 201, as well as ResNet (He et al., 2015b), encompassing versions 18, 34, 50, 101, and 152. Furthermore, the utilized experimental dataset comprises $\{D\}$ = {SVHN, CIFAR-100, CIFAR-10, Tiny ImageNet, iSUN, LSUN}.

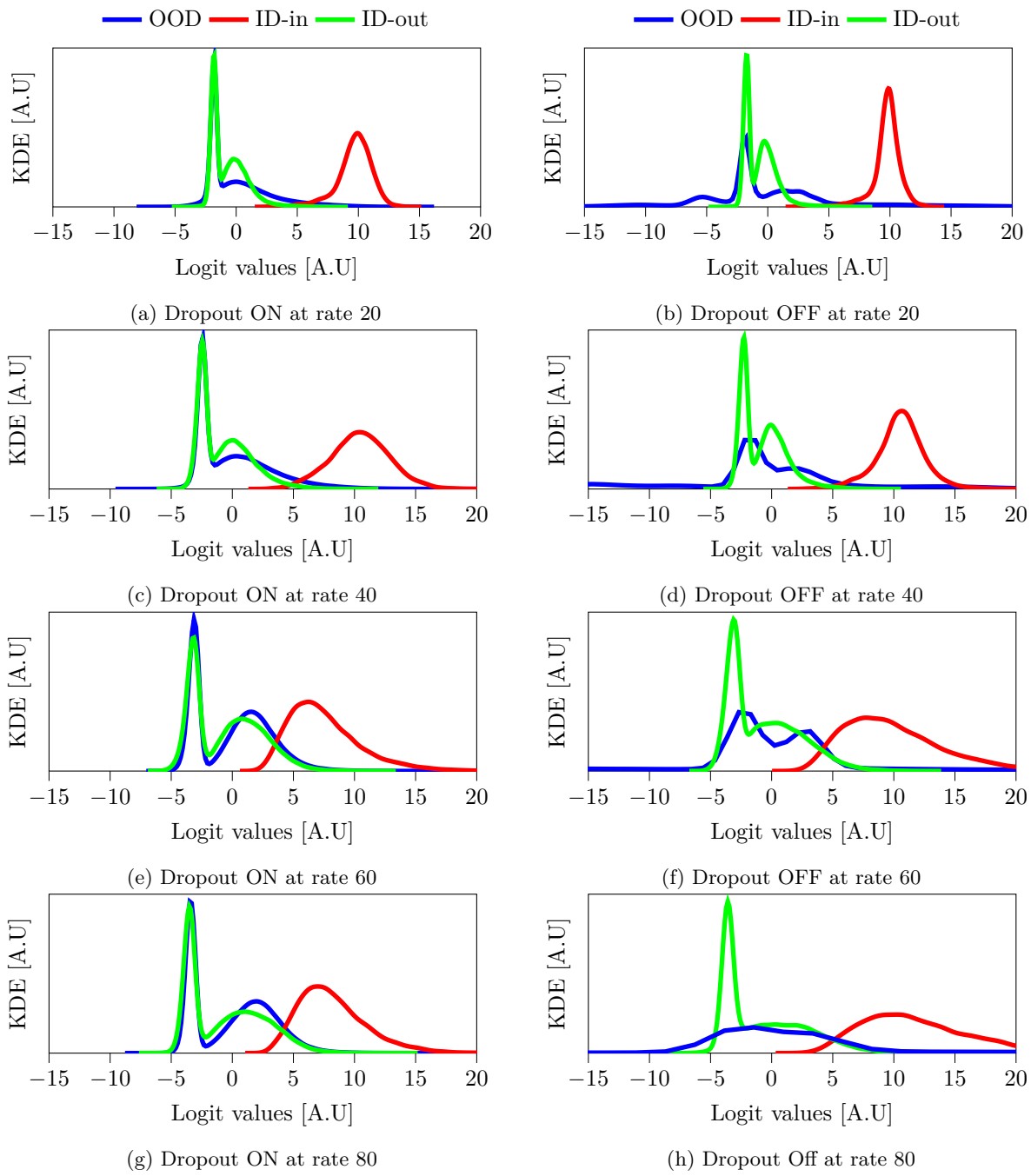

Figure 4: An analysis of the density over logits across eight distinct dropout rates is presented. The ResNet-34 architecture is utilized and trained on the CIFAR-10 dataset as the ID data, while the OOD includes $\{D/\text{CIFAR-10}\}$.

Densenet and ResNet models are trained using SGD with a cyclical learning rate starting at $lr = 10^{-3}$ with a cosine annealing operation with a periodicity of 200. Furthermore, the momentum is 0.9 while the weight decay $5 * 10^{-4}$. A batch size of 256 is applied for both test and train data, while the number of epochs is 200. ReLU is utilized as an activation function for every layer. No regularization is applied to the training process, while the training data are augmented with random flipping and cropping. Each version of Densenet and ResNet undergoes separate training on CIFAR-10 and SVHN as ID datasets.

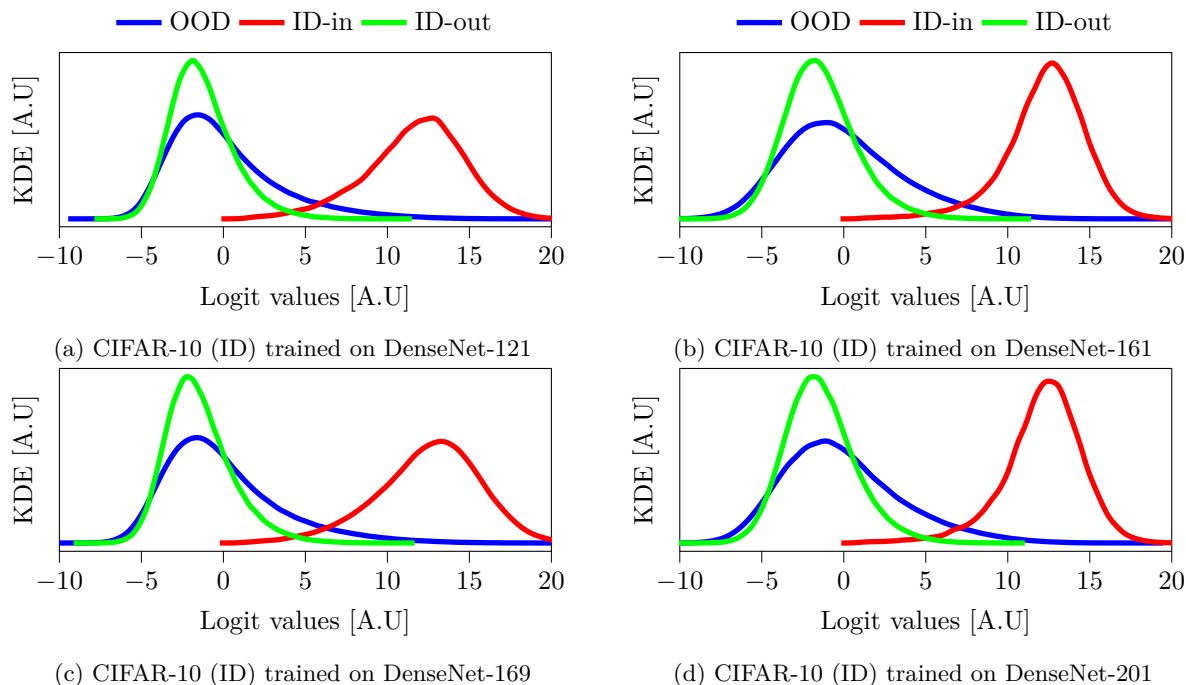

Figure 5: An analysis of the density over aggregated logits across distinct DenseNet architecture trained on the CIFAR-10 dataset as the ID data, while the OOD includes $\{D\}$/CIFAR-10. For a more detailed comparison, check figs. 28 to 35 in Appendix E.

When CIFAR-10 is utilized as ID, the remaining datasets are employed as OOD data, specifically$\{D\}$ without CIFAR-10 (i.e., $\{D\}$/CIFAR-10) is utilized as OOD. Similarly, when SVHN is utilized as ID, the remaining datasets are employed as OOD data, specifically$\{D\}$ without SVHN (i.e., $\{D\}$/SVHN) is utilized as OOD.

Observations indicate that the ID-in logits consistently tend toward higher positive values across various versions of DenseNet (see fig. 5 and fig. 26 in Appendix E) and ResNet (see fig. 6 and fig. 27 in Appendix E). Contrarily, ID-out and OOD logits tend to be concentrated around zero.

Interestingly, the spread and the degree of overlap between ID and OOD logits—remain consistent across different model architectures, including both ResNet and DenseNet variants. This suggests that these properties are largely architecture-agnostic, reinforcing the generalizability of our findings.

Notice that the density plots in figs. 26 and 27 demonstrate the distribution of the ID-in, ID-out, and OOD for all logit cells aggregated together. For thorough visual representations on a per-logit-cell basis, across all different versions of DensetNet and ResNet see figs. 28 to 45 in Appendix E.

**Experiments on different vision transformers:** Contrary to traditional convolutional neural networks (CNN) (e.g., DenseNet, ResNet), which process image patches exclusively on a spatial level, vision transformers (ViT) incorporate an additional component of interleaved processing among patches through attention mechanism (Dosovitskiy et al., 2021). To examine the effects of this interleaved processing on the arrangement of OOD and ID logits, we carried out experiments with various ViT configurations, including the base (ViT-B) and large (ViT-L) models, each with two different patch sizes: 16x16 and 32x32 pixels.

Furthermore, the utlized experimental dataset comprises $\{D\}$ = { SVHN, CIFAR-100, CIFAR-10, Tiny ImageNet, iSUN, LSUN}. Each model undergoes separate training on CIFAR-10 and SVHN as ID datasets. The remaining datasets are employed as OOD data, specifically $\{D\}$/CIFAR-10 and $\{D\}$/SVHN.

In fig. 7, one can notice that for all versions of the ViT, ID-in logits converge towards higher positive values as expected. Contrarily, the logits for both the ID-out and OOD samples predominantly cluster around the center of the logit space.

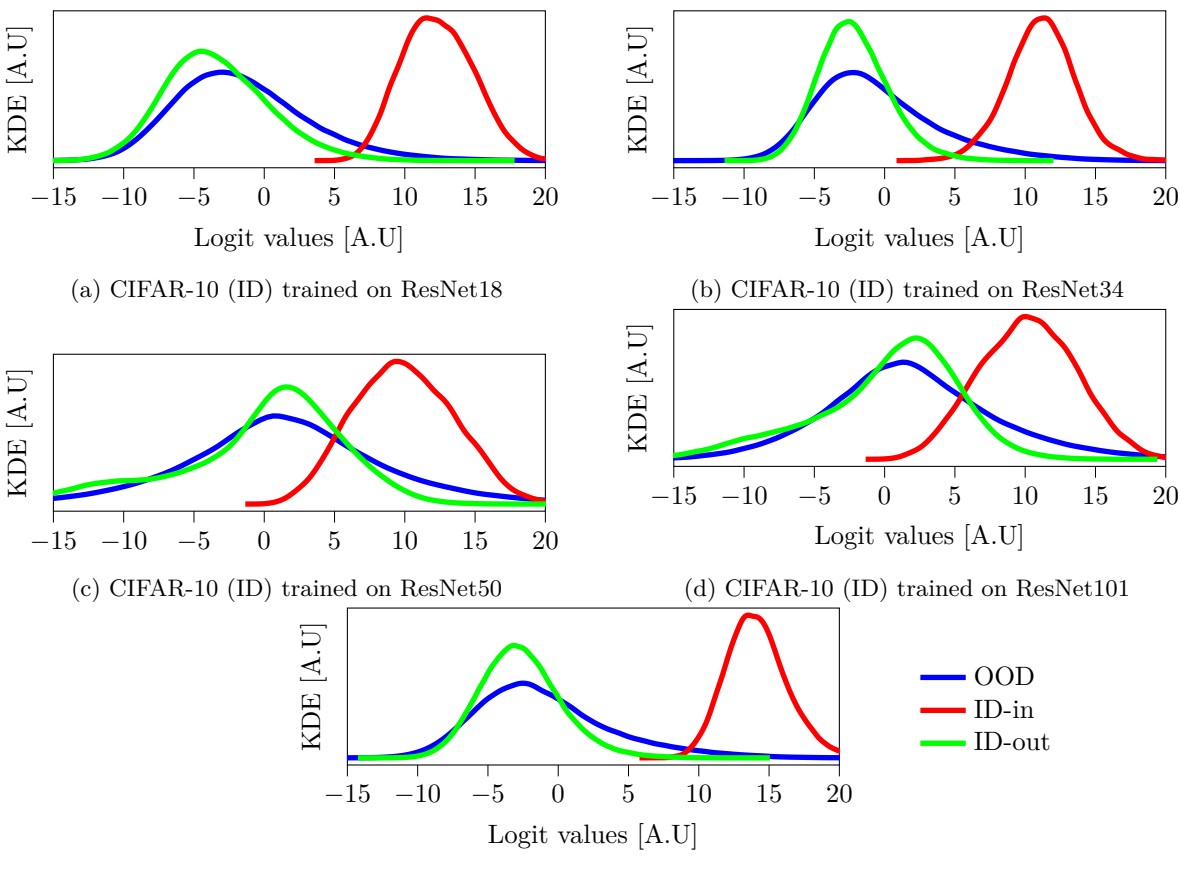

(a) CIFAR-10 (ID) trained on ResNet18

(b) CIFAR-10 (ID) trained on ResNet34

(c) CIFAR-10 (ID) trained on ResNet50

(d) CIFAR-10 (ID) trained on ResNet101

(e) CIFAR-10 (ID) trained on ResNet152

Figure 6: An analysis of the density over aggregated logits across distinct ResNet architecture trained on the CIFAR-10 dataset as the ID data, while the OOD includes $\{D\}$/CIFAR-10. For a more detailed comparison check figs. 36 to 45 in Appendix E.

The persistence of the anticipated logit configurations for both OOD and ID data demonstrates that, similar to CNNs, ViTs effectively parameterize the discriminative feature distribution of ID data. While CNNs exclusively leverage localized hierarchical features, ViTs augment these local patterns with global contextual information through self-attention mechanisms. Since self-attention operates via patch-wise dot-product interactions, it preserves the intrinsic feature structure of the ID data, avoiding spurious feature generation.

Furthermore, varying patch sizes (i.e., 16x16 to 32x32) in ViTs exhibit negligible impact on the resulting logit distributions, suggesting that even the small patch size is sufficient for the global context encoding. Furthermore, this invariance underscores that the parameterization of discriminative features is largely unaffected by patch-wise tokenization, reinforcing the stability of ViTs in modeling ID data distributions.

Similarly, scaling the ViT from the base configuration (12 layers, 768 hidden dimensions, 12 attention heads) to the larger variant (24 layers, 1024 hidden dimensions, 16 heads) preserves the overall logit distribution structure. Despite the significantly expanded parameter space, the larger model does not exhibit a substantial improvement in the separability between ID and OOD samples, suggesting that mere architectural scaling alone is insufficient to enhance OOD detection performance.

Observe that the density plots shown in fig. 7 depict the spread of ID-in, ID-out, and OOD aggregations over all logit cells. For a detailed visual analysis of each logit cell, refer to the various versions of ViT illustrated in figs. 47 to 54 in Appendix F.

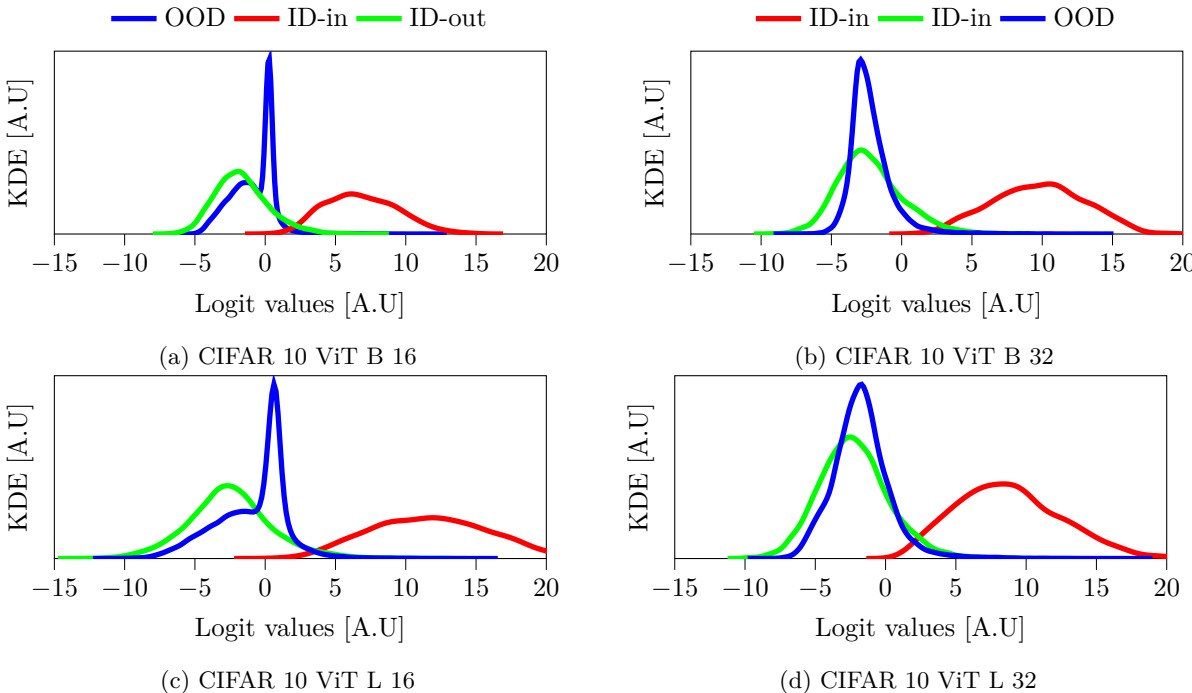

Figure 7: An analysis of the density over aggregated logits across distinct ViT architecture trained on the CIFAR-10 dataset as the ID data, while the OOD includes {$D$}/CIFAR-10. For a more detailed comparison check figs. 46 to 54 in Appendix F

## 5 Conclusion

While current research on OOD detection focuses on developing new methods that naturally give higher scores to ID data and, by default, lower scores to OOD samples, this study concentrates on analyzing the differences between OOD and ID logit distributions.

Specifically, we demonstrated the anticipated configuration of OODs and IDs logits, i.e. that ID logits are clustered by class towards the positive region of the logit space, aligning with the orthogonal axis that spans this space. Additionally, OOD logits remain consistently distinct from ID logits, clustering around the center of the logit space.

This behavior of OOD and ID logits is consistent across various architectures (i.e., convolutional neural network models and vision transformers) and activation functions tested on a set of large and diverse OOD data. However, elevated dropout rates on the convolutional layers have been identified as a significant factor in increasing overlap between OOD and ID samples.

As a future direction, the observed patterns within OOD, ID-in, and ID-out logits indicate the potential for a novel approach that leverages ID-out logits as proxies for OOD instances. This approach will facilitate the development of a binary classifier neural network designed to differentiate between OOD and ID samples, employing ID-out logits as representative proxies for OOD instances. Consequently, this method addresses OOD detection as a straightforward classification challenge, thus mitigating the need for threshold-based discrimination methods.

An additional crucial application of the observed logit configuration is the detection of ID data shifts. Since ID values are typically oriented towards positive values along the corresponding axis, this characteristic can be utilized to develop a more accurate and scalable approximation of the Wasserstein distance. Consequently, enabling a more sensitive metric to detect shifts toward the center of the logit space in the ID test data.

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

# A Toy example

The training configuration for the model outlined in table 1 includes a batch size of 64, a learning rate of 0.001, and 30 training epochs. To combat overfitting, a dropout rate of 0.8 is employed.

Table 1: Architecture of the MLP model.

| Layer Type | Output Size | Additional Information |
|---|---|---|
| Linear | 2048 | in_features=3 |
| ReLU | 2048 | - |
| Dropout | 2048 | p=0.8 |
| Linear | 2048 | in_features=2048 |
| ReLU | 2048 | - |
| Dropout | 2048 | p=0.8 |
| Linear | 2 | in_features=2048 |

# B In distribution positioning in the logit space during training

**Theorem 1.** *In the training process of a deep learning classifier utilizing an activation function that suppresses negative values, the logit corresponding to the true class (i.e., ID-in denoted by $\hat{L}(j)$), attain big positive magnitude ($\hat{L}(j) \to +\infty$). Simultaneously, the logits representing the incorrect classes (i.e., ID-out denoted by $\hat{L}(i)$ for $i \neq j$) converge towards minimal magnitude values ($\hat{L}(i)_{i \neq j} \to 0$).*

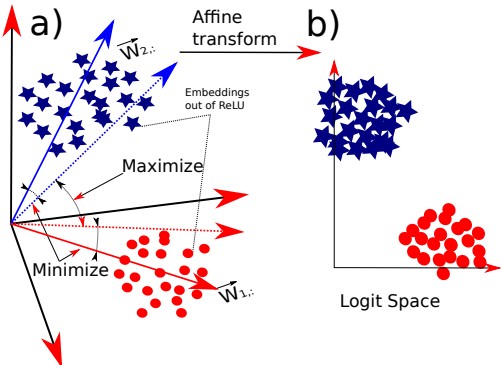

Figure 8: This toy example shows the separation of ID in a binary classification task. Figure a) contains the embeddings ($E$) rectified with a ReLU. Figure b) shows the linear separation of class-wise clustering of ID data logits ($\hat{L}$). The smaller the angle between $\vec{E}$ and $\vec{W}_{1,:}$, the higher the dot-product $\langle W_{1,i}, E_i \rangle$ Figure a); thus the more distanced from the center the ID logits are (Figure b). The bigger the angle between ($\vec{E}$) and $\vec{W}_{2,:}$, the higher the dot-product $\langle \vec{W}_{2,i}, \vec{E}_i \rangle$ (see, fig. 8 a), the more compact the ID logits are.

*Proof.* To establish the constraint towards zero for the logit cells not corresponding to the correct class (i.e., ID-out $\hat{L}(i)_{i \neq j} \to 0$), it is crucial to acknowledge that the predecessor latent space ($\hat{E}(i)$) is confined to positive values as the negative values are suppressed (see, fig. 8.a). The layer preceding the softmax constitutes a linear transformation of the data from high-dimensional embeddings ($\hat{E}$) to the logit space ($\hat{L} = \hat{E} \times W$, where $\times$ denotes matrix multiplication) with dimensions aligning with the number of specified classes (see, fig. 8.b). Since the optimizer seeks maximum response for the logit cell $\hat{L}[i]$ (i.e., ID-in), it aims to maximize the dot-product $\arg \max_{W[i,:]} \langle \hat{E}[:], W[i,:] \rangle^1$, $s.t : \hat{E}[:] \geq 0$.

Considering the embeddings $\hat{E}[:]$ and $W[i,:]$ as vectors in the vector space (see, fig. 8a), maximizing $\langle \vec{E}[:], \vec{W}[i,:] \rangle$ results in the minimization of the angle between $\vec{E}[:]$ and $\vec{W}[i,:]$ $(i.e., \min \angle(\vec{W}[i,:], \vec{E}[:]))$ while

---

[1] $\langle , \rangle$ indicates the dot-product

ensuring the former always remains in the positive regions. The optimization aims to maintain the direction of the vector $\vec{W}[i,:]$ akin to the cluster of vectors $\vec{E}[:]$, specifically within the positive regions (see, fig. 8.a).

Moreover, the optimization aims to achieve a minimum response for every other logit cell $\hat{L}[j \neq i]$ that does not correspond to the correct class, expressed as $\arg\min_{W[j \neq i,:]} \langle \hat{E}[:], W[j \neq i,:] \rangle$, subject to the constraint $\hat{E}[:] \geq 0$. In essence, it seeks to maximize the angle between $\vec{W}[j \neq i,:]$ and the cluster of vector data $\vec{E}[:]$ $\left( i.e., \max \angle(\vec{W}[j \neq i,:], \vec{E}[:]) \right)$ (see, fig. 8.a).

Thus, the clusters associated with different classes try to attain maximum angular separation from one another, leading the parameter vectors $\vec{W}[i,:]$ to align accordingly. Given that all vectors $\vec{E}[:]$ are angularly separated within the positive region, the maximum angle between these two vectors approaches perpendicularity (see Lemma 1) asymptotically. Consequently, the minimized logit values $\left( \arg\min(\vec{W}[j \neq i,:], \vec{E}[:]) \approx 0 \right)$ would asymptotically approach zero during training.

Consequently, the asymptotic behavior of the data configuration in the logit space compels the data points to form compact clusters far from the center of the space, corresponding to their respective classes. This process leads to the minimization of interclass distances and the maximization of intraclass distances.

$\square$

**Lemma 1.** *In the positive region of a high-dimensional space, the maximum angle that two vectors can attain is perpendicular.*

*Proof.* One way to establish this lemma involves employing the concept of cosine similarity. Let us consider two arbitrary vectors in an N-dimensional space, denoted as $X$ and $Y$, where $X, Y \in \mathbb{R}^N$. The cosine similarity between these vectors is defined as:

$$\cos_{\text{sim}}(X, Y) = \frac{\sum_{i=1}^{N} X_i Y_i}{\|X\| \|Y\|} \tag{1}$$

Given that both vectors reside in the positive region of the vector space, meaning that each component of the vectors satisfies $X_i \geq 0, Y_i \geq 0 \forall i \in [1,..,N]$, it is evident that any two vectors in the positive region cannot yield a negative value for the cosine similarity. This is because the numerator, representing the dot-product of the vectors, comprises products of non-negative components. Consequently, the numerator cannot be negative. Therefore, the minimum value that $\cos_{\text{sim}}(X, Y)$ can attain is zero, corresponding to perpendicular vectors. $\square$

Theorem 1 does not imply that all weights in the model must be non-negative after training. It's important to note that the activation function suppresses negative values, leading to nullified gradients for these outputs that do not contribute to the update process during training. Typically, such negative outputs originate from dot-products involving negative weights, suggesting that these negative weights might not consistently be updated during training sessions. Furthermore, since the predictive response predominantly depends on pathways utilizing positive weights (Lim et al., 2024), these positive weights are more frequently adjusted during the optimization process as outlined in Theorem 1.

## C    Experimentation on CIFAR-100 (ID) vs CIFAR-10 (OOD)

Resnet-9 is trained using stochastic gradient descent (SGD) with a learning rate starting at $lr = 10^{-1}$ The batch size is 256, and the number of epochs is 200. The learning rate is decimated every quarter of epochs. Within each quarter, the learning rate is scheduled using a 1cycle learning rate. ReLU is utilized as an activation function for every layer. No regularization is applied to the training process, while the training data are augmented with random flipping and cropping. In figs. 9a and 9b, we present the distributions of logit values for ID samples, with the former displaying densities corresponding to the ID-in and the latter for the ID-out. OOD densities are depicted for each logit in fig. 9c.

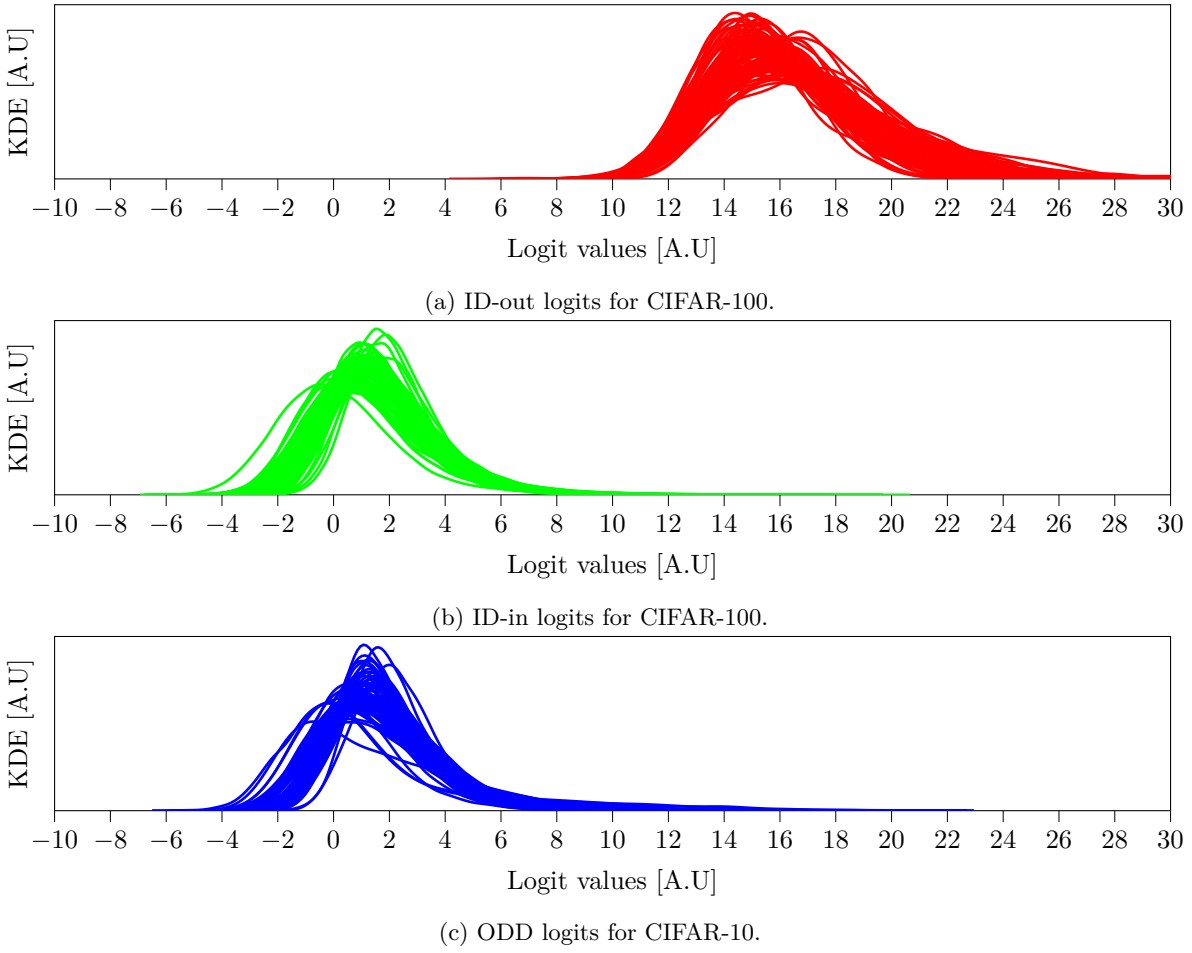

(a) ID-out logits for CIFAR-100.

(b) ID-in logits for CIFAR-100.

(c) ODD logits for CIFAR-10.

Figure 9: KDE response CIFAR-100 (ID) vs CIFAR-10 (OOD) while using Resnet-9 with ReLU activation function.

## D   Detailed visualization of density plots on individual logit cells

Figures 10 to 17 showcase a detailed visualization of the ID and OOD logits for each cell across different types of activation functions.

$$\text{Relu:} f(x) = \max(0, x)$$

$$\text{Celu:} f(x) = \max(0, x) + \min(0, \alpha(e^{x/\alpha} - 1))$$

$$\text{Elu:} f(x) = \begin{cases} x & \text{if } x > 0 \\ \alpha(e^x - 1) & \text{if } x \leq 0 \end{cases}$$

$$\text{GELU:} f(x) = x\Phi(x)$$

where $\Phi(x)$ is the cumulative distribution function of the standard Gaussian distribution:

$$\Phi(x) = \frac{1}{2}\left[1 + \text{erf}\left(\frac{x}{\sqrt{2}}\right)\right]$$

$$\text{Selu:} f(x) = \lambda \begin{cases} x & \text{if } x > 0 \\ \alpha e^x - \alpha & \text{if } x \leq 0 \end{cases}$$

$$\text{Silu:} f(x) = \frac{x}{1 + e^{-x}}$$

$$\text{Leaky-Relu:} f(x) = \begin{cases} x & \text{if } x > 0 \\ \alpha x & \text{if } x \leq 0 \end{cases}$$

$$\text{Mish:} f(x) = x \tanh(\ln(1 + e^x))$$

Similarly, figs. 18 to 25 provide a comprehensive visualization of the ID and OOD logits for each cell, delineating the impacts of various dropout rates.

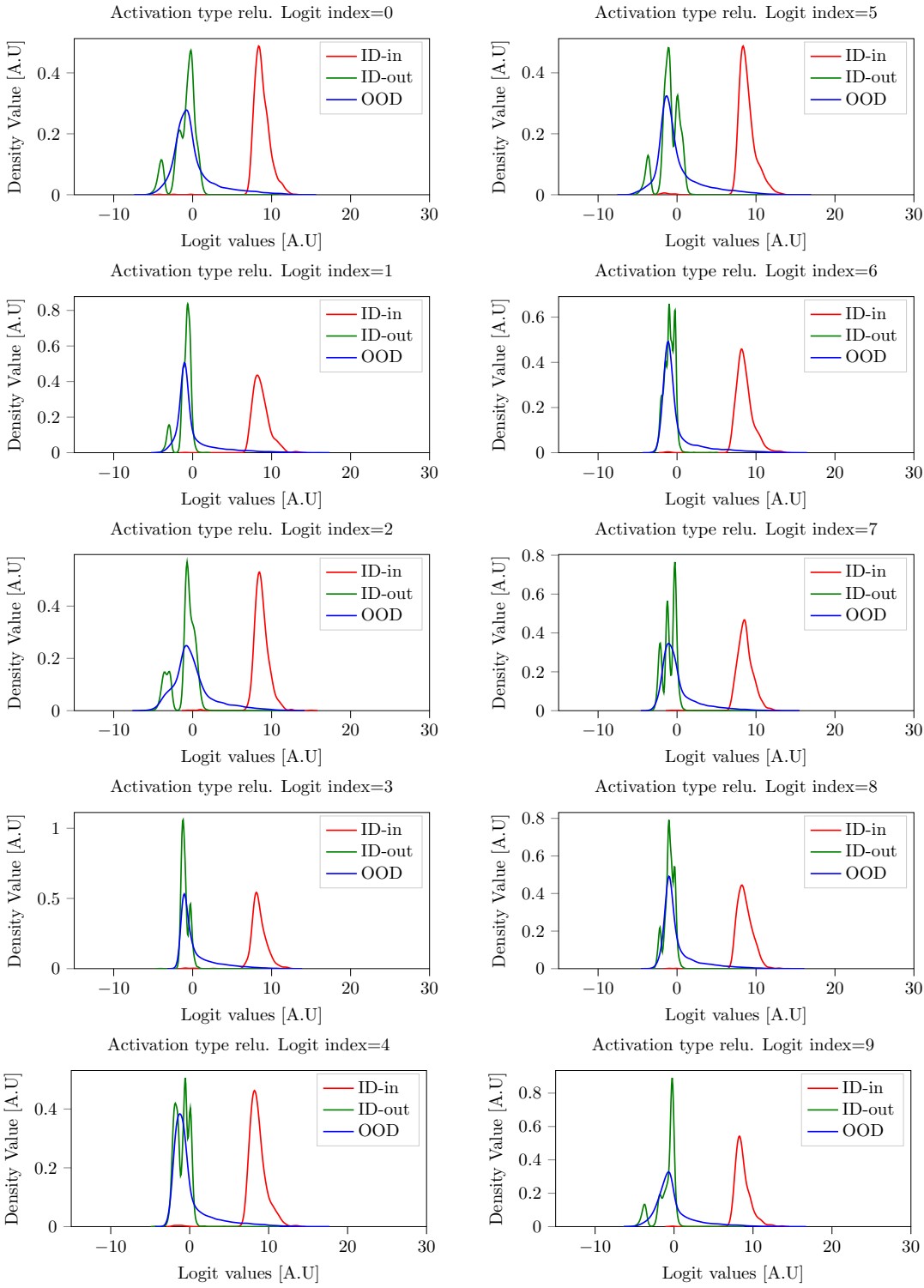

Figure 10: Densities over each logit cell from a Resnet-34 classifier with Relu activation.

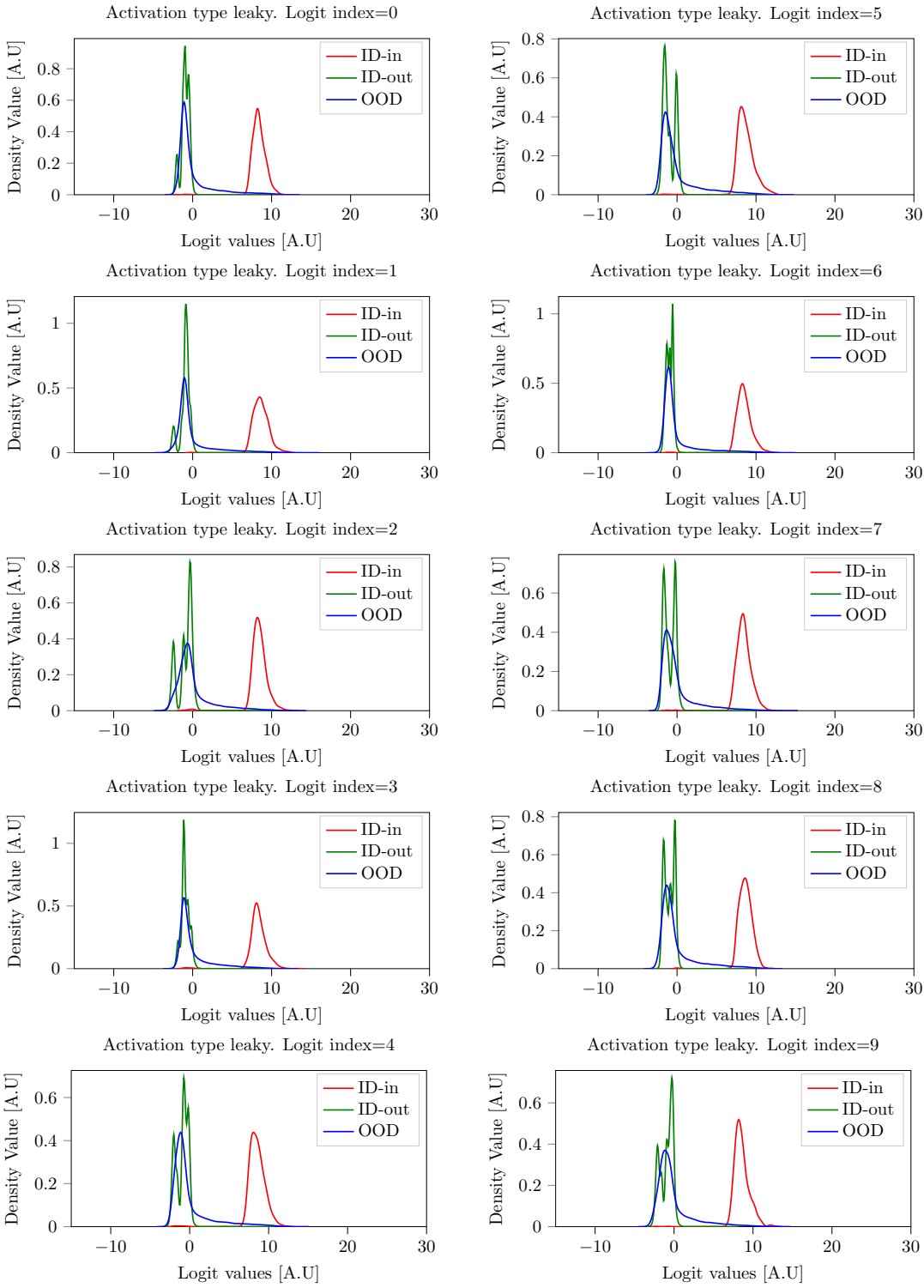

Figure 11: Densities over each logit cell from a Resnet-34 classifier with Leaky Relu activation.

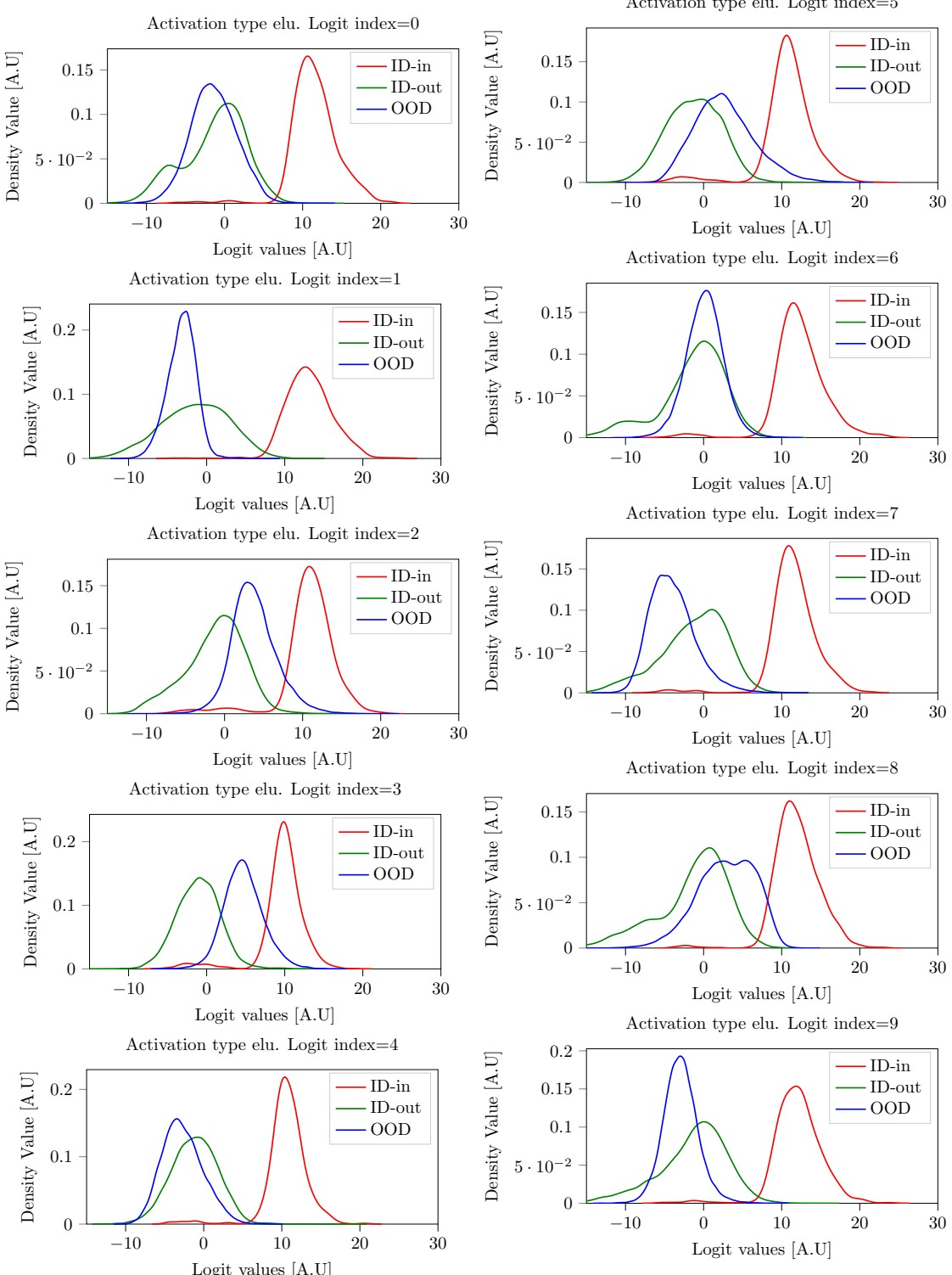

Figure 12: Densities over each logit cell from a Resnet-34 classifier with Elu activation.

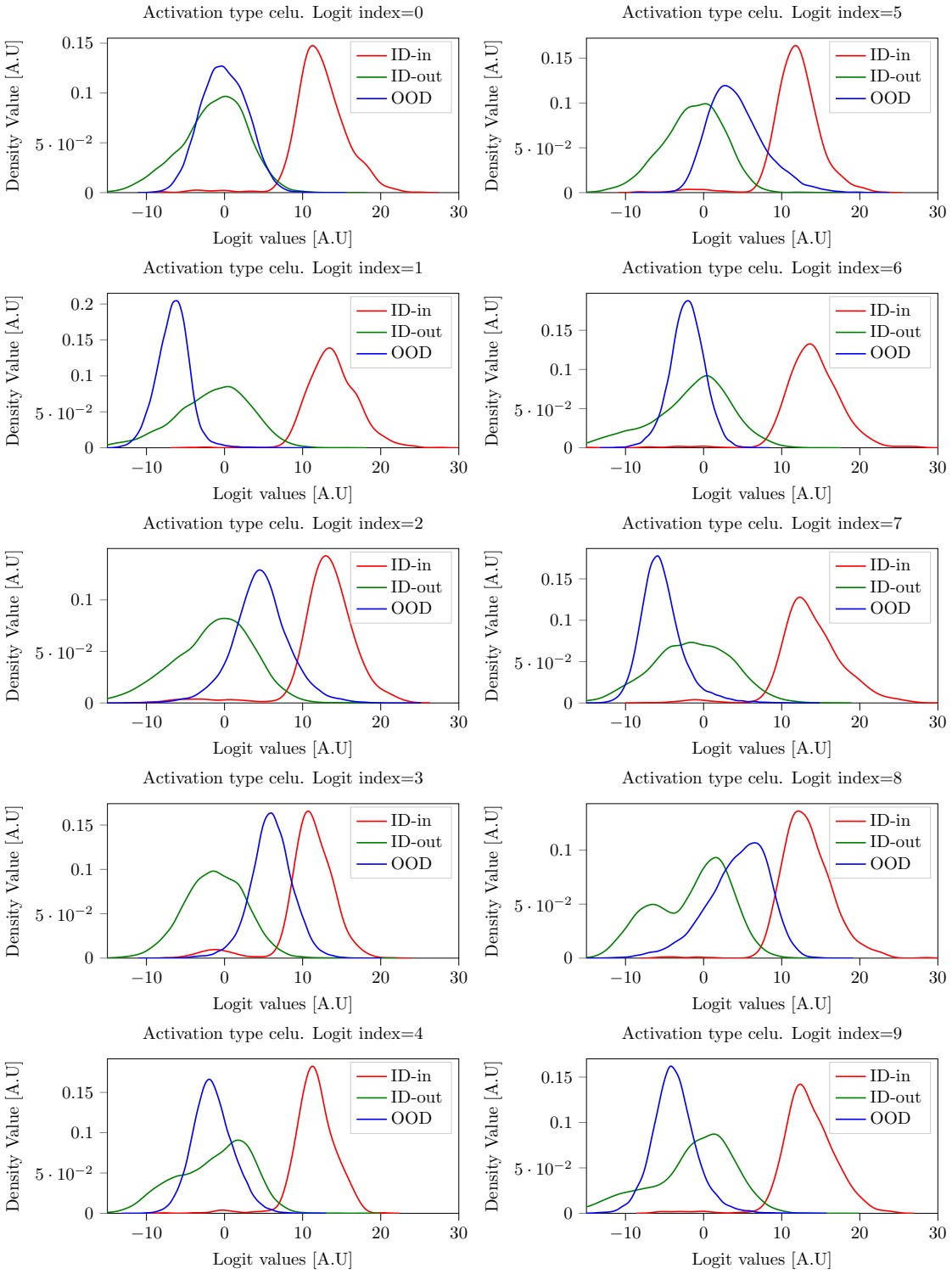

Figure 13: Densities over each logit cell from a Resnet-34 classifier with Celu activation.

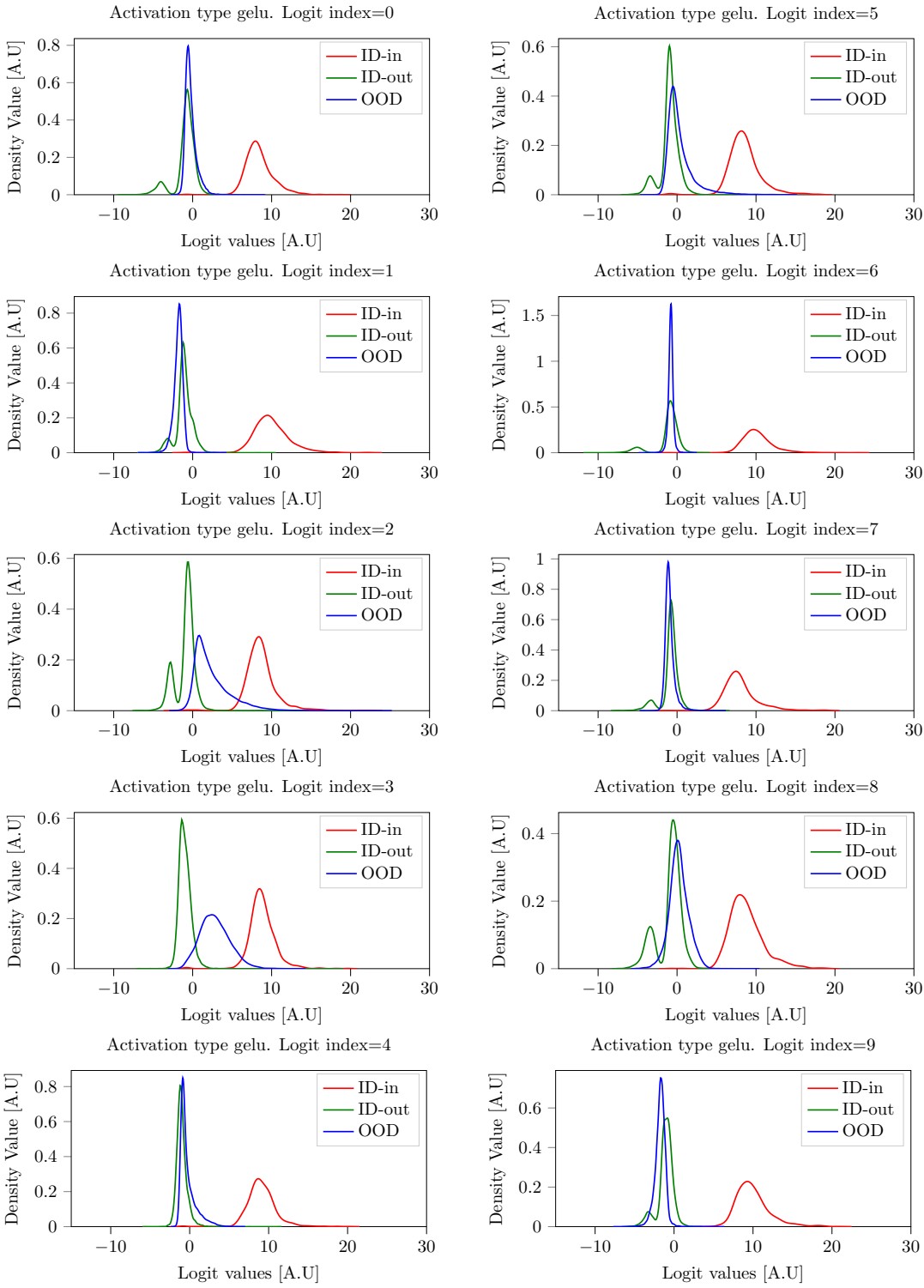

Figure 14: Densities over each logit cell from a Resnet-34 classifier with Gelu activation.

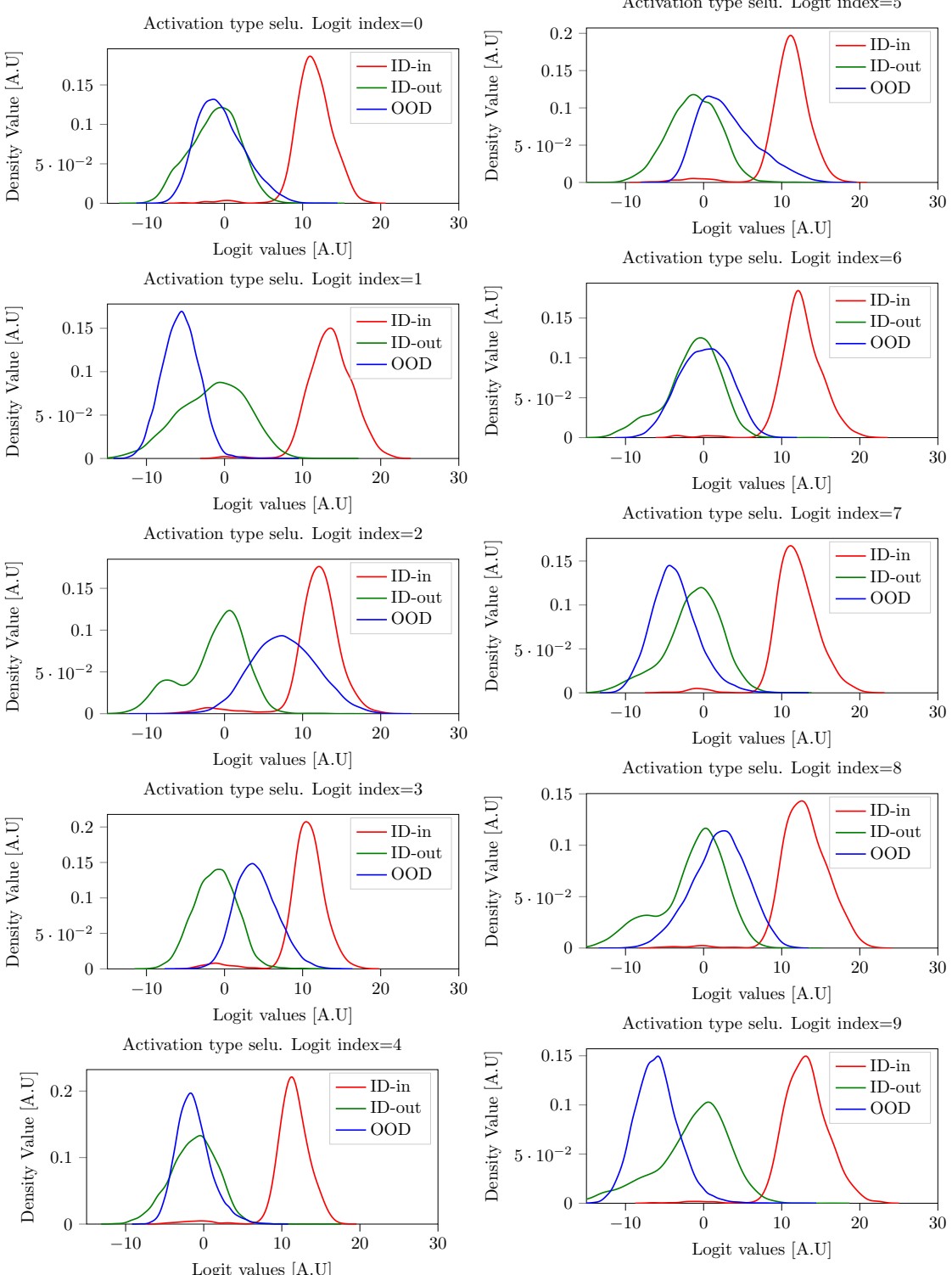

Figure 15: Densities over each logit cell from a Resnet-34 classifier with Selu activation.

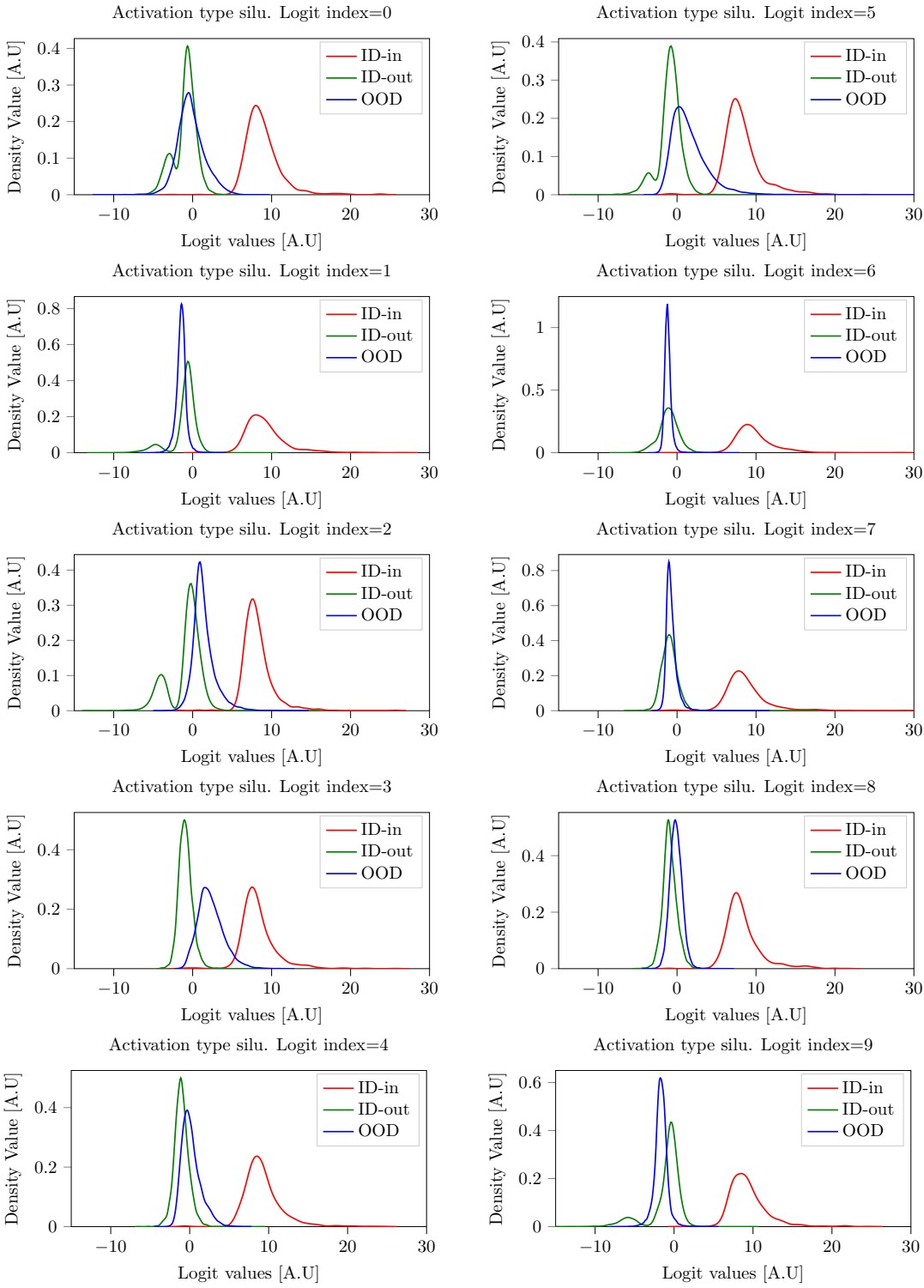

Figure 16: Densities over each logit cell from a Resnet-34 classifier with Silu activation.

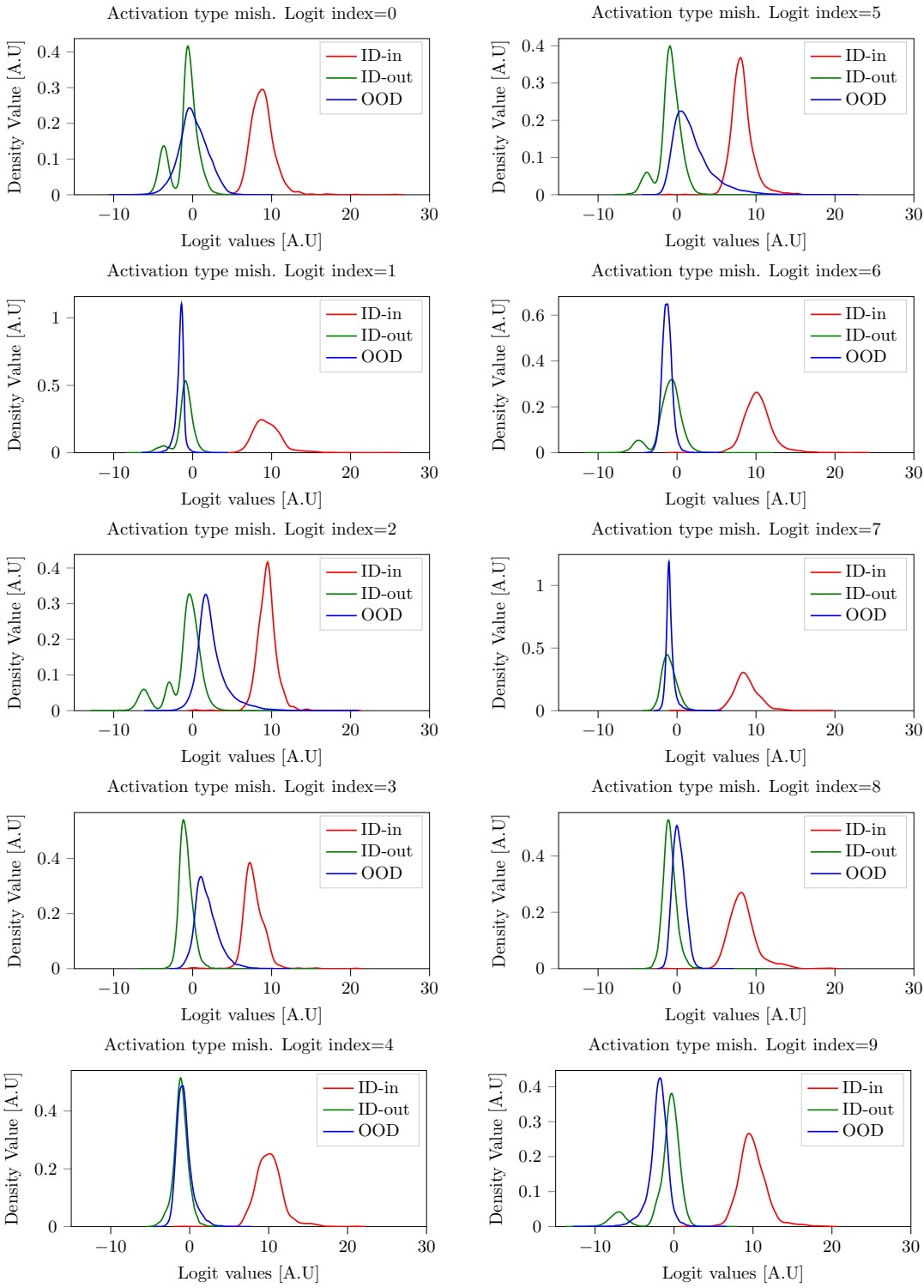

Figure 17: Densities over each logit cell from a Resnet-34 classifier with Mish activation.

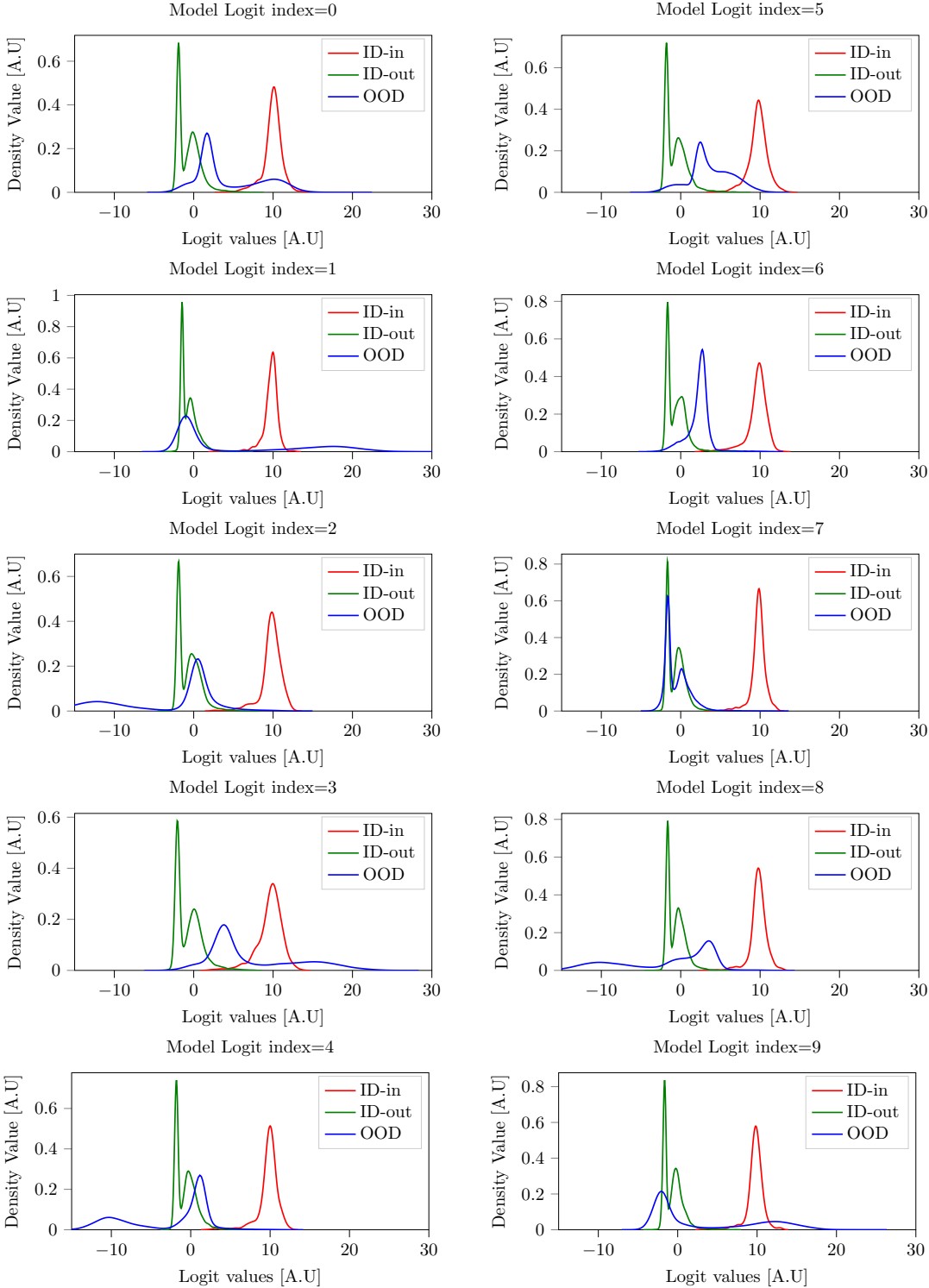

Figure 18: Densities over each logit cell from a Resnet-34 with a dropout 20% which is deactivated post train.

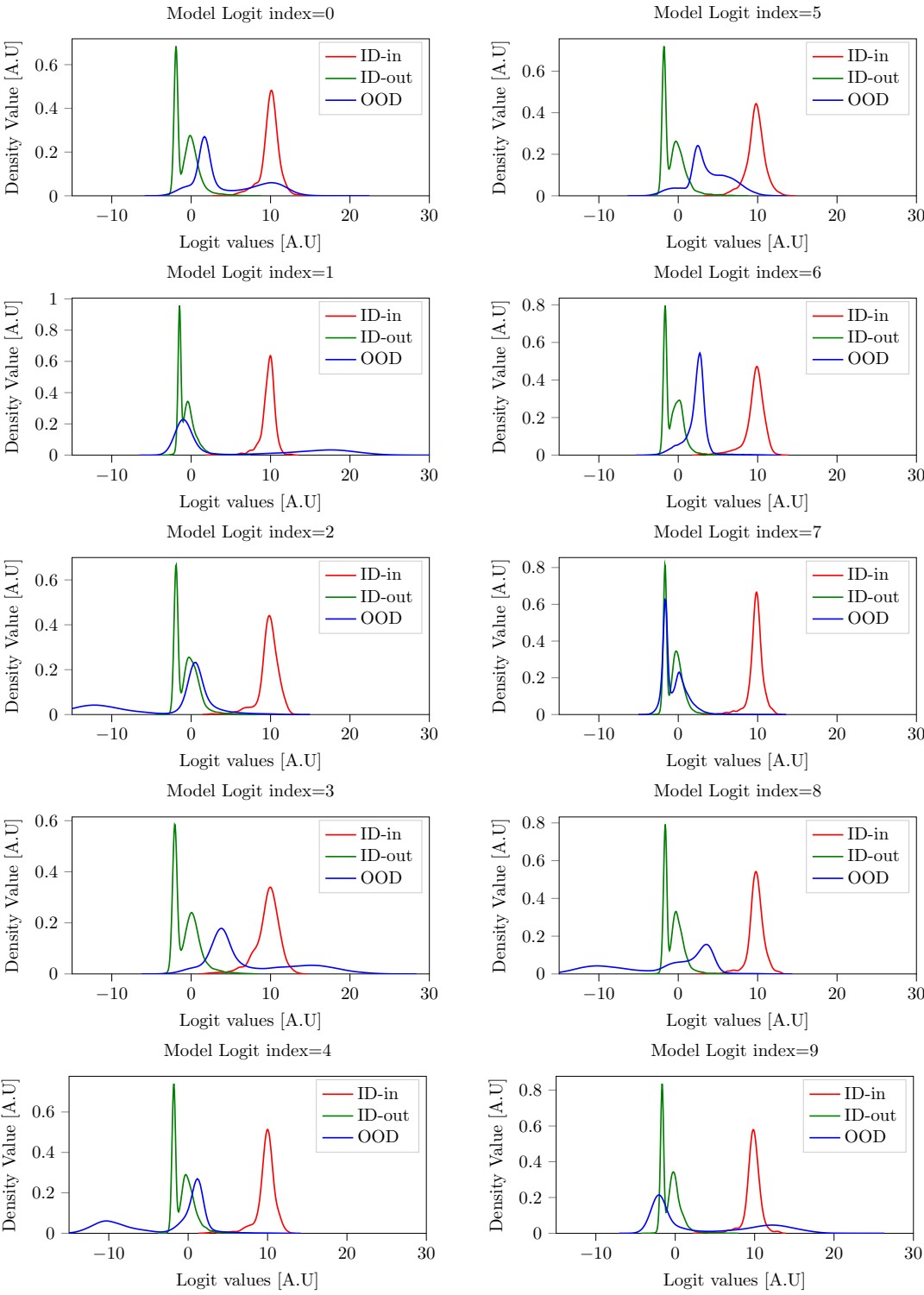

Figure 19: Densities over each logit cell from a Resnet-34 with a dropout 40% which is deactivated post train.

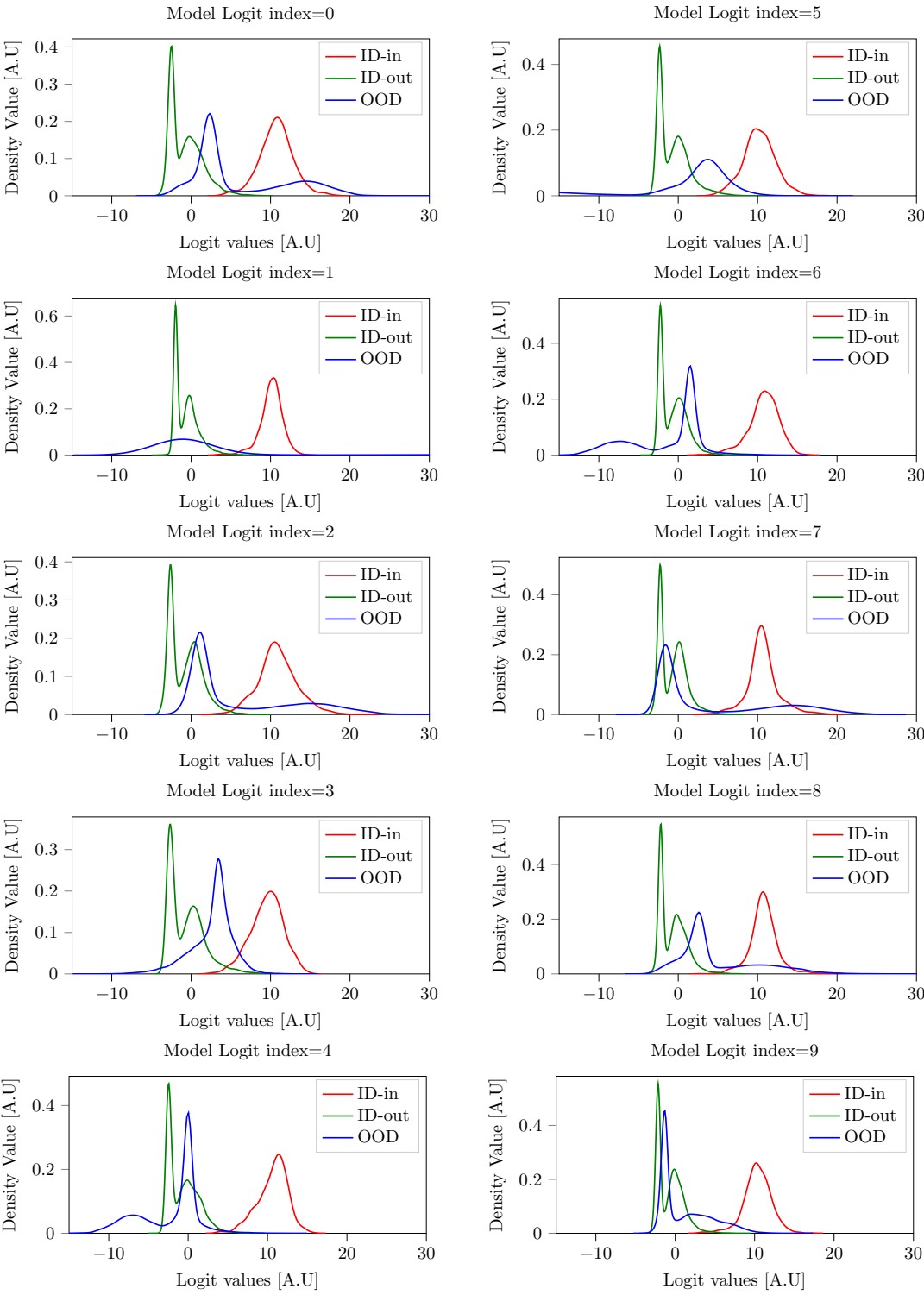

Figure 20: Densities over each logit cell from a Resnet-34 with a dropout 60% which is deactivated post train.

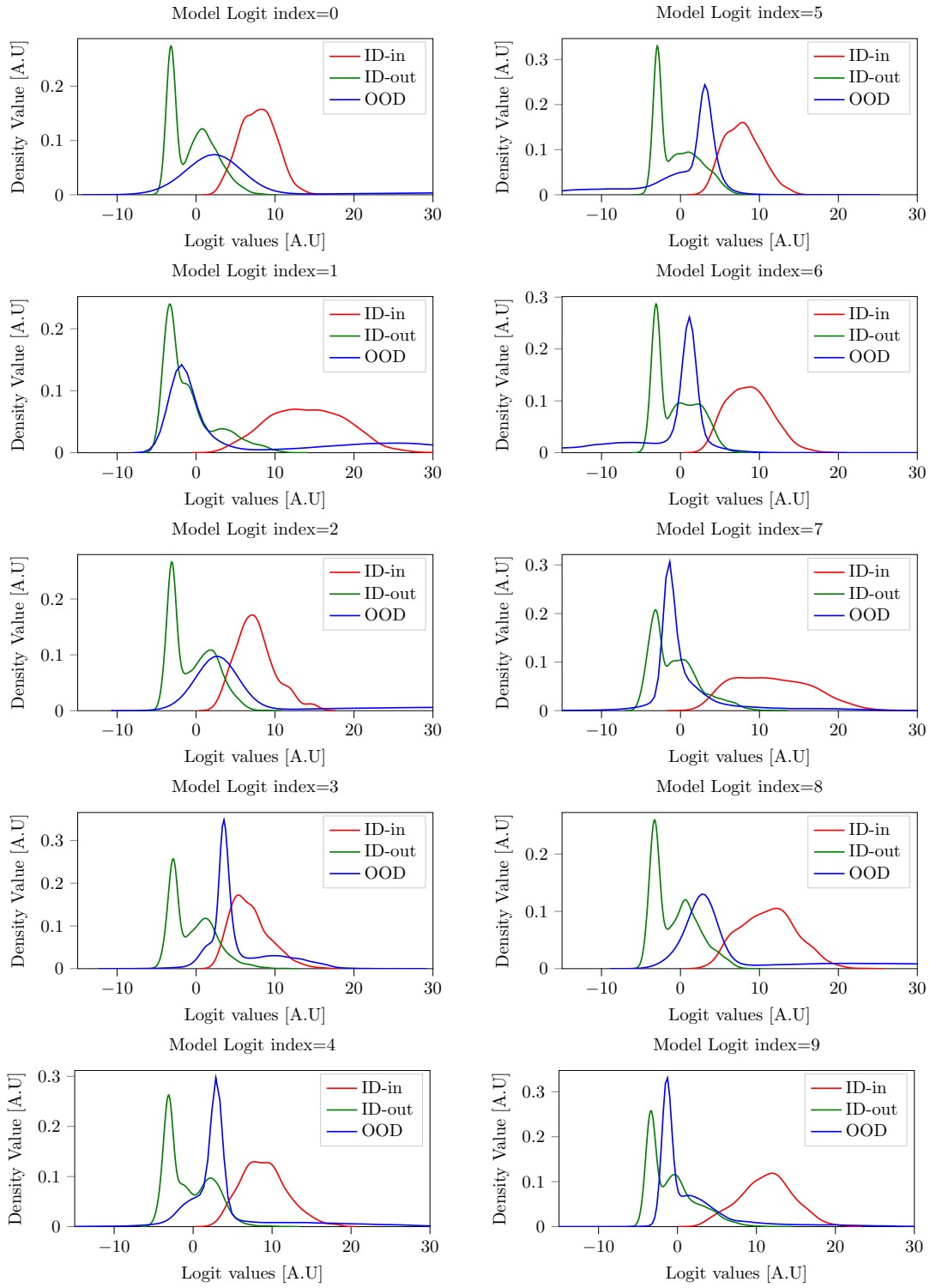

Figure 21: Densities over each logit cell from a Resnet-34 with a dropout 80% which is deactivated post train.

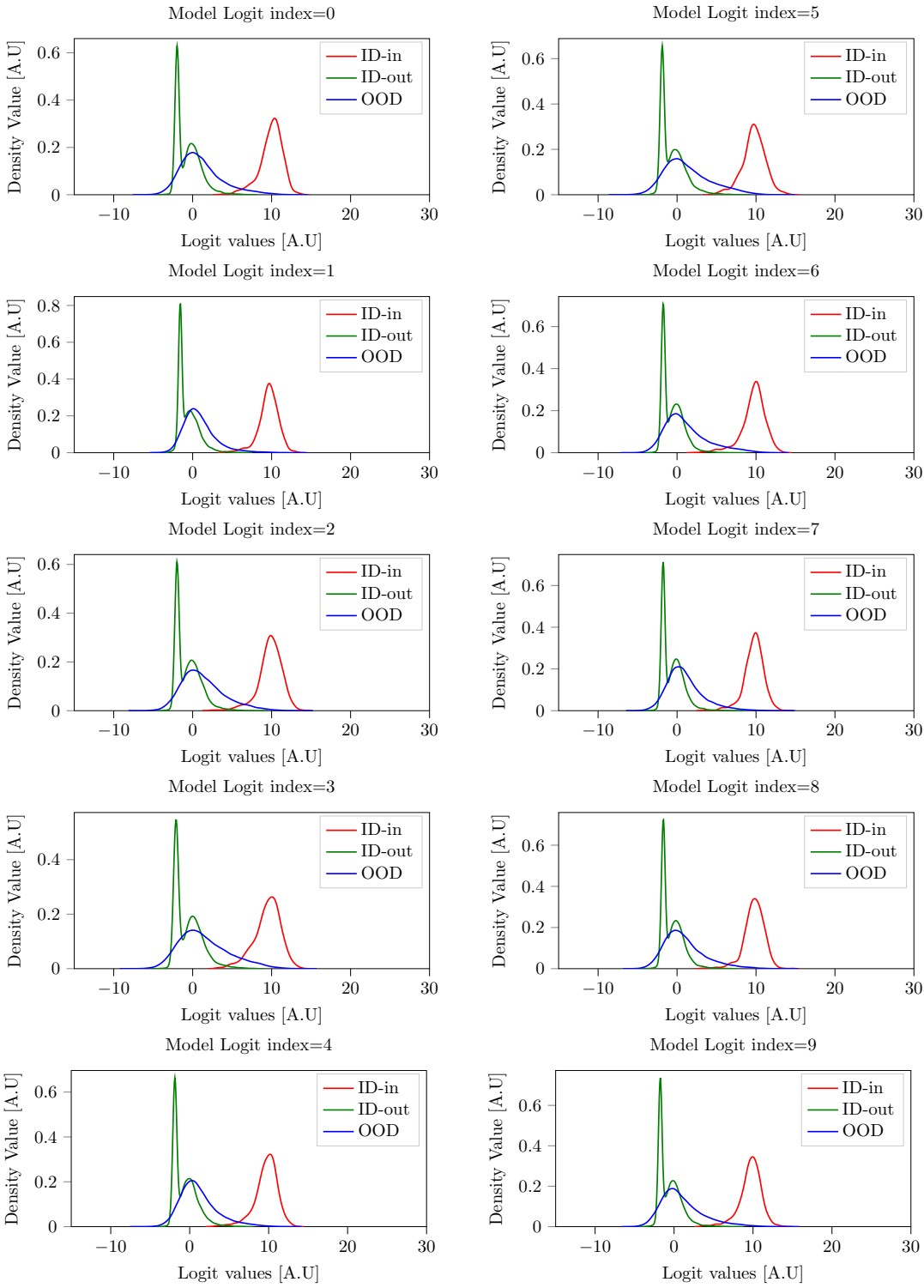

Figure 22: Densities over each logit cell from a Resnet-34 with a dropout 20%, which remains activated post train.

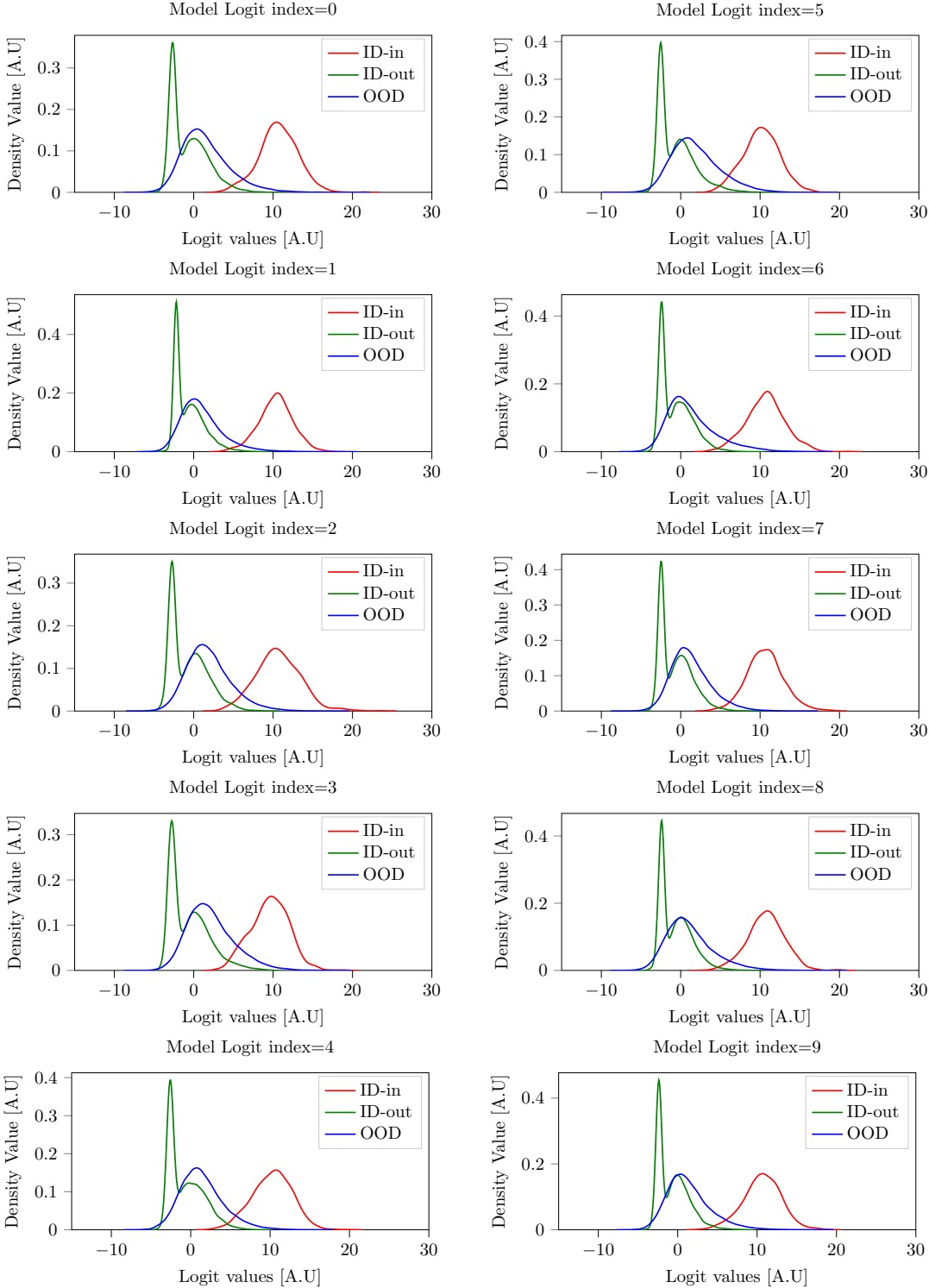

Figure 23: Densities over each logit cell from a Resnet-34 with a dropout 40%, which remains activated post train.

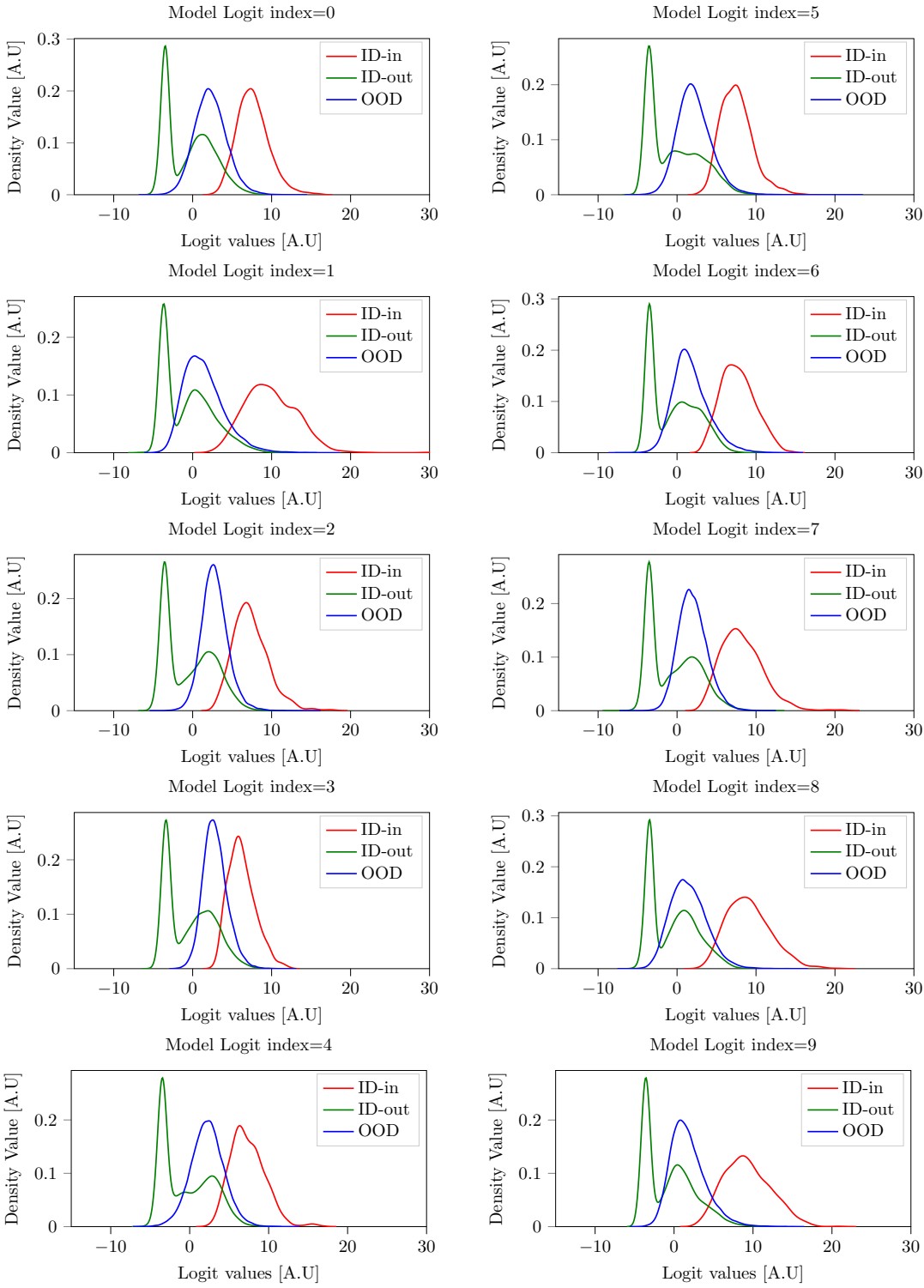

Figure 24: Densities over each logit cell from a Resnet-34 with a dropout 60%, which remains activated post train.

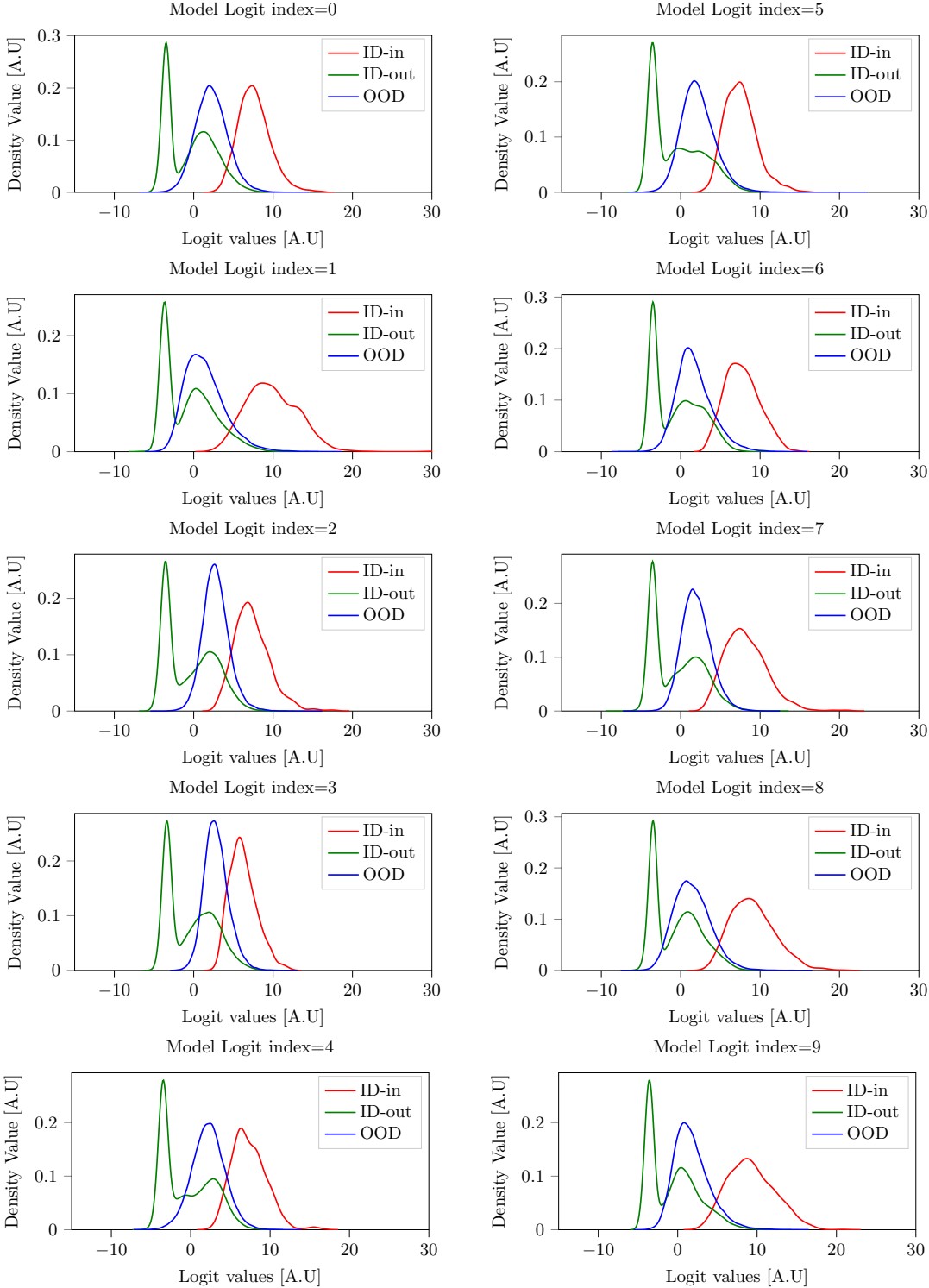

Figure 25: Densities over each logit cell from a Resnet-34 with a dropout 80%, which remains activated post train.

# E Experiments on different classifiers

Moreover, figs. 28 to 45 offer a detailed visualization of the ID and OOD logits for each cell across various versions of Densenet and Resnet.

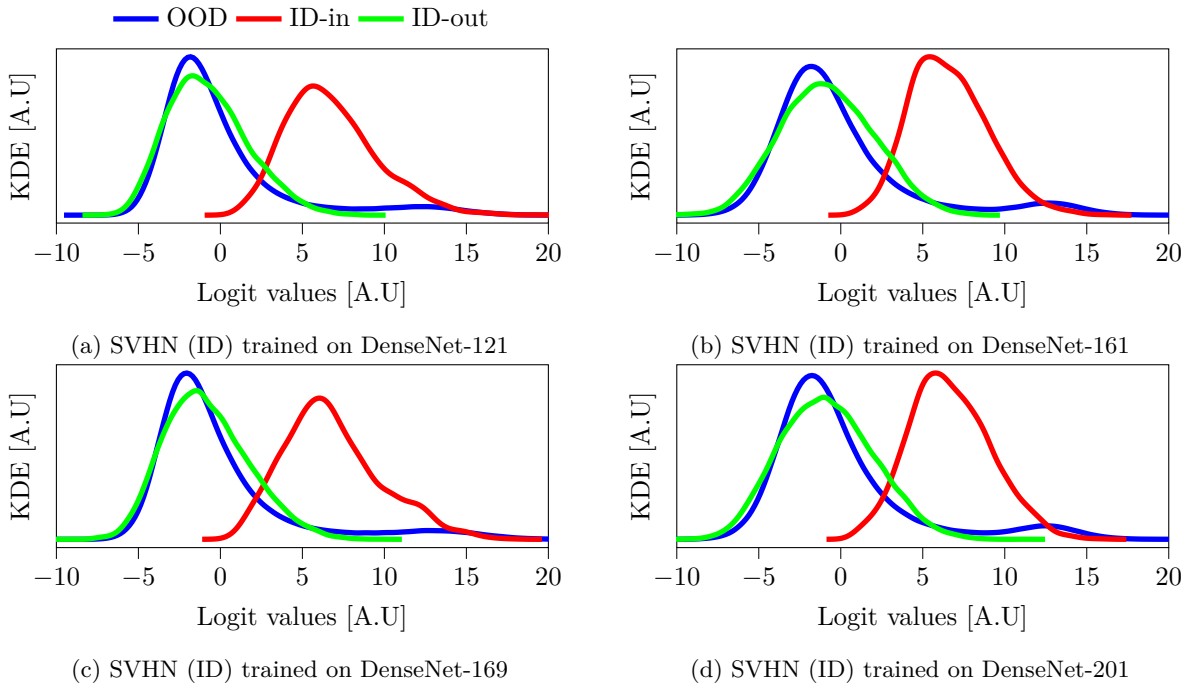

(a) SVHN (ID) trained on DenseNet-121

(b) SVHN (ID) trained on DenseNet-161

(c) SVHN (ID) trained on DenseNet-169

(d) SVHN (ID) trained on DenseNet-201

Figure 26: An analysis of the density over aggregated logits across distinct DenseNet architecture trained on the SVHN dataset as the ID data, while the OOD includes $\{D\}$/SVHN. For a more detailed comparison, check figs. 28 to 35

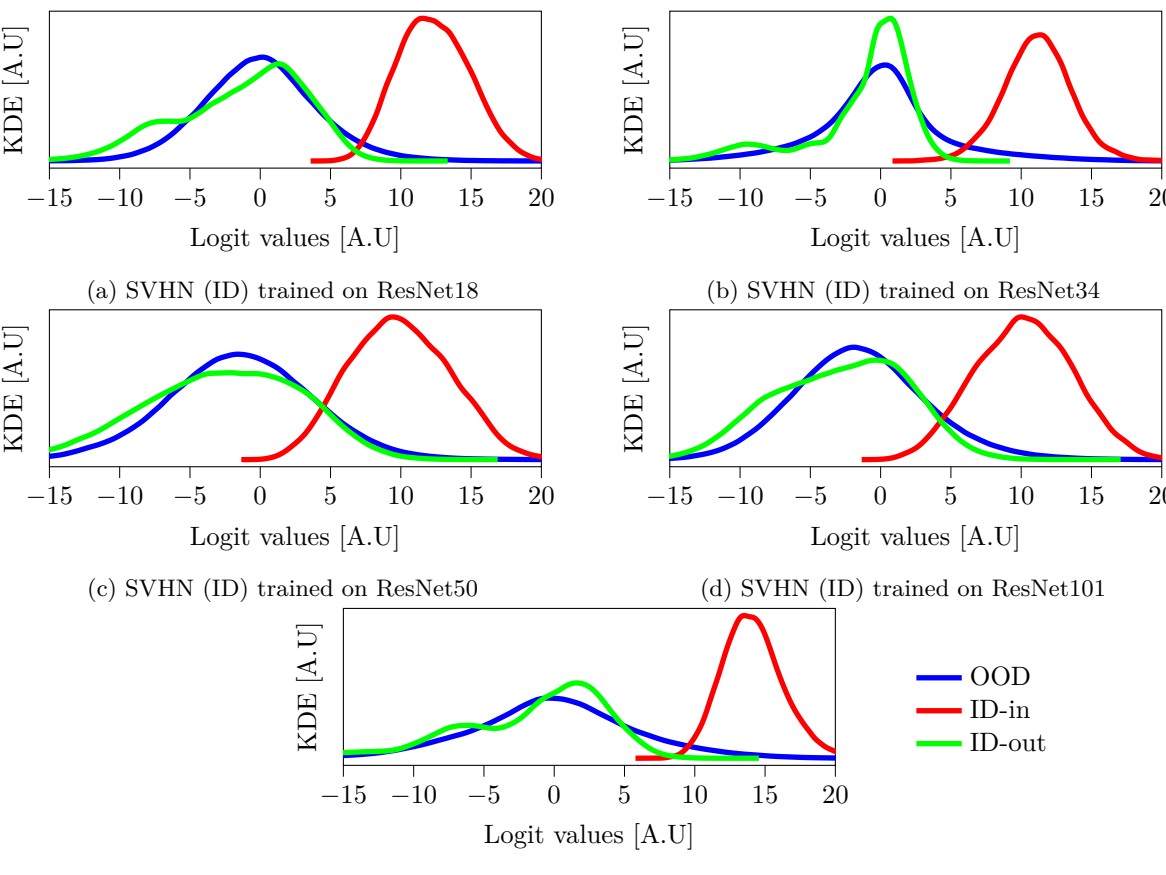

(a) SVHN (ID) trained on ResNet18

(b) SVHN (ID) trained on ResNet34

(c) SVHN (ID) trained on ResNet50

(d) SVHN (ID) trained on ResNet101

(e) SVHN (ID) trained on ResNet152

Figure 27: An analysis of the density over aggregated logits across distinct DenseNet architecture trained on the CIFAR-10 dataset as the ID data, while the OOD includes $\{D\}$/CIFAR-10. For a more detailed comparison, check figs. 28 to 35.

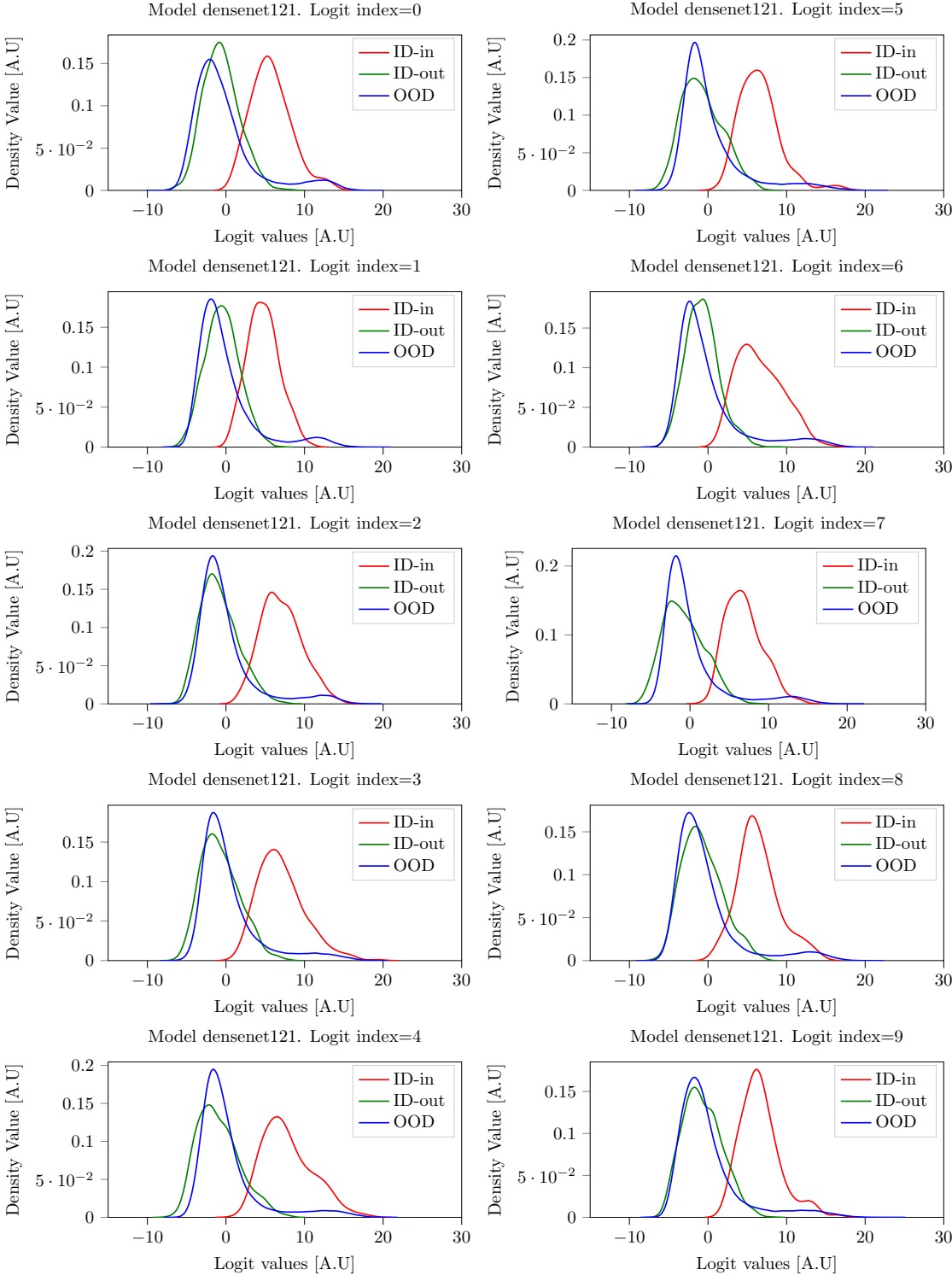

Figure 28: Logit cell densities for SVHN as ID with Densenet121.

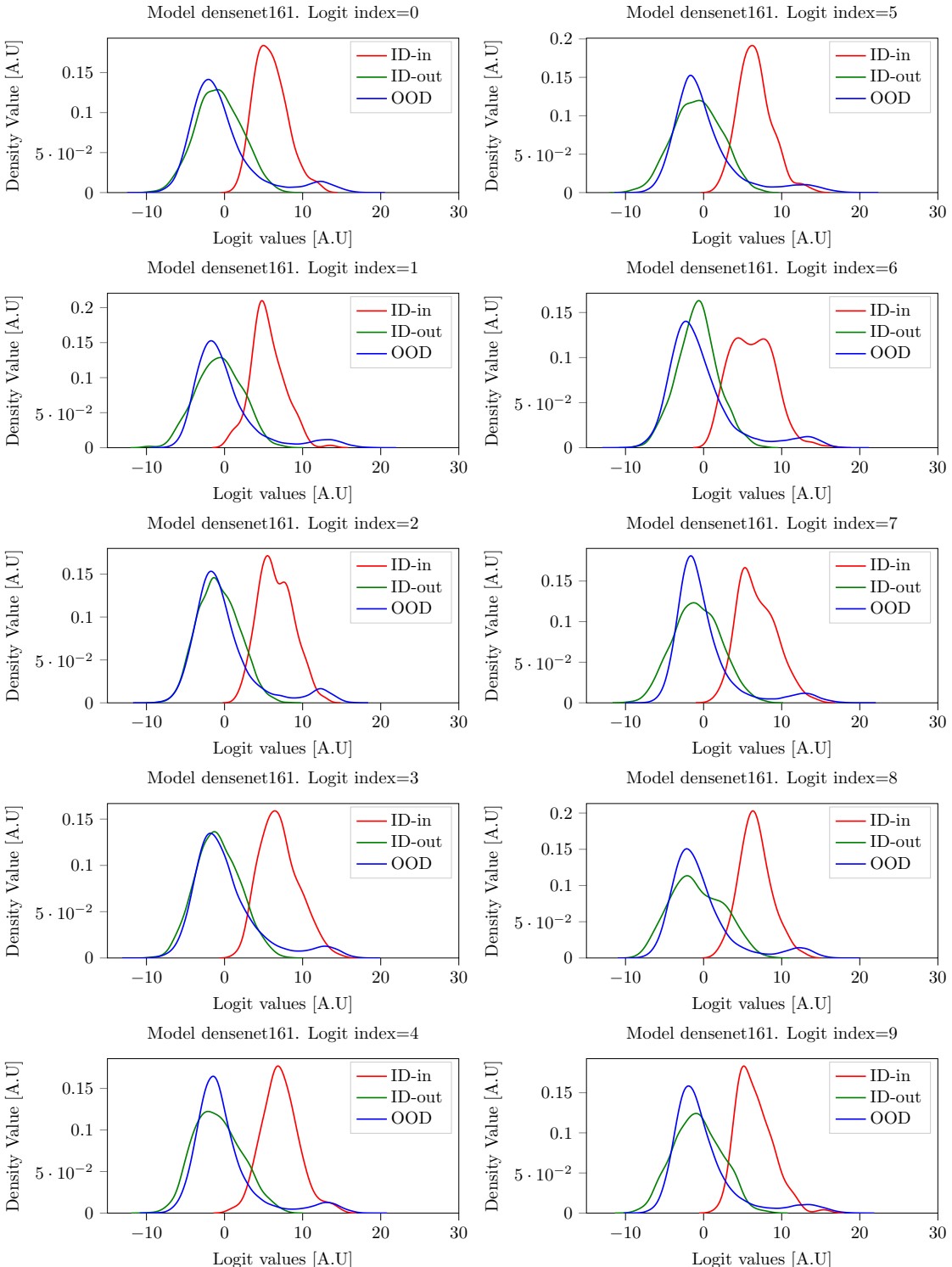

Figure 29: Logit cell densities for SVHN as ID with Densenet161.

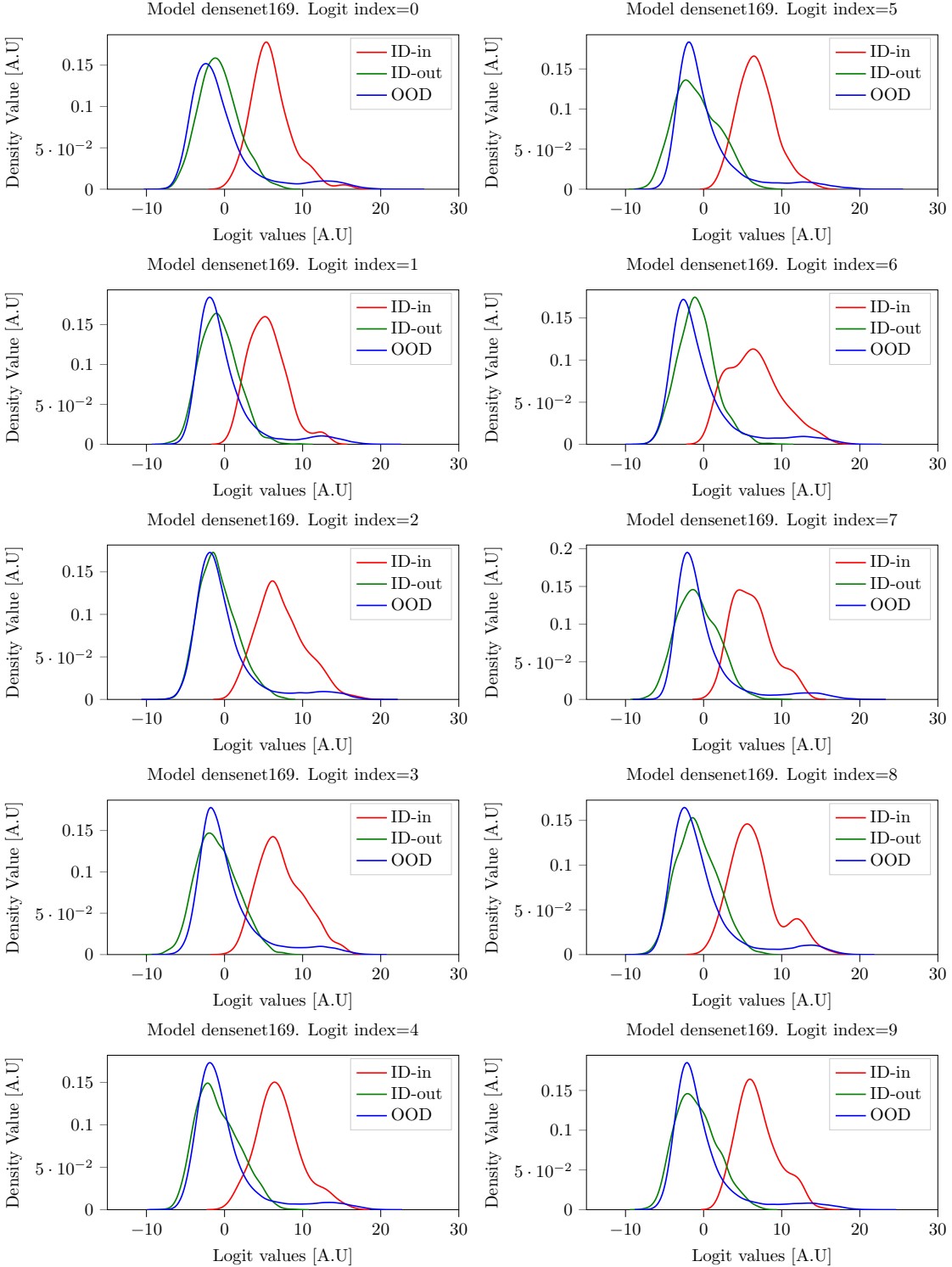

Figure 30: Logit cell densities for SVHN as ID with Densenet169.

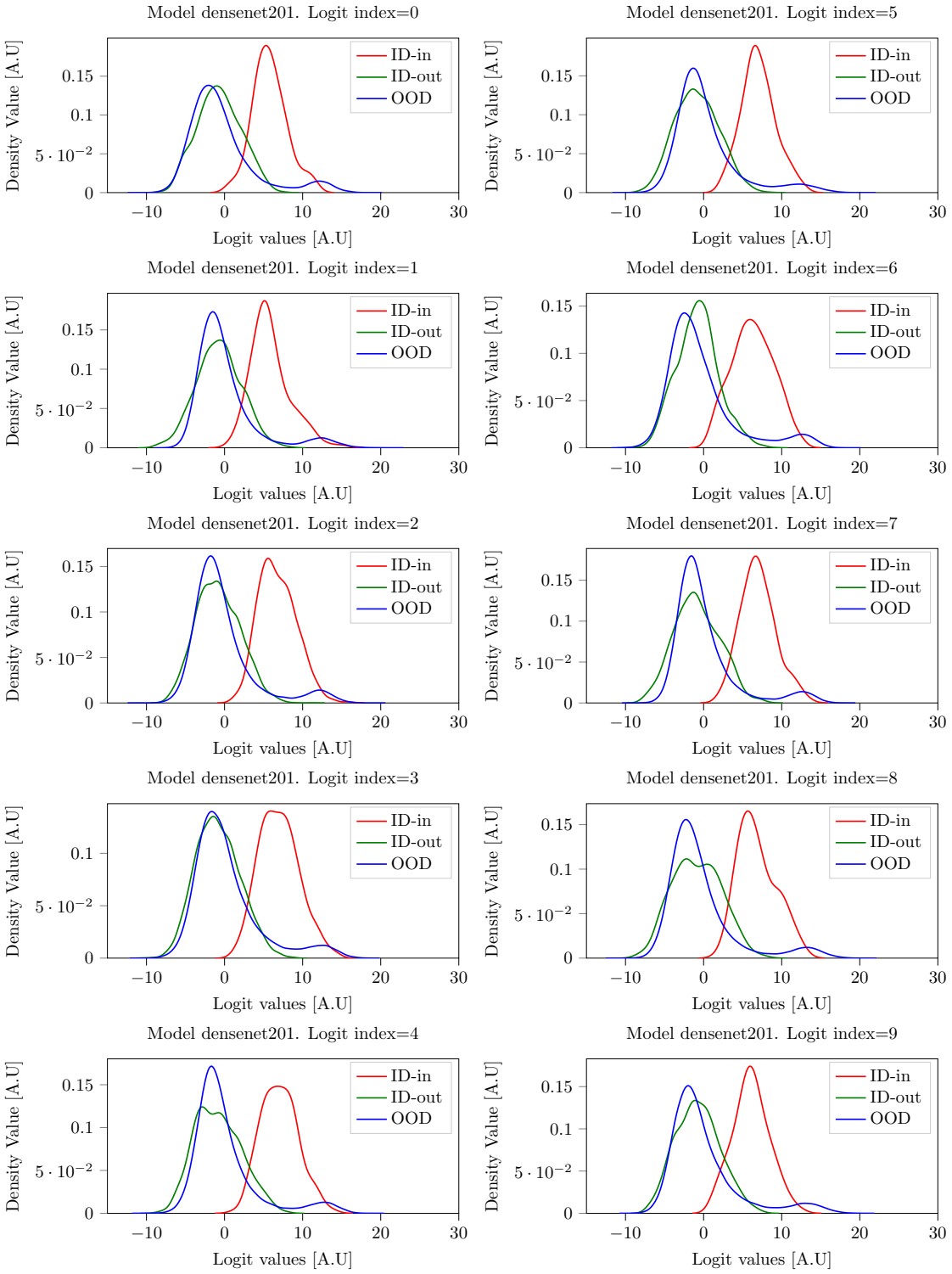

Figure 31: Logit cell densities for SVHN as ID with Densenet201.

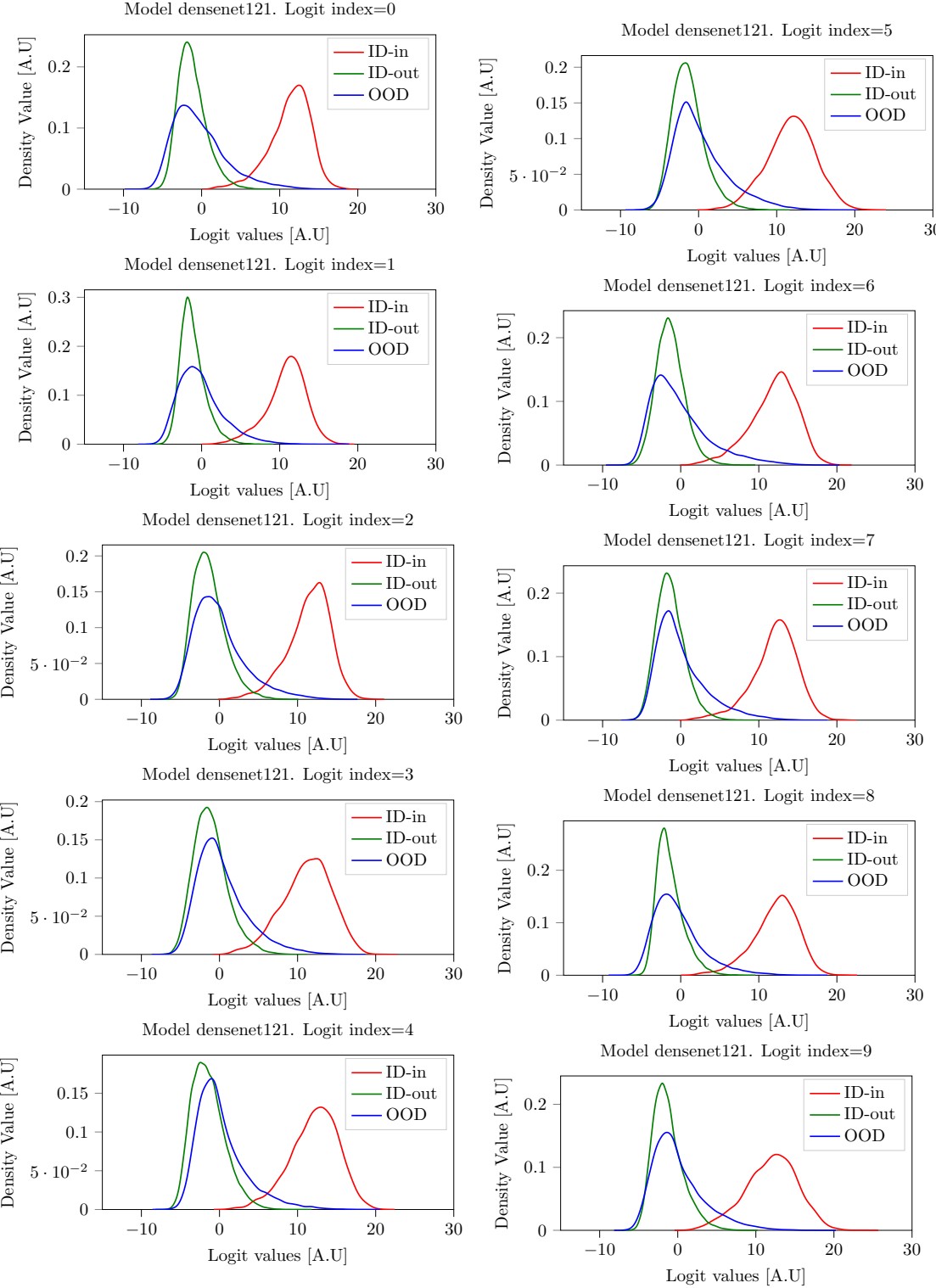

Figure 32: Logit cell densities for CIFAR-10 as ID with Densenet121.

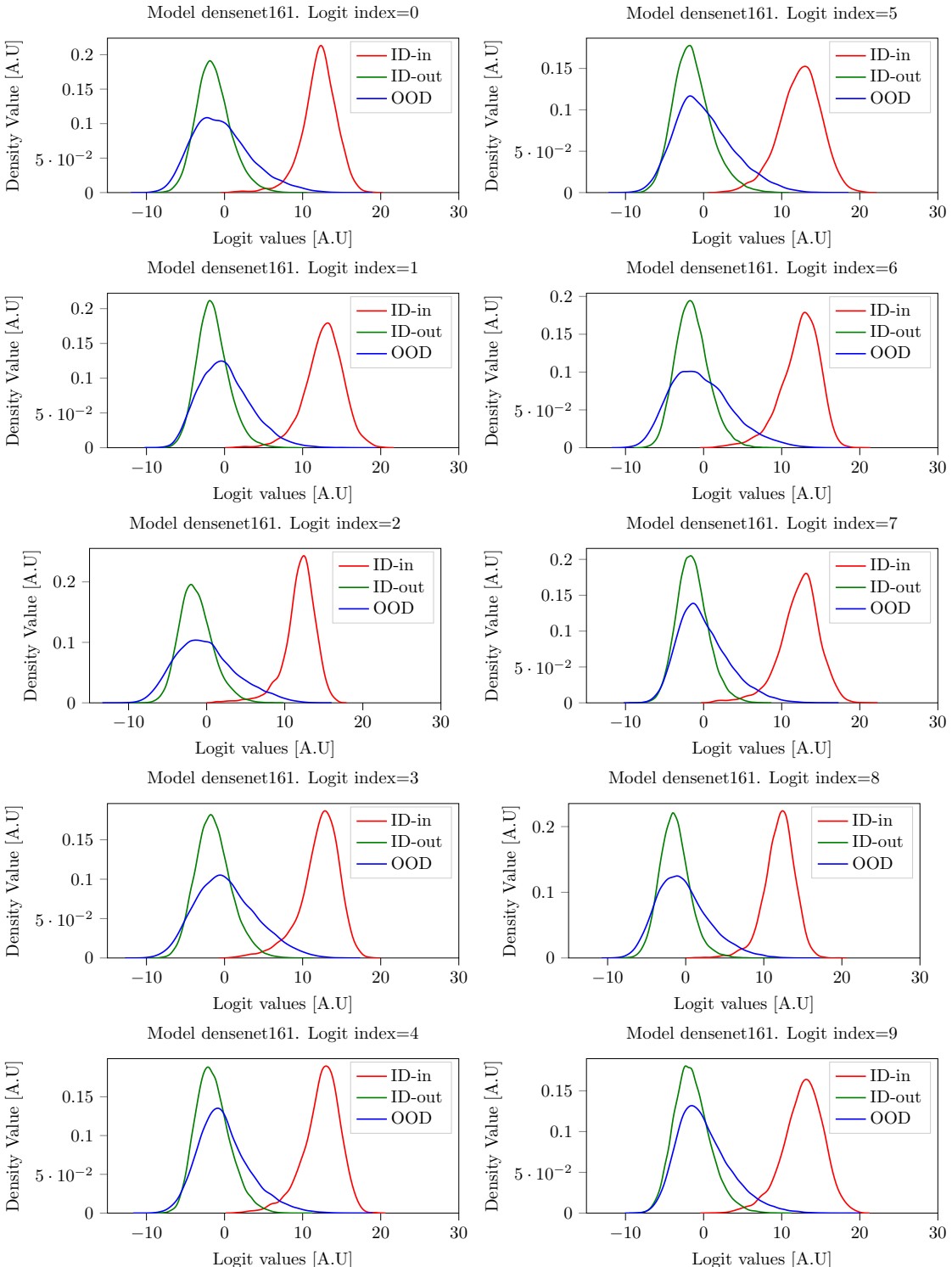

Figure 33: Logit cell densities for CIFAR-10 as ID with Densenet161.

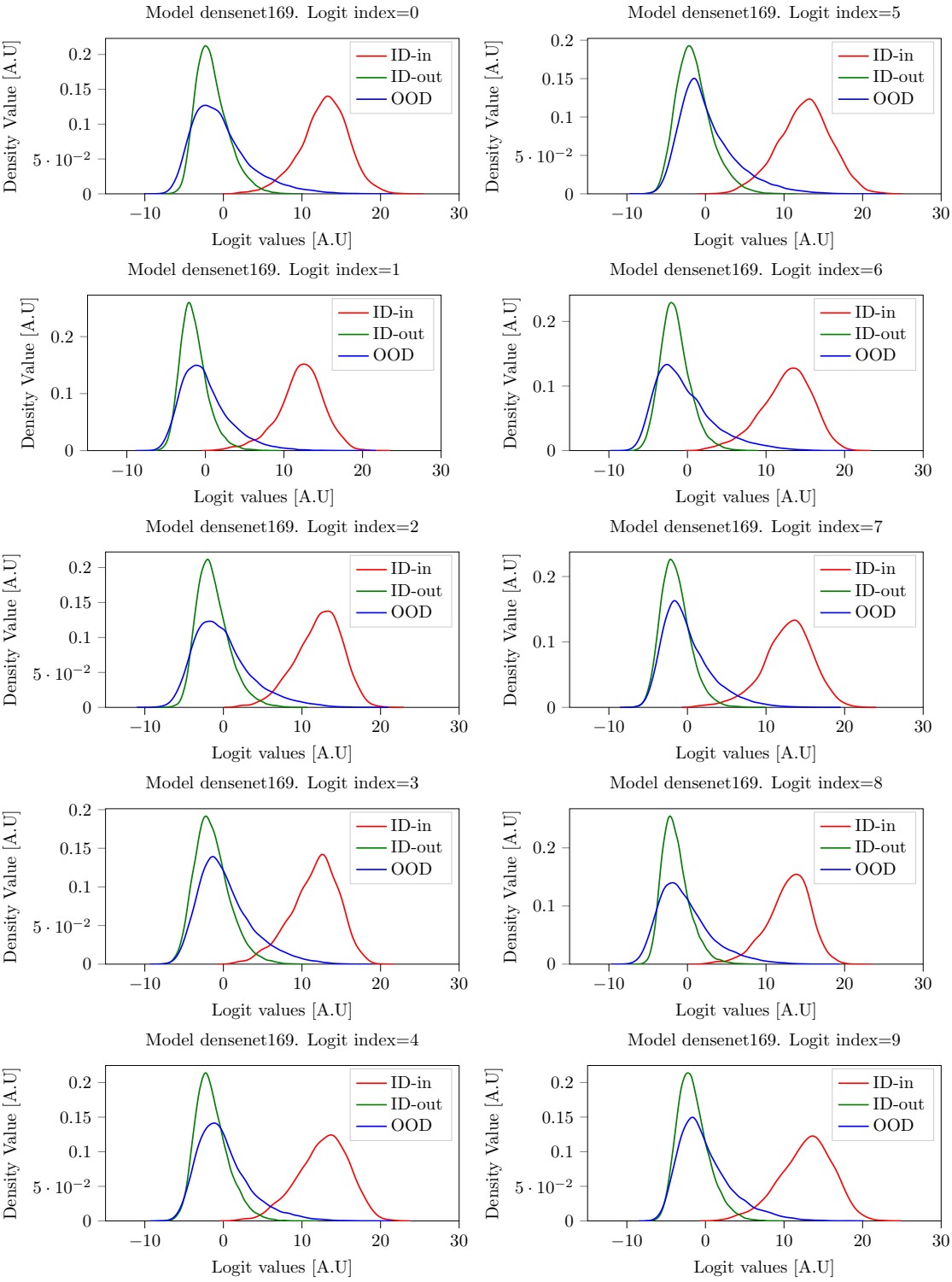

Figure 34: Logit cell densities for CIFAR-10 as ID with Densenet169.

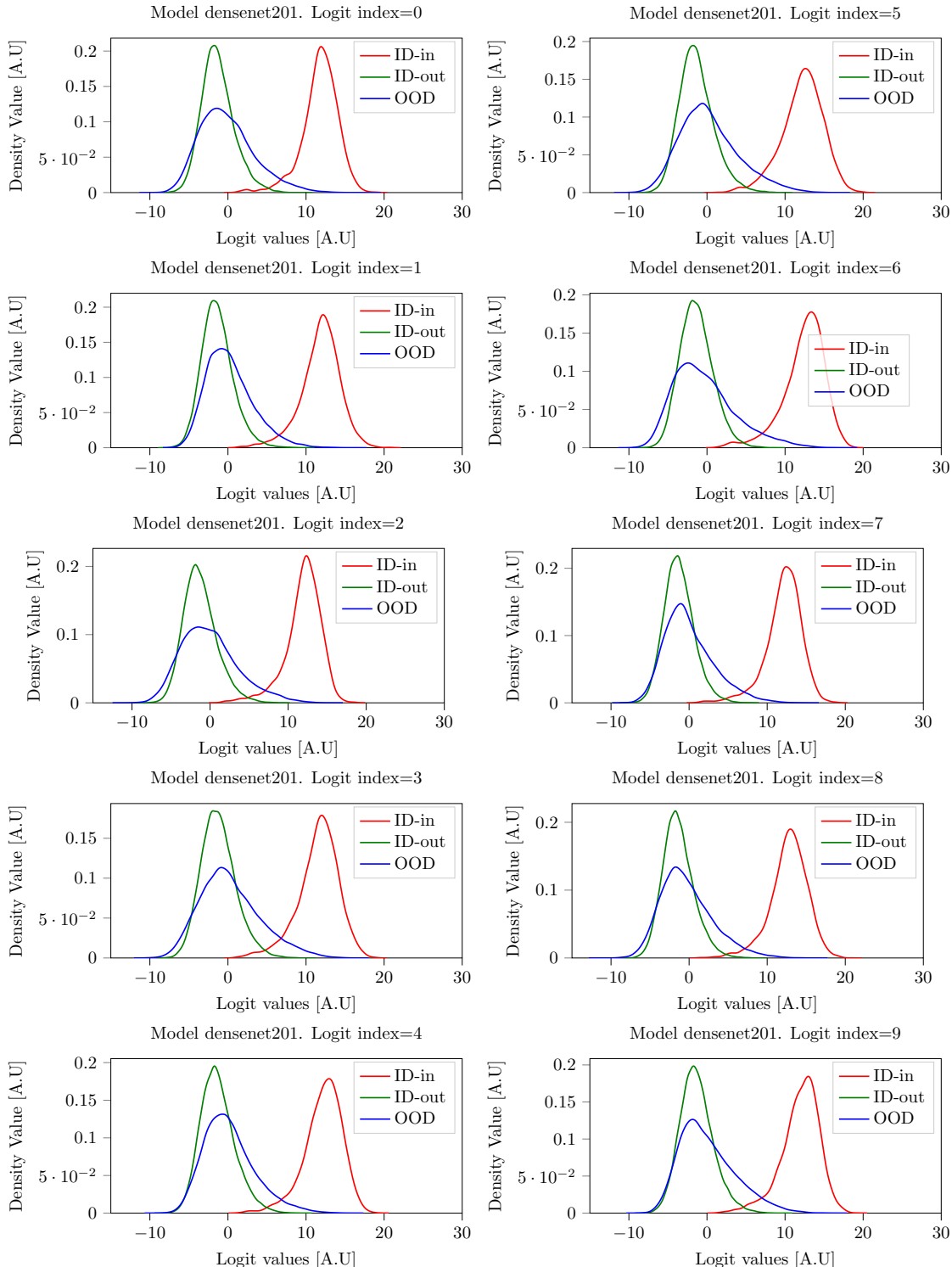

Figure 35: Logit cell densities for CIFAR-10 as ID with Densenet201.

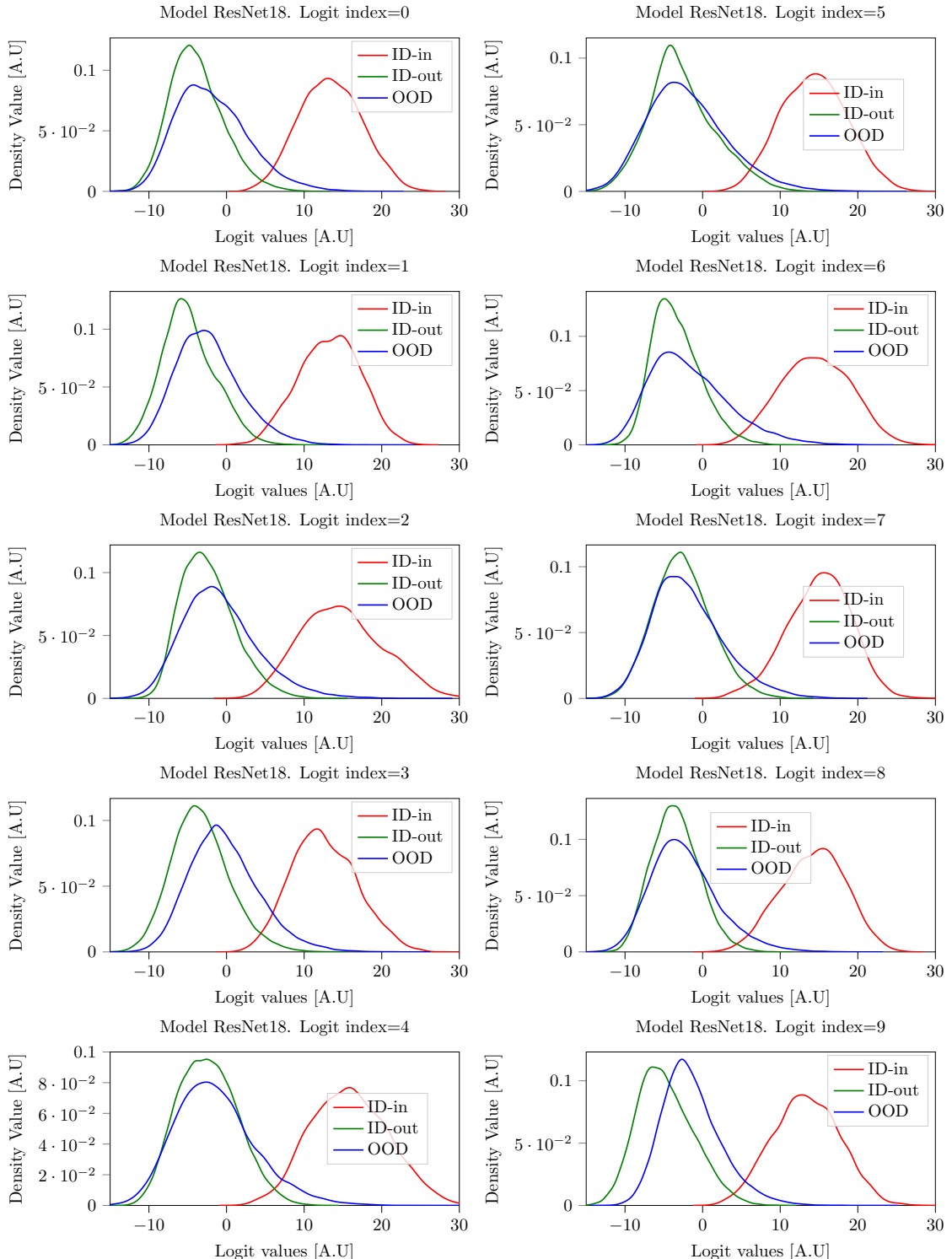

Figure 36: Logit cell densities for CIFAR-10 as ID with ResNet18.

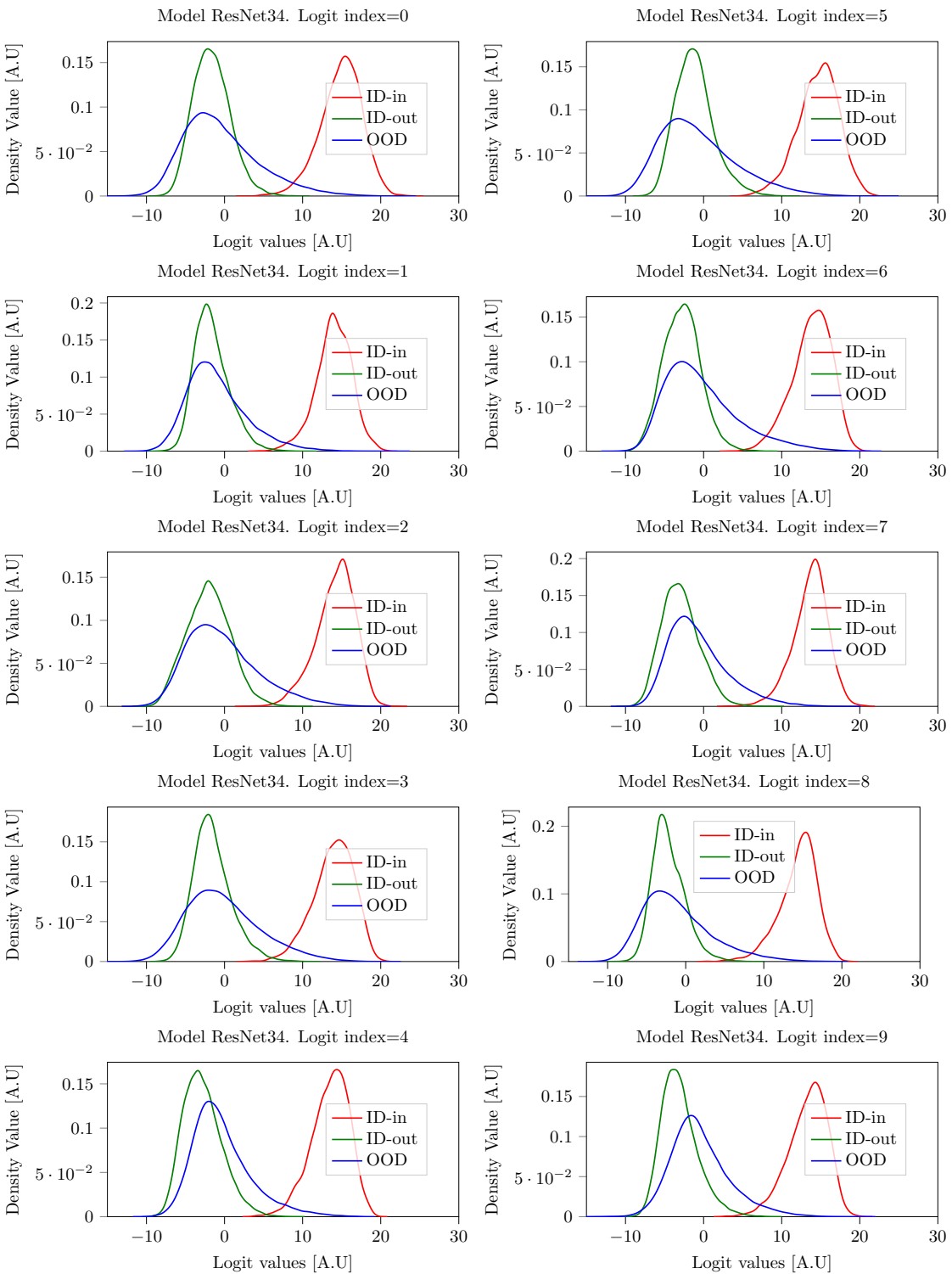

Figure 37: Logit cell densities for CIFAR-10 as ID with ResNet34.

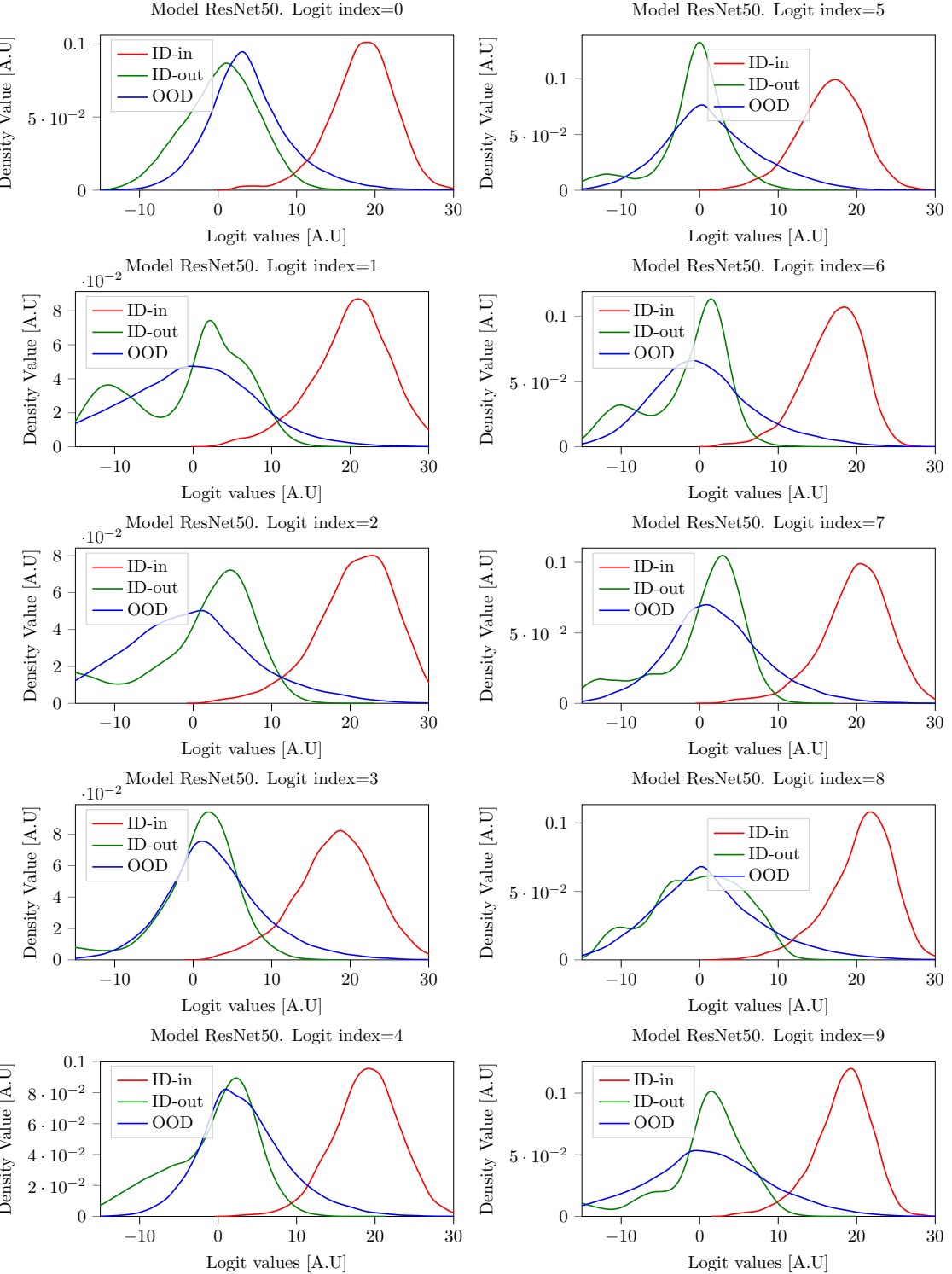

Figure 38: Logit cell densities for CIFAR-10 as ID with ResNet50.

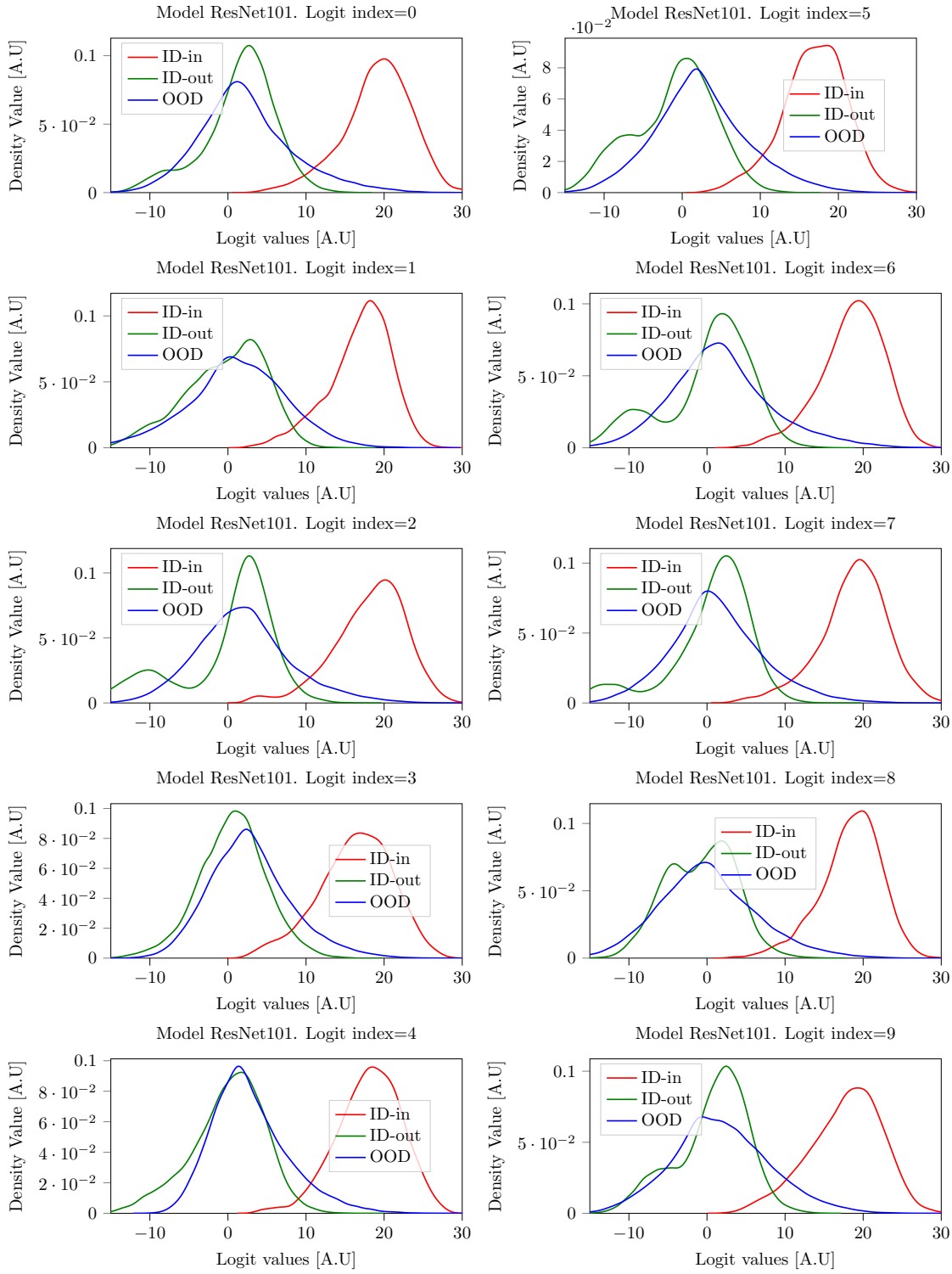

Figure 39: Logit cell densities for CIFAR-10 as ID with ResNet101.

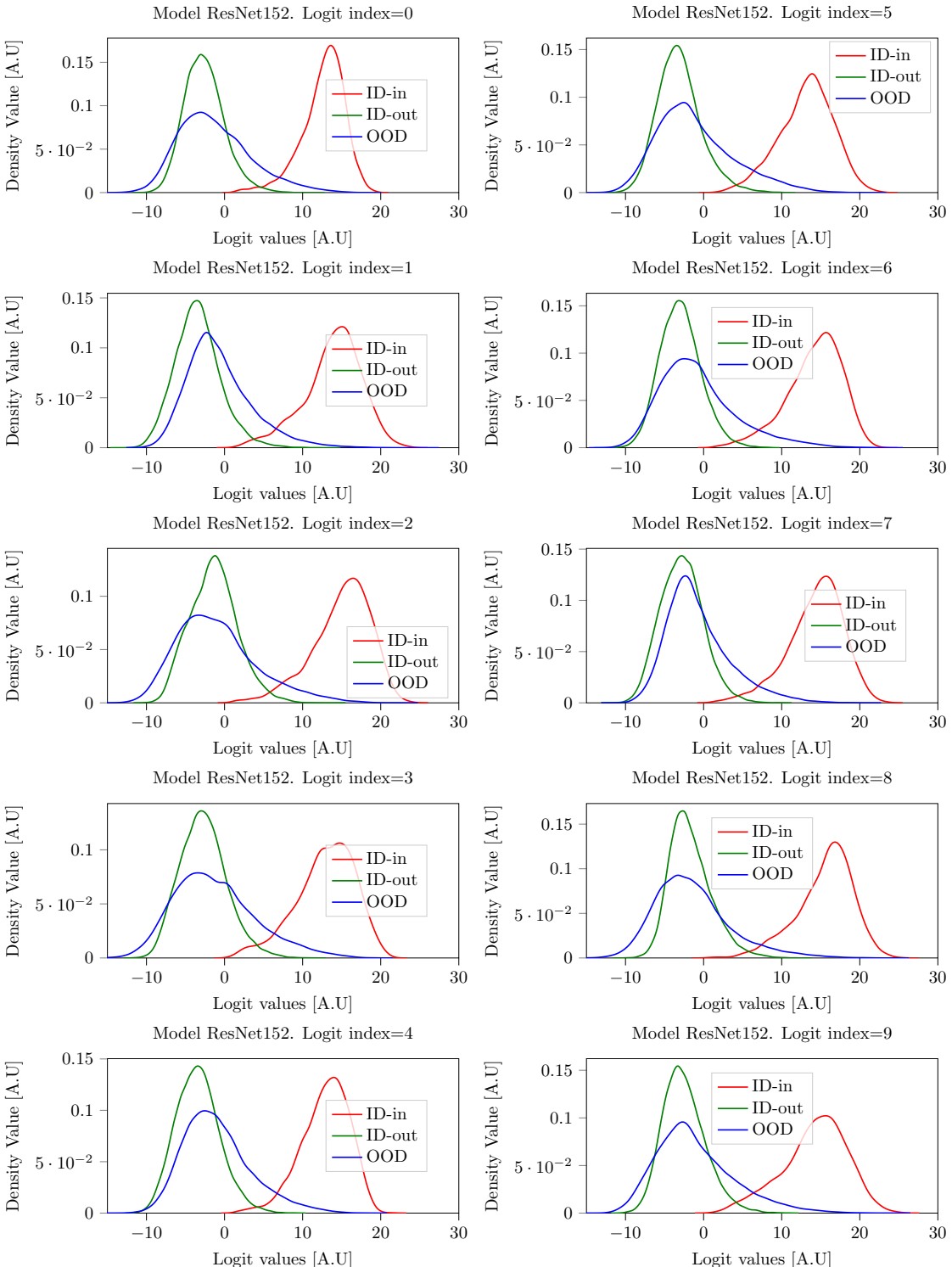

Figure 40: Logit cell densities for CIFAR-10 as ID with ResNet152.

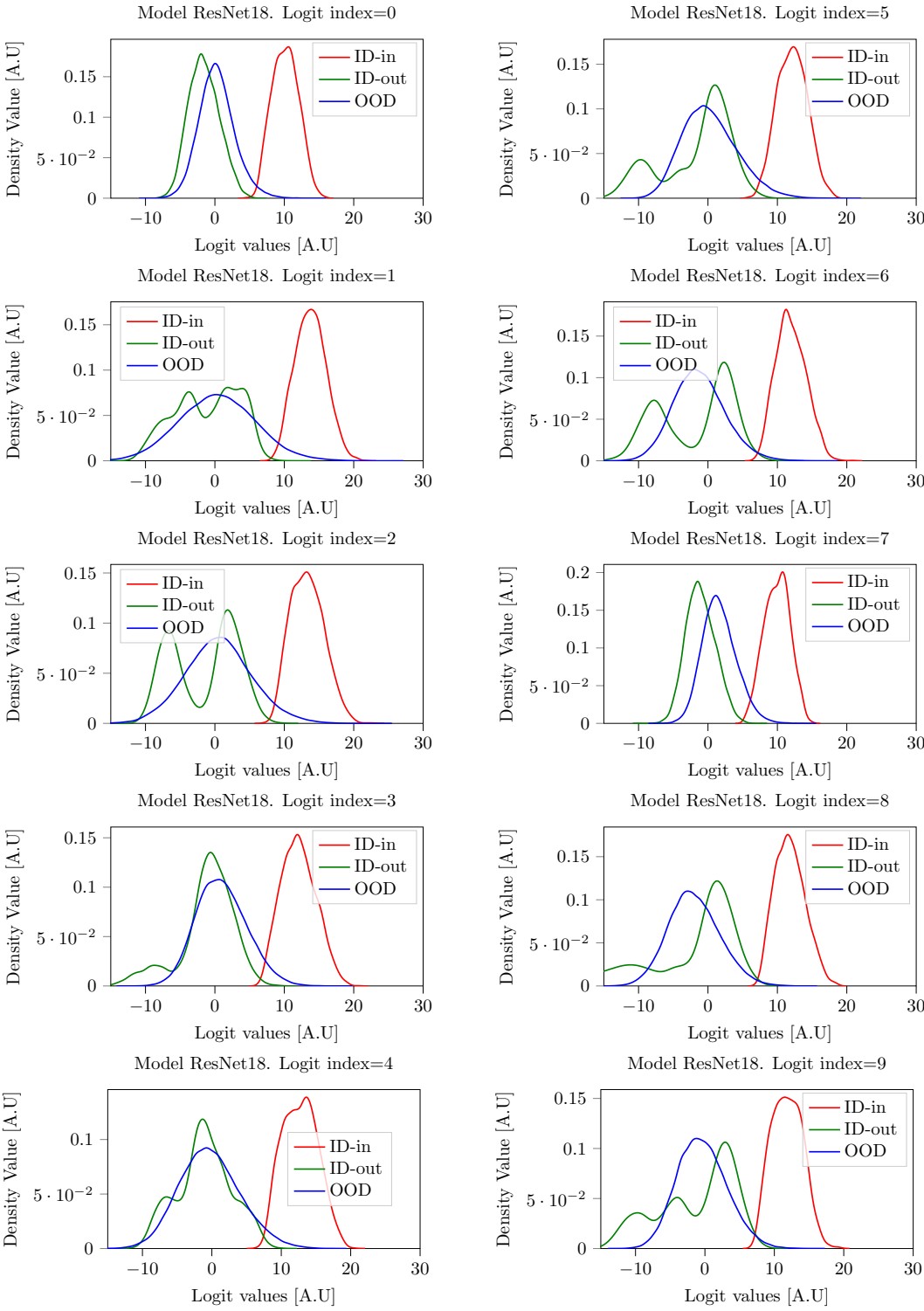

Figure 41: Logit cell densities for SVHN as ID with ResNet18.

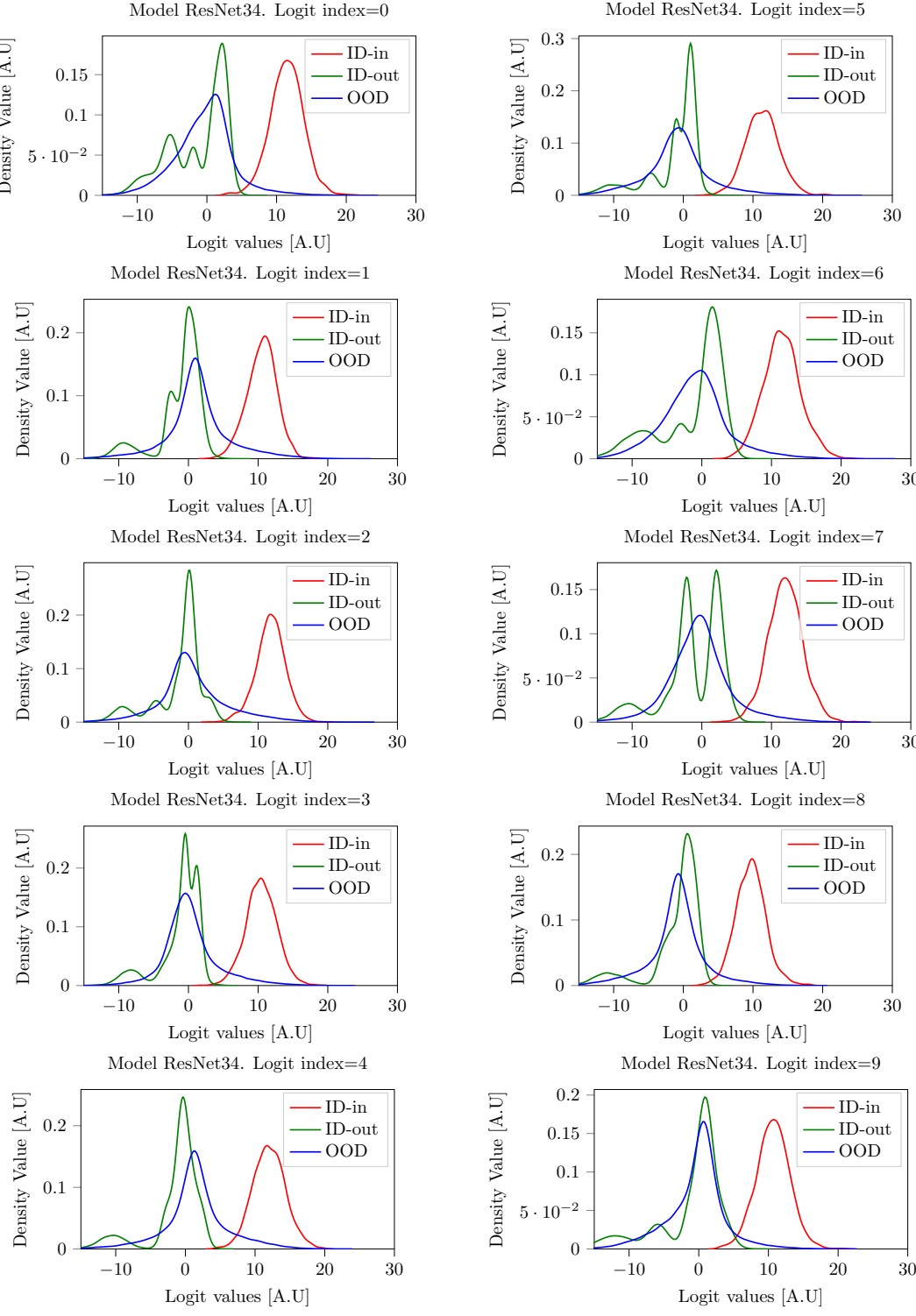

Figure 42: Logit cell densities for SVHN as ID with ResNet34.

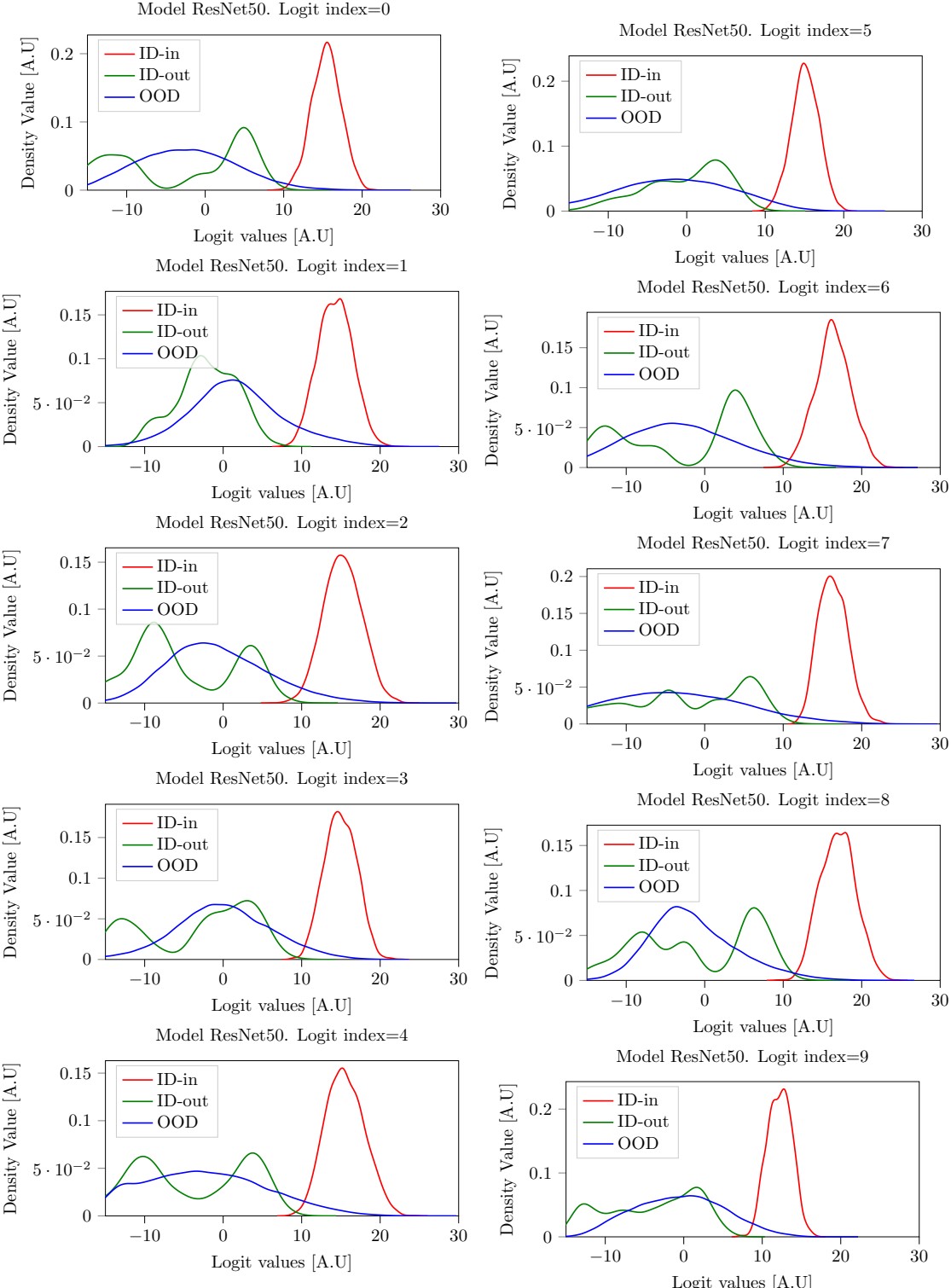

Figure 43: Logit cell densities for SVHN as ID with ResNet50.

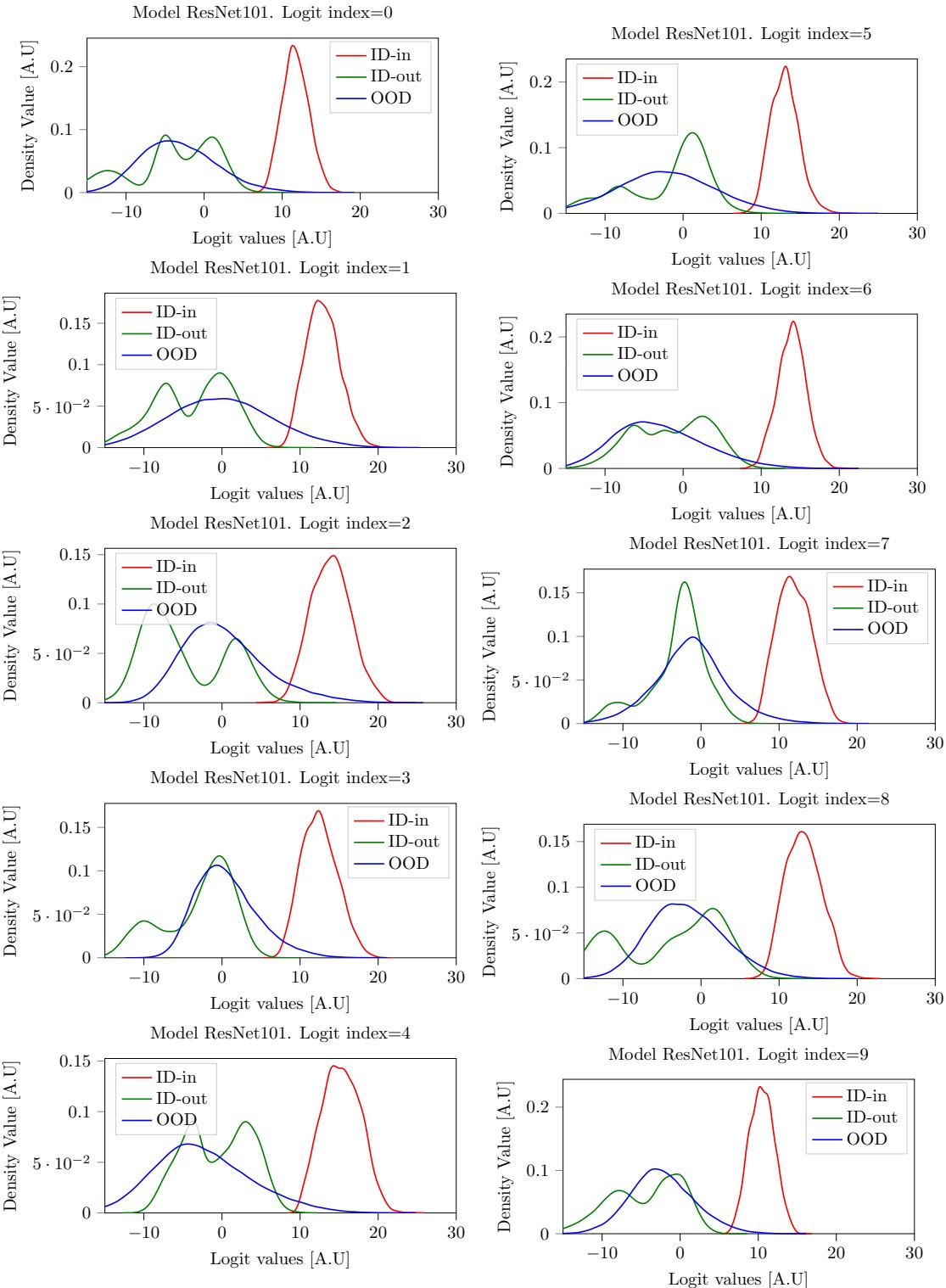

Figure 44: Logit cell densities for SVHN as ID with ResNet101.

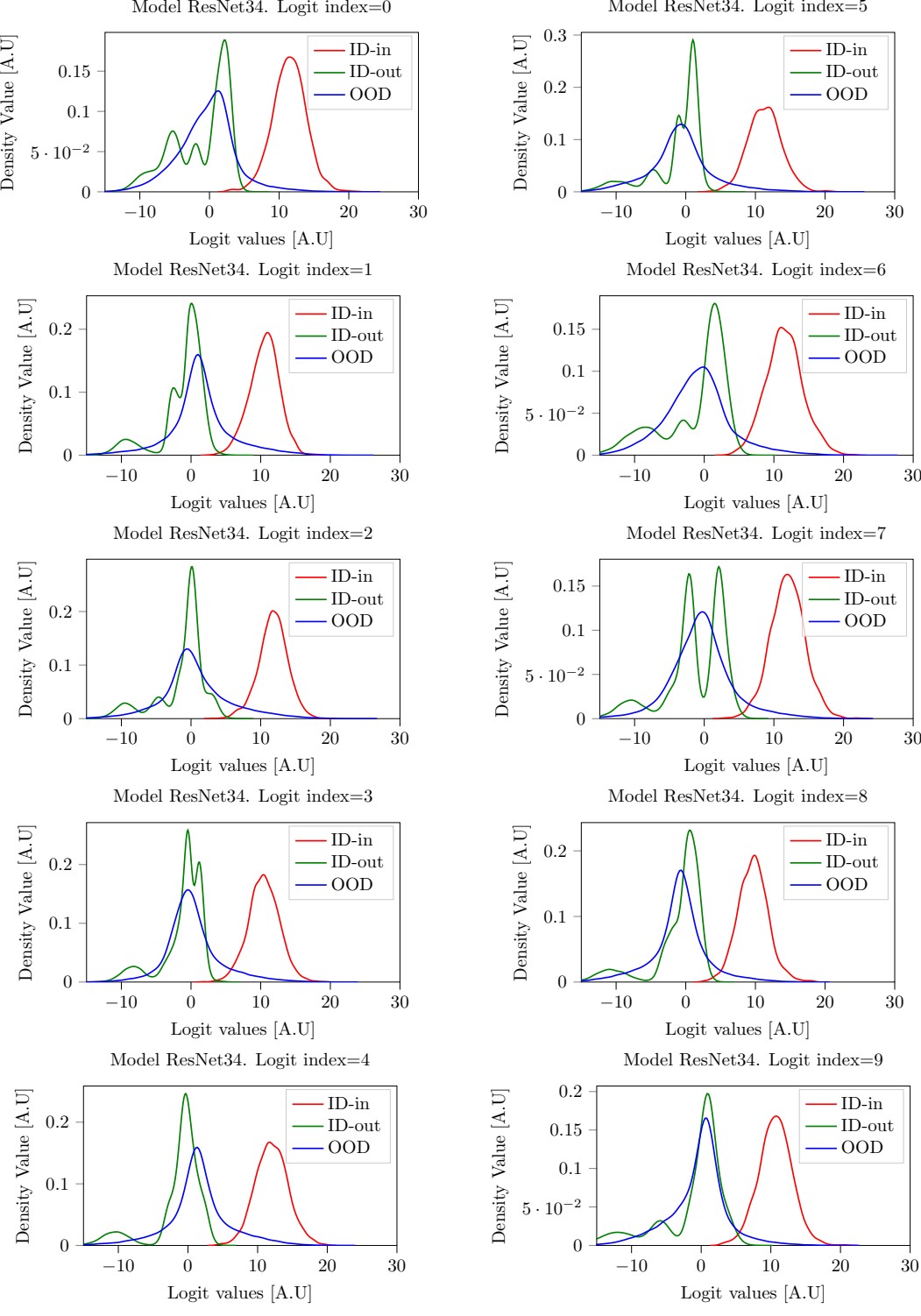

Figure 45: Logit cell densities for SVHN as ID with ResNet152.

# F   Experiments on different vision transformers

Last but not least figs. 47 to 54 showcase a detailed visualization of the ID and OOD logits for each cell across different variants of Vision Transformers for comprehensive comparative analysis.

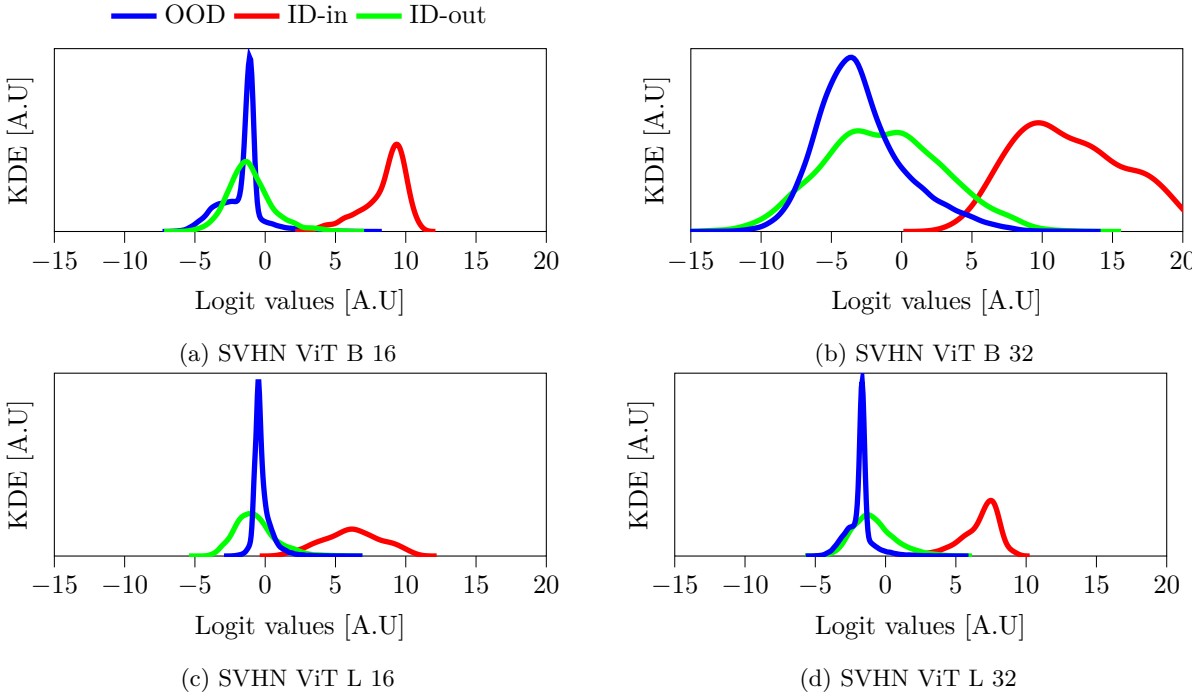

Figure 46: An analysis of the density over aggregated logits across distinct ViT architecture trained on the SVHN as the ID data, while the OOD includes $\{D\}$/SVHN.

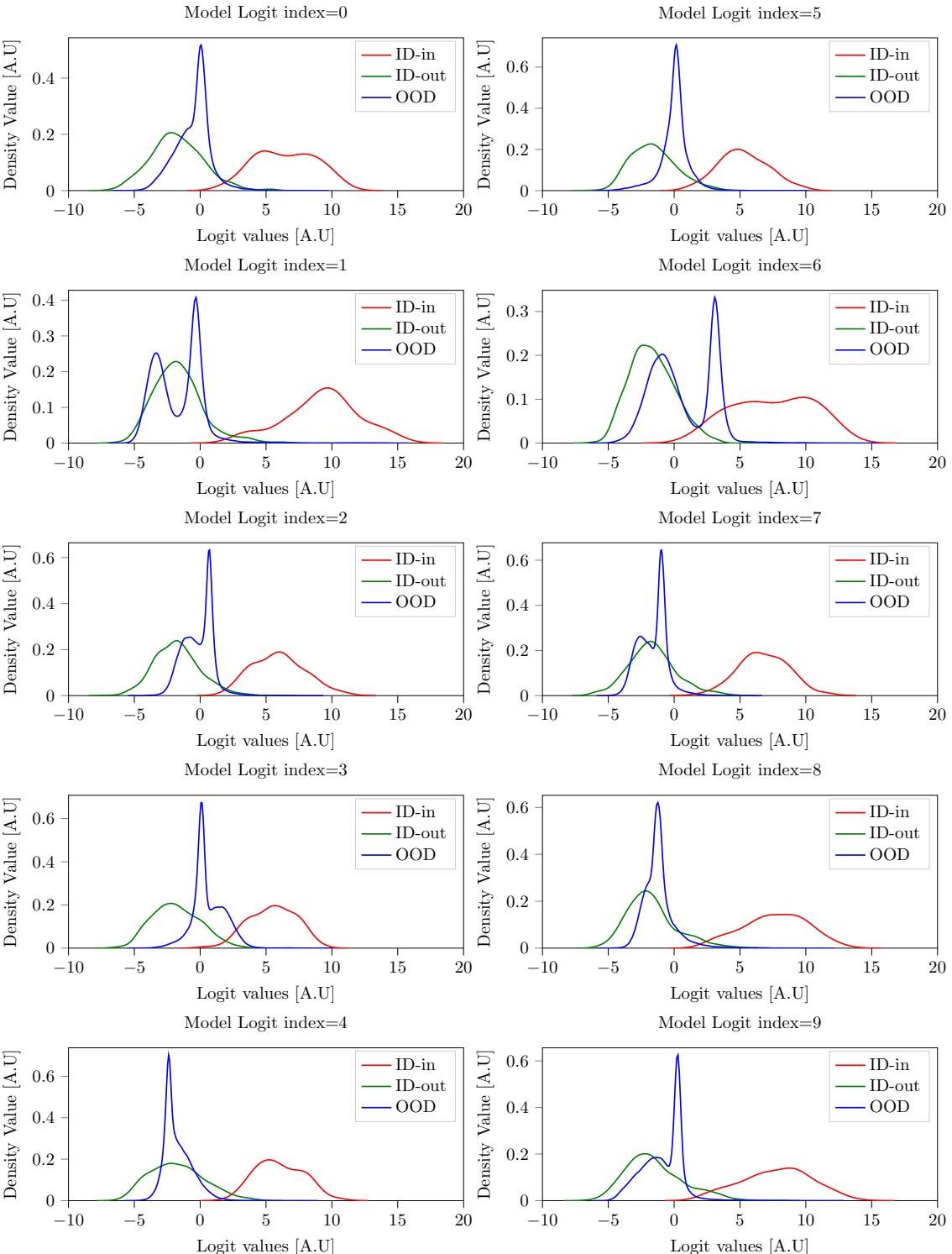

Figure 47: Logit cell densities for CIFAR-10 as ID with ViT-B-16.

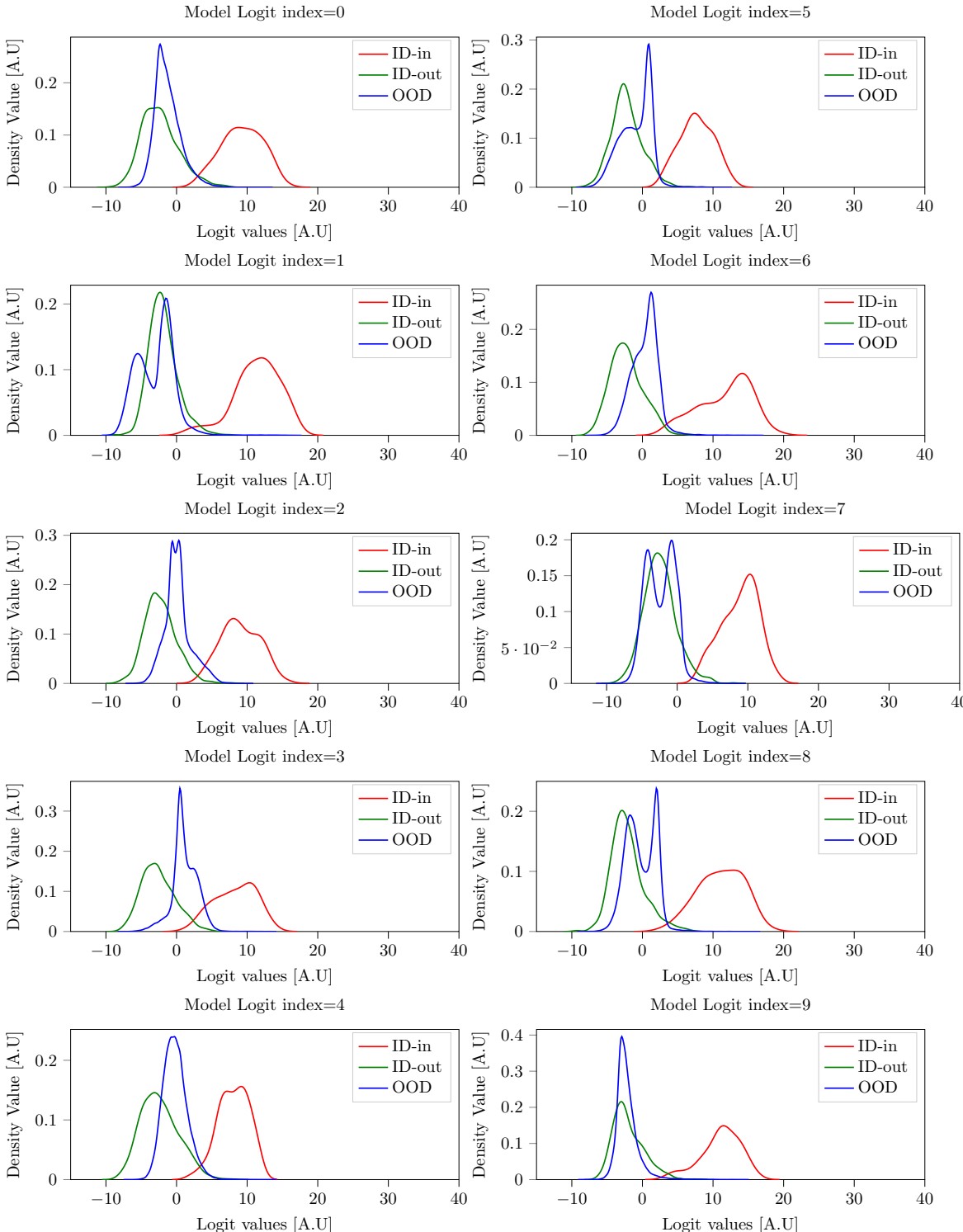

Figure 48: Logit cell densities for CIFAR-10 as ID with ViT-B-32.

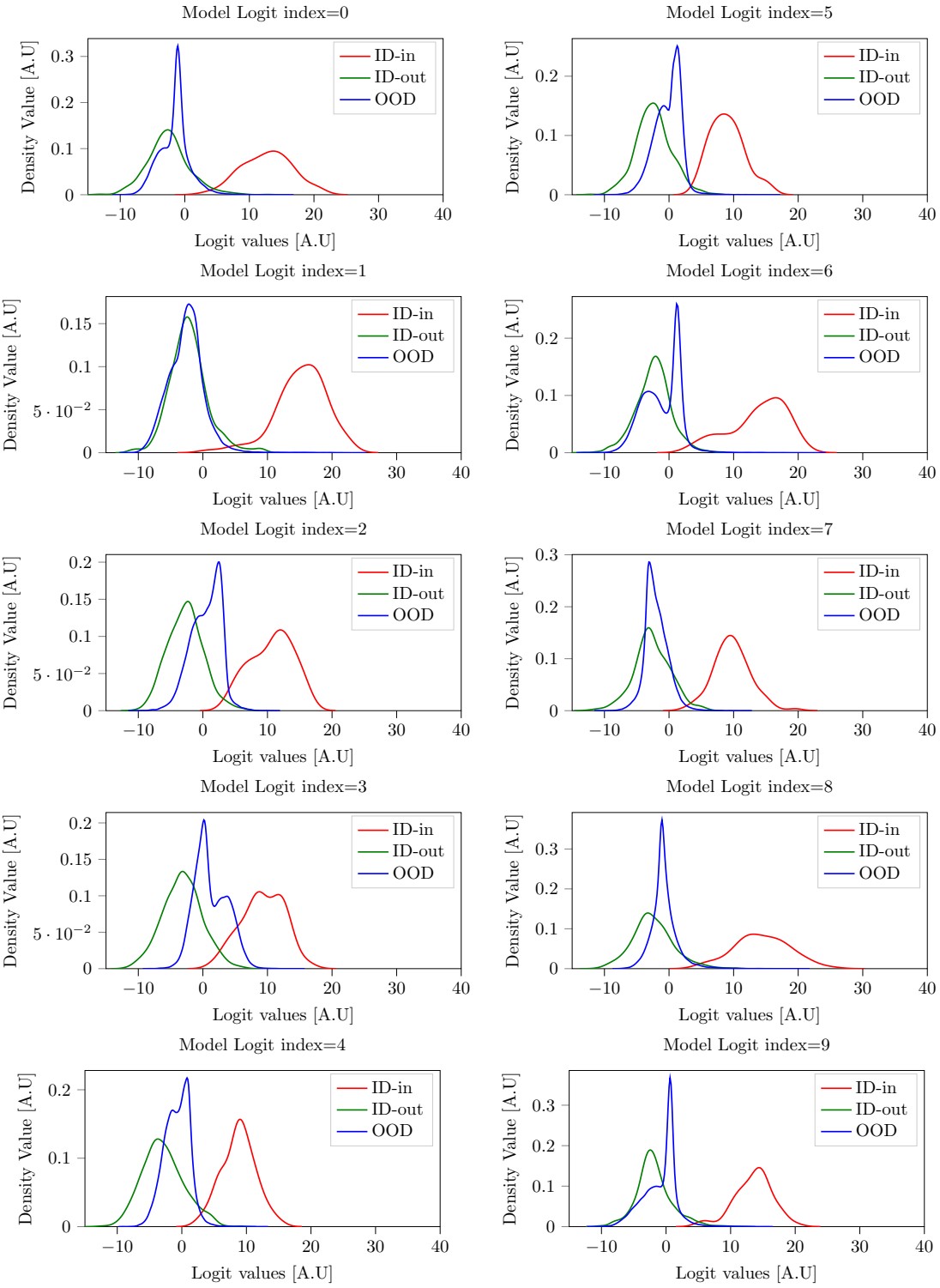

Figure 49: Logit cell densities for CIFAR-10 as ID with ViT-L-16.

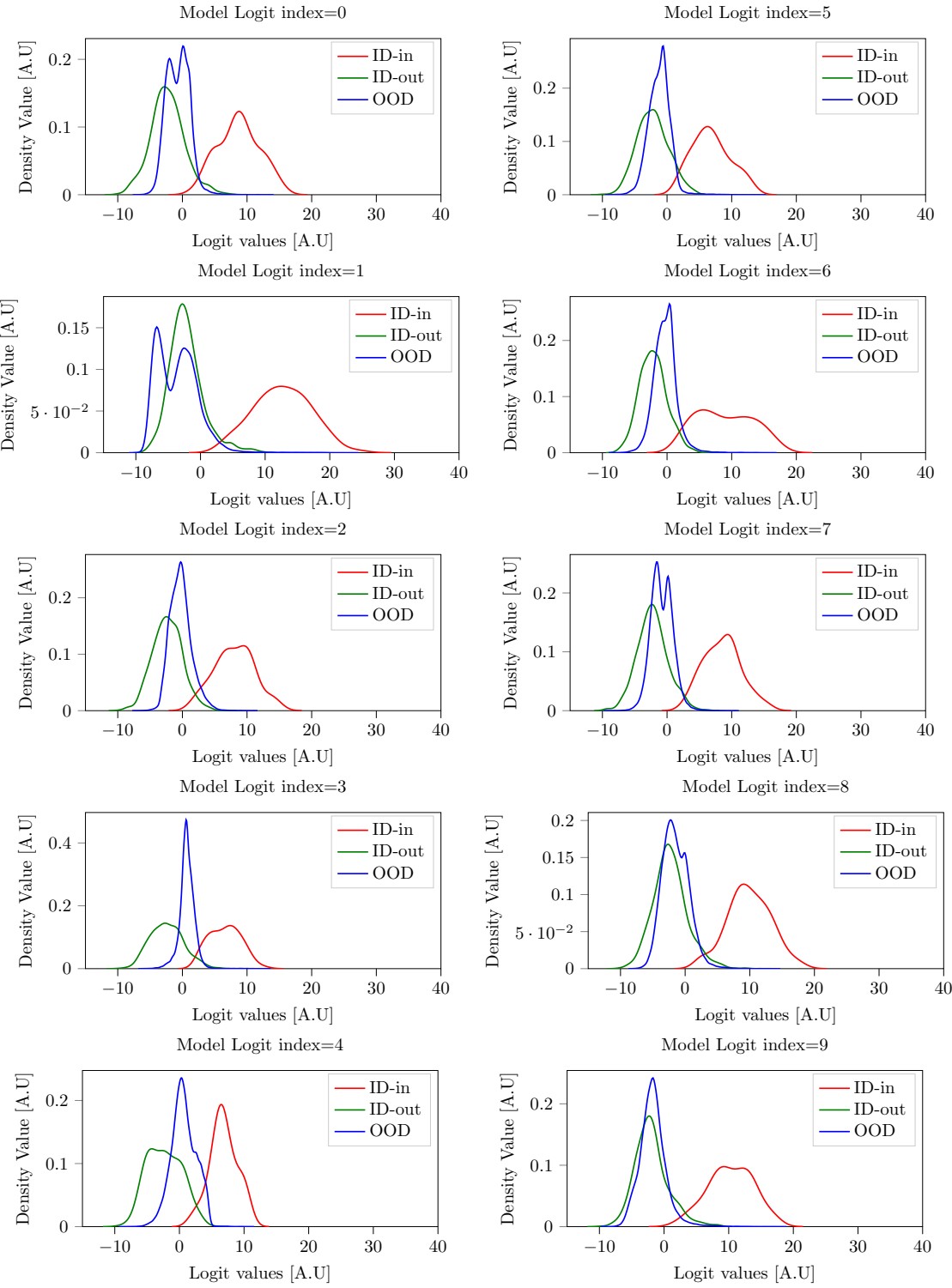

Figure 50: Logit cell densities for CIFAR-10 as ID with ViT-L-32.

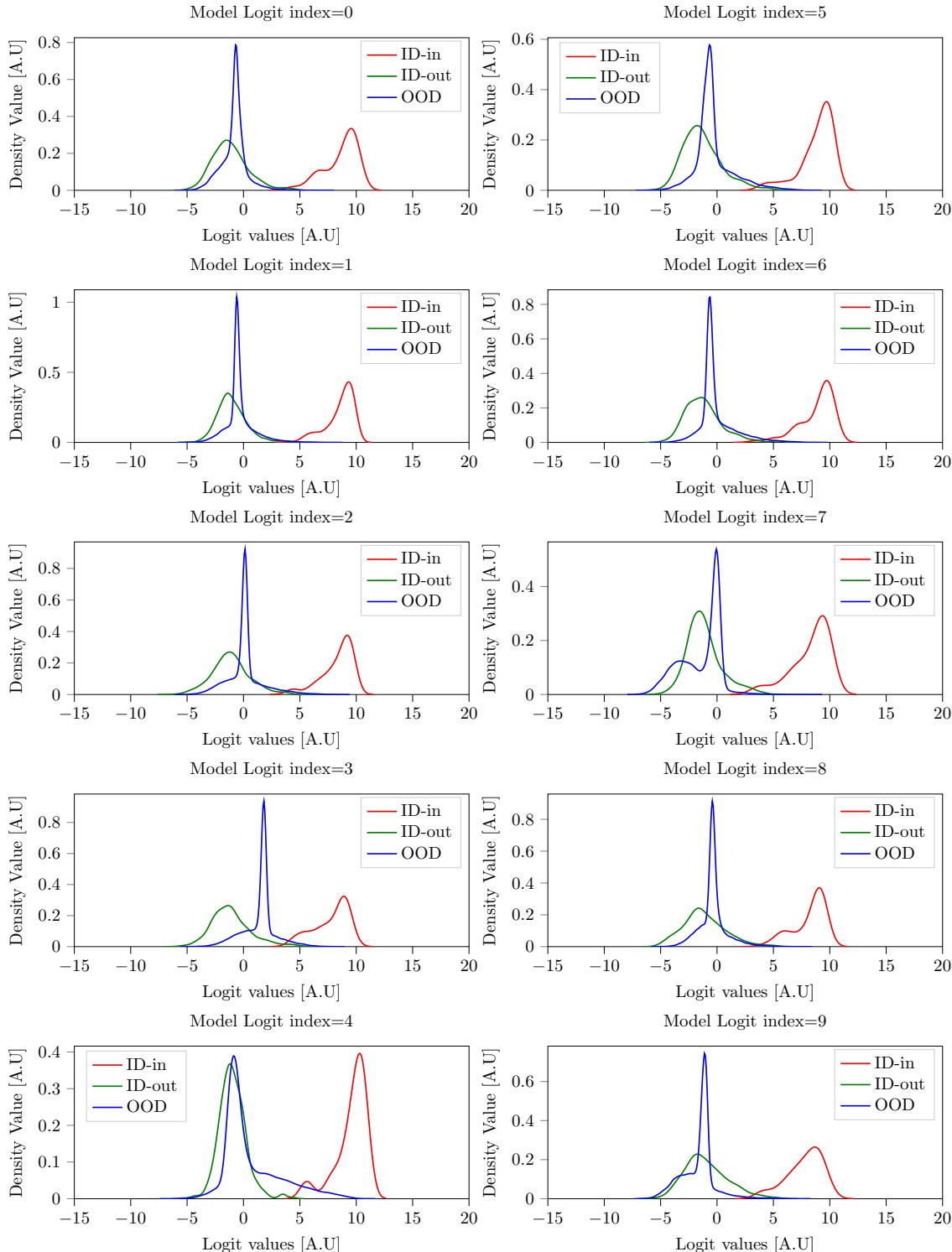

Figure 51: Logit cell densities for SVHN as ID with ViT-B-16.

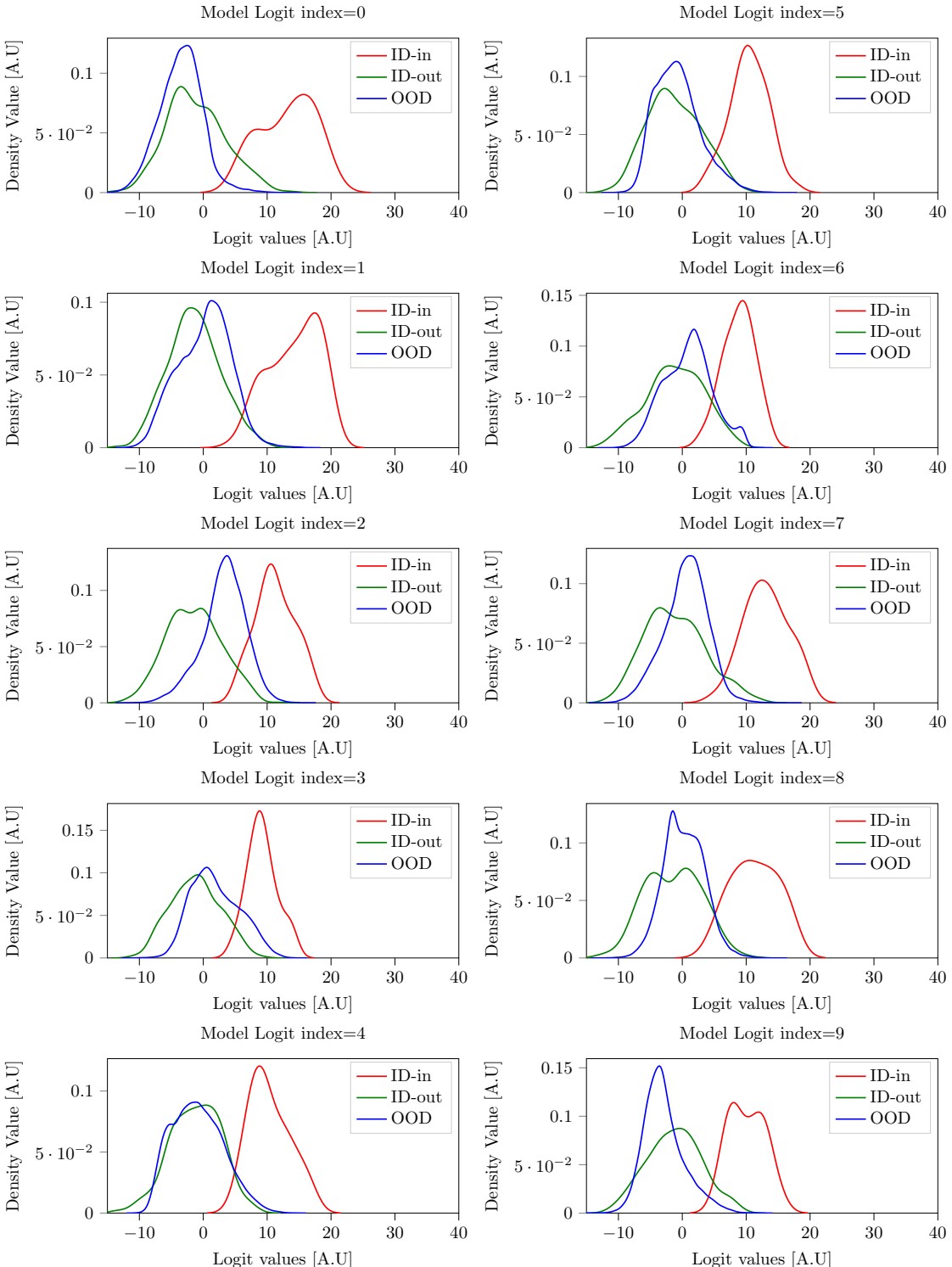

Figure 52: Logit cell densities for SVHN as ID with ViT-B-32.

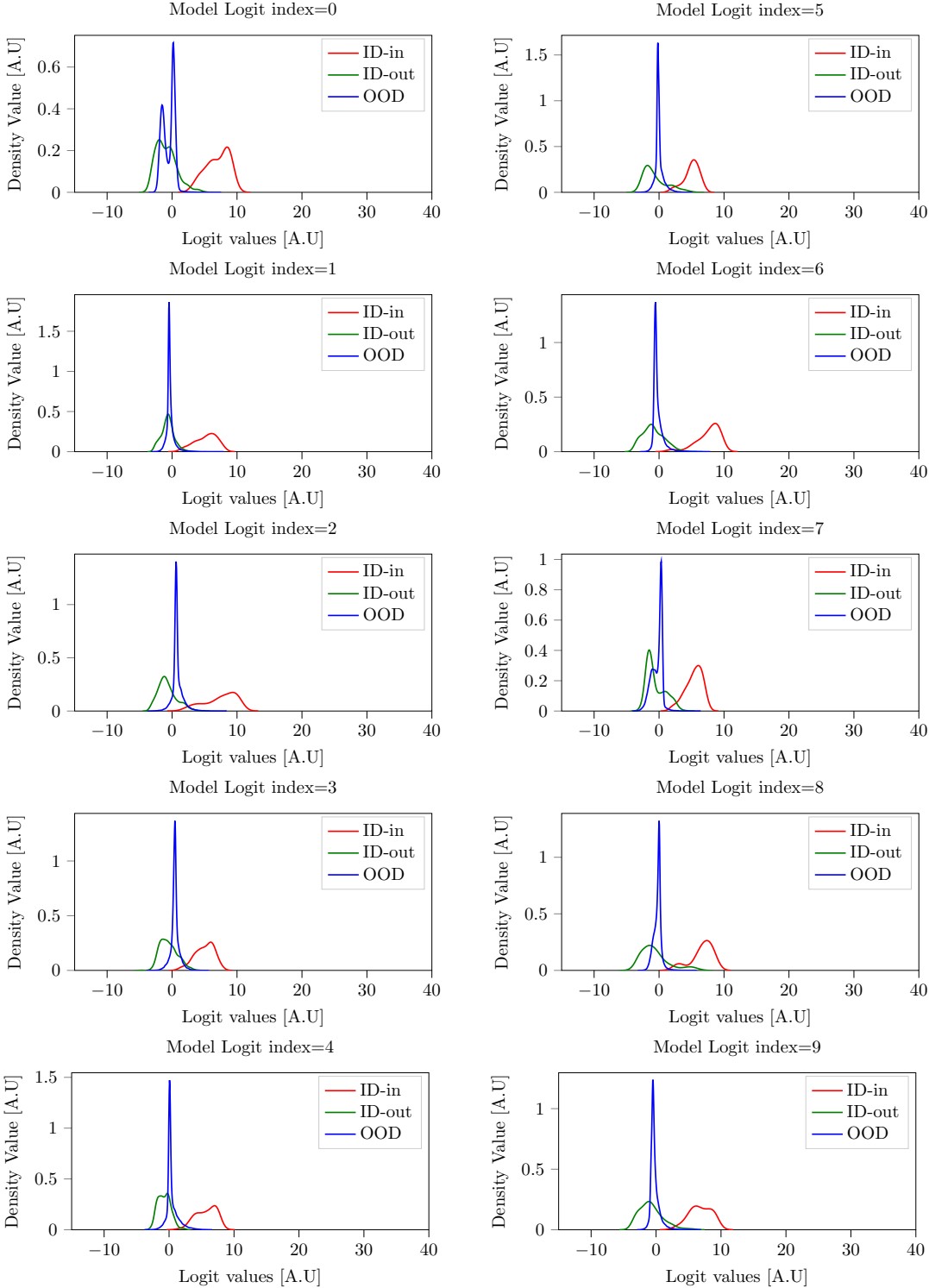

Figure 53: Logit cell densities for SVHN as ID with ViT-L-16.

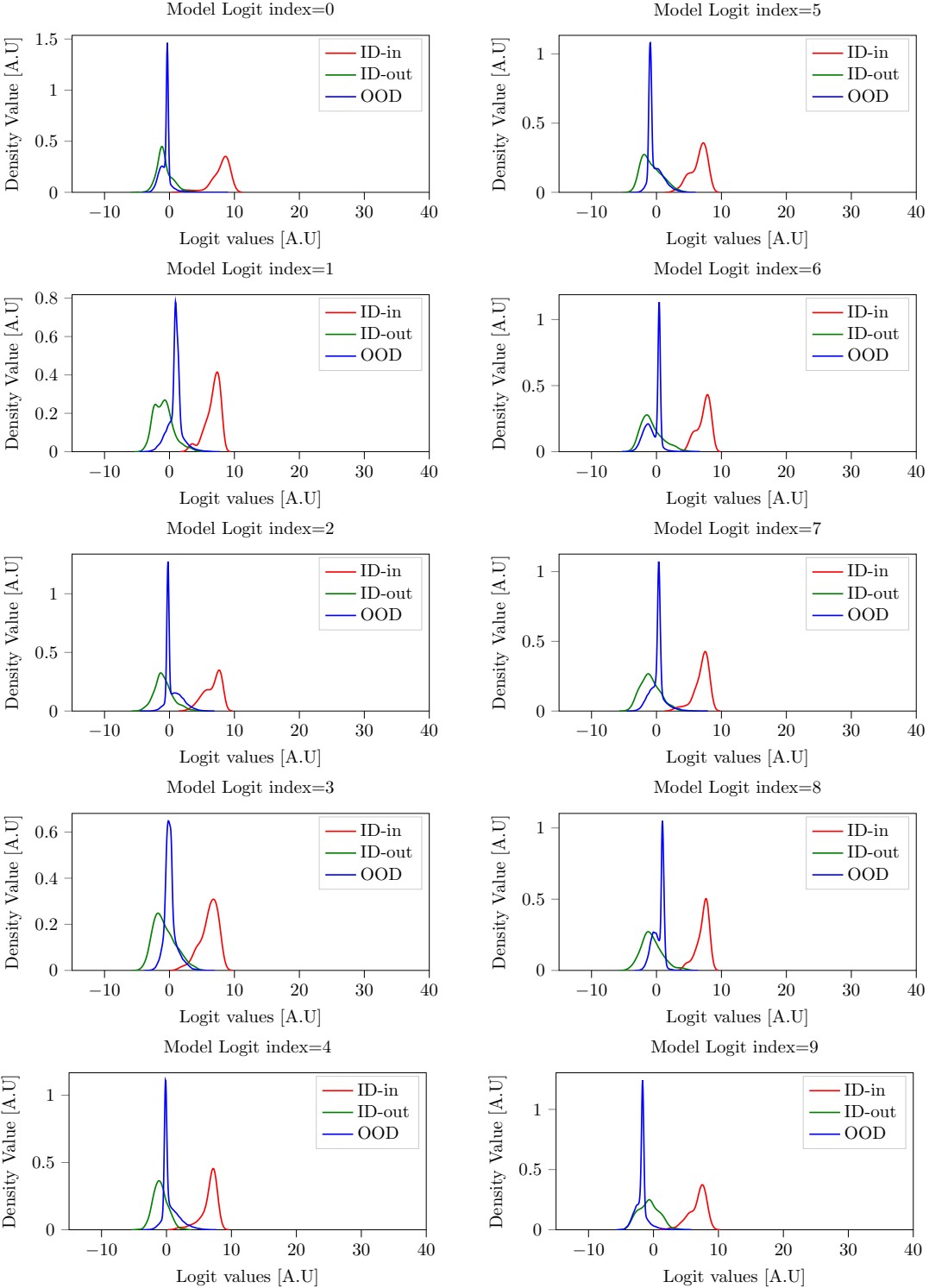

Figure 54: Logit cell densities for SVHN as ID with ViT-L-32.

## G   Additional experimentation on grayscale image

The classifier model, which is used for this purpose, consists of three convolutional layers followed by two fully connected layers (see table 2). This model is then trained using the Adam optimizer (Kingma & Ba, 2017) via a learning rate of $lr = 10^{-4}$ with weight decay $w_{decay} = 10^{-6}$ and with $\beta_1 = 0.8$ and $\beta_2 = 0.999$. A batch size of 256 is applied for both test and train data. No augmentation or regularization is applied to the training process. ReLU activation is utilized at every layer of the network. For a detailed visual analysis of each logit cell, refer to figs. 55 and 56.

Table 2: A convolutional neural network model for the experiment on the fashion-MNIST and MNIST datasets.

| Layer (Type) | Matrix | Nr Parameters |
|---|---|---|
| Conv2d-1 | [64,28,28] | 640 |
| BatchNorm2d-2 | [64,28,28] | 128 |
| ReLU | [64,28,28] | 0 |
| Conv2d-3 | [128,14,14] | 73,856 |
| BatchNorm2d-4 | [128,14,14] | 256 |
| ReLU | [128,14,14] | 0 |
| Conv2d-5 | [256,5,5] | 295,168 |
| BatchNorm2d-6 | [256,5,5] | 512 |
| ReLU | [256,5,5] | 0 |
| Linear-7 | [16] | 16,400 |
| ReLU | [16] | 0 |
| Linear-8 | [10] | 170 |

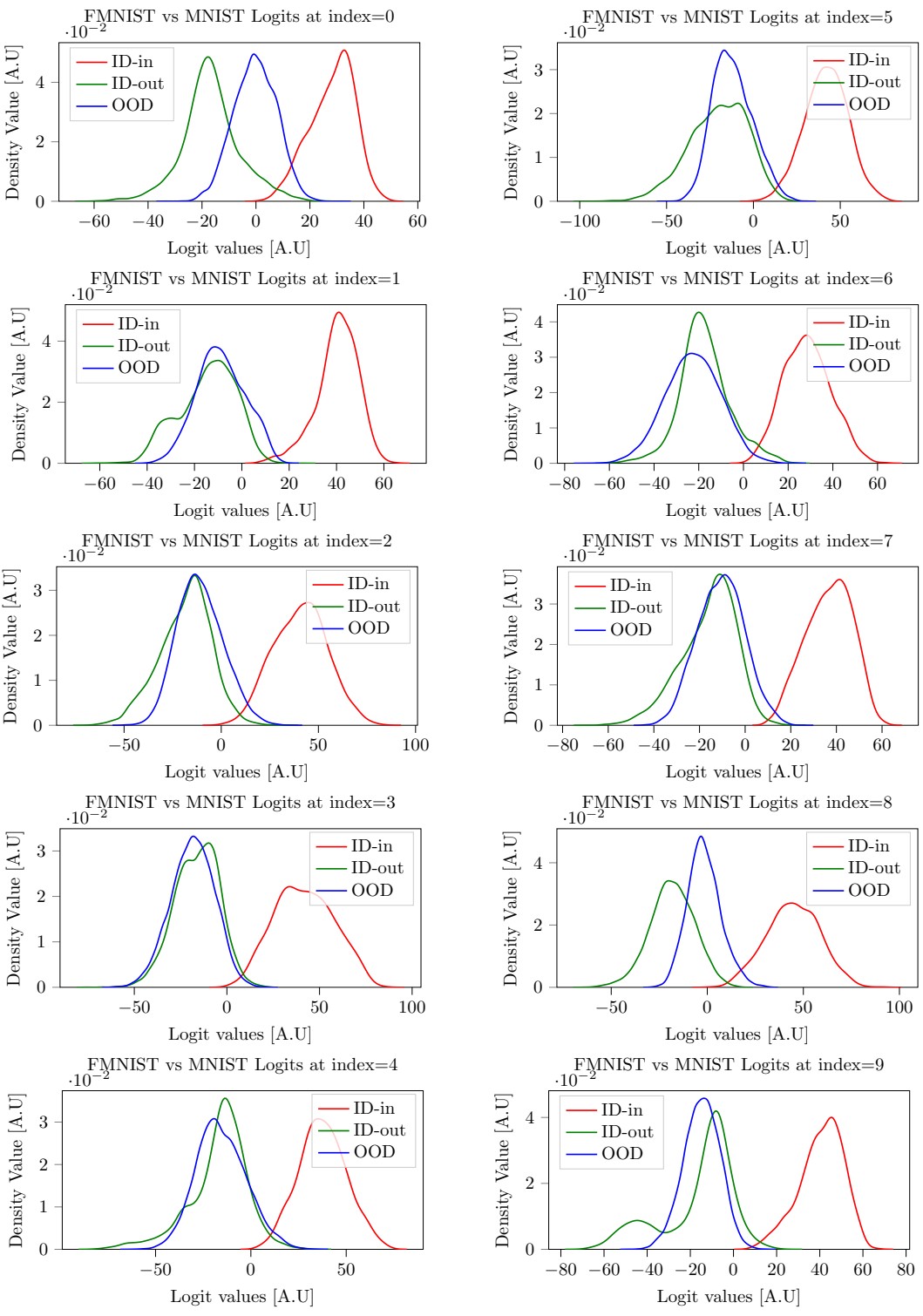

Figure 55: Logit cell densities for FMNIST as ID and MNIST as OOD with the model in table 2.

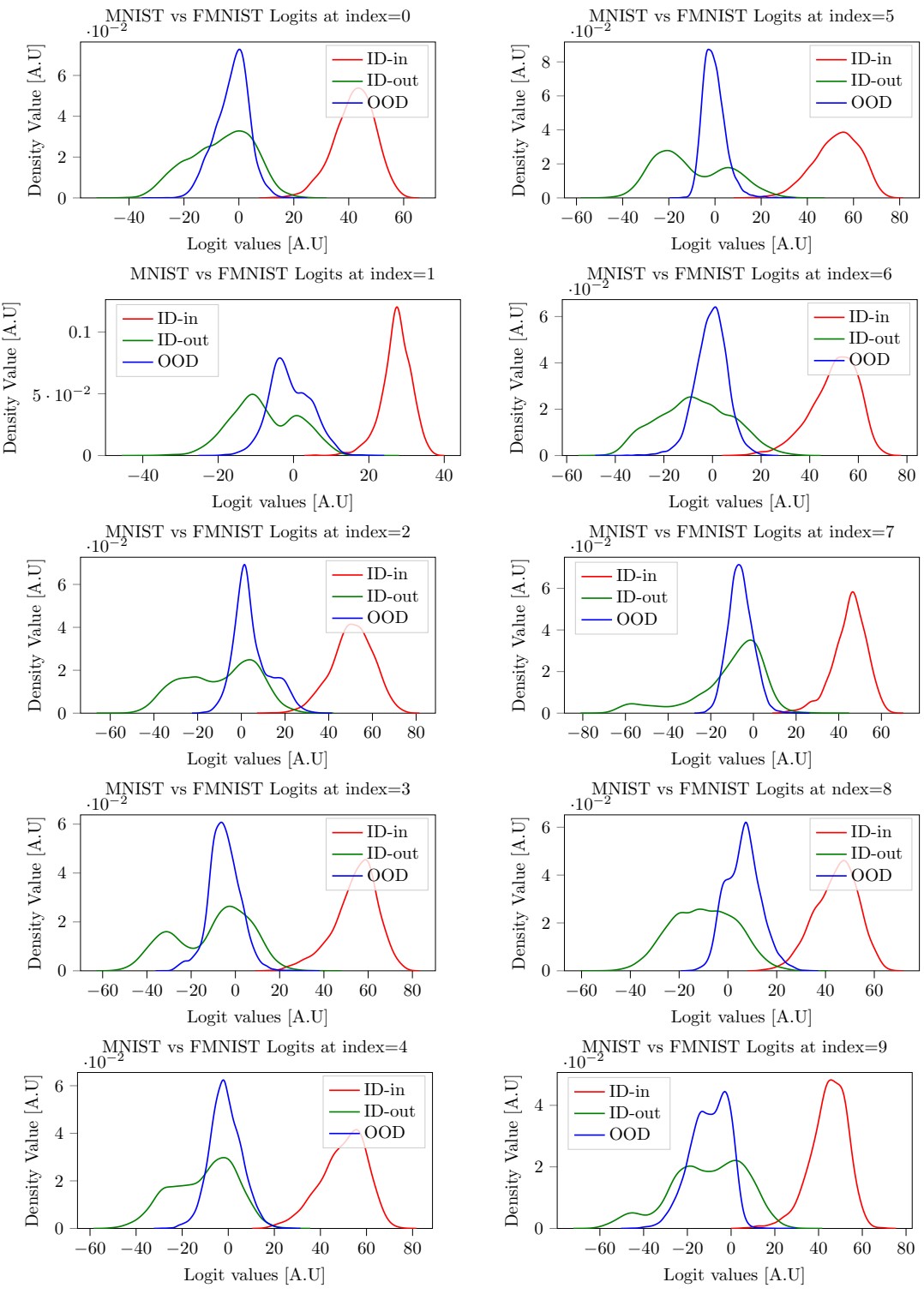

Figure 56: Logit cell densities for MNIST as ID and FMNIST as OOD with the model in table 2.

