# OpenReview forum: "Exploring the Spatial Dynamics of In-Distribution and Out-of-Distribution Data in Logit Space"
_TMLR — Rejected by TMLR_

### Review · Reviewer_iBLy · 2025-03-06

**Summary Of Contributions:**

This work authors study the trends exhibited by OOD and ID samples in the logits space and demonstrate that OOD samples reside near the center of the logit space and ID samples tend to be situated far from the center predominantly in the positive region of the logit space.

**Audience:**

Yes

**Claims And Evidence:**

No

**Requested Changes:**

- I believe authors need to clearly define what is meant by OOD in their setup along with explaining why their definition is justified with respect to the real-world setting and update the claims and corollary accordingly.

- I request the authors to restructure Section 4 to explain the setup before diving deeper into the results and conclusions. I also believe the paper would improve by more detailed discussion on the results.

- I would ask the authors to test their hypothesis adversarially as mentioned in point 2 of my comments and questions. Does the hypothesis hold true in these cases as well?

- It would be interesting to see the implications of the findings of the work. Looking at the works that use logit space to distinguish between OOD and ID. Can this work shed light on why those are sucessful in OOD?

**Strengths And Weaknesses:**

# Strengths
- Authors explain the motivation and intuition behind the trends they analyized.

# Comments and Questions
- In corollary authors assume ID and OOD are statistically independent. **I am not sure if this assumption is fair for the setting of OOD detection**. I agree with the authors if we assume ID and OOD are statistically independent it might be easier to distinguish between the two and we can comment and explain the trends. However for the setting of OOD detection I believe the evaluation of the detection algorithm should be done adversarially, that is, test with the images which are close to the ID but are guaranteed to be OOD. For example, permuation of an Image (figure 12 in [1]) or even Mixup [2] create an image which does not come from ID. I believe authors need to justify the basis for this assumption.

- In the case where we train with MixUp (or any Vicinal Risk Minimization algorithm ), we are technically training the model on OOD data as inputs. This is because when we mix 2 images we are by definition creating synthetic images which have to lie outside ID. In this scenario I believe the argument that OOD data lies around the center or near the origin might not hold true and it might be difficult to make comments about the trends. Can authors comment on this or add some discussion to the paper to clarify the same.

- I believe there are works investigating the logit space for OOD detection. The work in [3]  states "There are OOD samples that are easy to identify in the feature space while hard to distinguish in the logit space and vice versa." Additionally, I believe [3] paper also outlines other works that use the logits space for OOD detection. **I strongly believe the authors should update the claim "none of the existing works have explicitly investigated the segregation of ID and OOD data within the logit space." and add the relevant citations for the same.**

- Section 4, while presenting valuable experimental results, currently reads more like a detailed report than a focused analysis within a research paper. I believe it would benefit from significant restructuring. Specifically, the section begins abruptly with 'In these experiments...' without first establishing the experimental setup, the datasets used, or the context for the subsequent plots and conclusions. Providing a clearer introduction to the experimental design would greatly enhance the reader's understanding.
Furthermore, I suggest a review of the balance between visual presentation and in-depth discussion. While the plots are informative, dedicating nearly a full page to each in several instances may detract from the overall flow and focus of the paper. I recommend considering moving some of the less critical plots to the appendix. Conversely, some of the insightful findings currently in the appendix could potentially be elevated to the main body, or the discussion surrounding the most important experimental results could be expanded to thoroughly explore their implications. This would allow for a more streamlined presentation and a deeper exploration of the research's contributions.


# Reference
[1] Mukherjee, Amitangshu, Timur Ibrayev, and Kaushik Roy. "On Inherent Adversarial Robustness of Active Vision Systems." Transactions on Machine Learning Research (2024).

[2] Verma, Vikas, et al. "Manifold mixup: Better representations by interpolating hidden states." International conference on machine learning. PMLR, 2019.

[3] Wang, Haoqi, et al. "Vim: Out-of-distribution with virtual-logit matching." Proceedings of the IEEE/CVF conference on computer vision and pattern recognition. 2022.

---

> ### Author Response · Authors · 2025-03-20
> **Comment the definition of OOD data**
>
> Dear Reviewer,
> thank you for providing detailed feedback on our work.
> We sincerely apologize for the delay in our response.
>
> We have altered the assumption of statistical independence regarding. Instead, we now emphasize the non-overlapping distribution of the discriminative features. Initially, we attempted to use the term **certain level** to imply that the features are not entirely statistically independent. However, upon further consideration, we concluded that describing it as a non-overlapping distribution of discriminative features is more precise and clearer.
>
> > **Comment** : I believe authors need to clearly define what is meant by OOD in their setup along with explaining why their definition is justified with respect to the real-world setting and update the claims and corollary accordingly.
>
> **In the method section we wrote the following**
>
> In exploring ID and OOD data, it is crucial to delineate their distinctions in relation to DL classifiers.
>
> The training dataset is regarded as the optimal empirical representation of ID data.
> Within this dataset, ID data tend to aggregate into class-specific clusters based on discriminative features corresponding to each class.
> The exact parameterization of this feature space remains unknown; however, the annotated empirical ID dataset serves as a practical surrogate.
> Assuming these discriminative features constitute a multimodal distribution where each mode corresponds to one of the target classes.
> For ID data classification, DL models are engineered to exploit these discriminative features distribution and the annotated labels to effectively map the ID data into respective class-specific clusters within the logit vector space.
>
> Whenever we encounter data whose features deviate significantly from this defined distribution of discriminative ID features, they are considered OOD.
> The degree to which the features of OOD data diverge from this distribution determines their shift with the ID features.
> Consequently, the more distant the OOD features are from the ID distribution, the lesser their resemblance to the ID discriminative features, rendering them less perceptible from the DL model.
>
> More concretely, a DL classifier trained to differentiate cats from dogs will encounter difficulties with photos of horses and wolves. However, because wolves' features are more similar to dogs, wolves represent closer OOD data than horses, which are farther from both trained categories.

---

> ### Author Response · Authors · 2025-03-20
> **Comment regarding the implications of this work.**
>
> Dear Reviewer,
>
> Many thanks for your comment regarding the implications of this work.
> We sincerely apologize for the delay in our response.
>
> > **Comment**: It would be interesting to see the implications of the findings of the work. Looking at the works that use logit space to distinguish between OOD and ID. Can this work shed light on why those are successful in OOD?
>
> a) The primary and most significant contribution of this work is the demonstration that by anticipating the most probable distribution of OOD and ID logits, we can develop a binary classifier capable of distinguishing between them without relying on thresholding methods that label any non-ID data as OOD.
> Specifically, by employing an exact likelihood estimator, we can define distinct regions corresponding to high-density areas of OOD and ID logits. This allows us to differentiate between the two by comparing their maximum likelihood values, eliminating the need for arbitrary thresholds altogether.
>
> b) Secondly, leveraging the near-orthogonal structure of ID clusters in the logit space enables us to employ a single unimodal density estimator amortized along each axis for precise likelihood estimation.
> This approach eliminates the need for per-class density estimators, which are commonly used by competing methods. Such per-class estimators not only demand more complex models and larger amounts of training data but are also more prone to errors.
> Even a minor imperfection in estimating the likelihood for a single class can lead to inaccurate overall likelihood estimation, ultimately compromising the performance of out-of-distribution (OOD) detection.
>
> Notice that the other approaches assume a natural displacement of OODs from IDs, which we showcase empirically.
> However, we go even further and explain the actual nature of the separation.

---

> ### Author Response · Authors · 2025-03-23
> **Alternation of the related methods section**
>
> We sincerely appreciate the reviewer's insightful comment. Upon careful consideration, we agree that the original statement did not fully capture the intended message. While it is true that several existing methods distinguish OOD from ID using logits, our work introduces a novel contribution by focusing on their anticipated configurations, which sets our approach apart.
>
> >**Comment**: I believe there are works investigating the logit space for OOD detection. The work in [3] states "There are OOD samples that are easy to identify in the feature space while hard to distinguish in the logit space and vice versa." Additionally, I believe [3] paper also outlines other works that use the logits space for OOD detection. I strongly believe the authors should update the claim "none of the existing works have explicitly investigated the segregation of ID and OOD data within the logit space." and add the relevant citations for the same.
>
> We instead altered the statement as follows:
>
> **Although the differentiation between OOD and ID logits has been extensively studied (references), existing methods do not anticipate their actual configuration.**
>
> Thank you for highlighting this point.

---

> ### Author Response · Authors · 2025-03-24
> **Comment regarding Vicinal Risk Minimization algorithms**
>
> Dear Reviewer,
>
> Thank you for your very insightful comment.
>
> >**Comment**: In the case where we train with MixUp (or any Vicinal Risk Minimization algorithm ), we are technically training the model on OOD data as inputs. This is because when we mix 2 images we are by definition creating synthetic images which have to lie outside ID. In this scenario I believe the argument that OOD data lies around the center or near the origin might not hold true and it might be difficult to make comments about the trends. Can authors comment on this or add some discussion to the paper to clarify the same.
>
>
> Indeed, mixing ID input data from different classes somewhat alters the data's discriminative features, resulting in these being perceived as near OOD. We conducted tests to verify this and confirmed that there is a shift towards the center of the ID-in relative to scenarios where no mixing occurs, where the ID-in logits are distinctly shifted towards positive values.
>
> The experiment logits in the Supplementary Material are derived using the Resnet on CIFAR-10 as the ID dataset.
>
> We have added an example of this observation in the Additional Material section for further clarification.

---

> ### Author Response · Authors · 2025-03-24
> **Comment regarding requested changes in Section 4**
>
> We sincerely appreciate the reviewer’s valuable feedback.
>
> >**Comment**:I request the authors to restructure Section 4 to explain the setup before diving deeper into the results and conclusions. I also believe the paper would improve by more detailed discussion on the results.
>
>
> In response to the comment, we have restructured Section 4 by relocating redundant plots to the Appendix to improve clarity and conciseness.
>
> Additionally, we have:
>
> Provided a more detailed explanation of the experimental setup,
>
> _To empirically validate the persistence of this configuration, we conduct a series of comprehensive experiments across diverse settings. Furthermore, to rigorously analyze the distributional properties of ID and OOD logits, we employ KDE to visualize their densities. This unified visualization framework enables a direct comparison of ID-in (the maximum logit values for correctly classified ID samples), ID-out (the remaining logit values for correctly classified ID samples), and OOD. By overlaying their KDE curves, we assess the divergence between ID-in and OOD, the separation between D-in and ID-out, and the degree of overlap between OOD and ID-out distributions._
>
>
> Expanded the discussion of results for each experiment to enhance interpretability.

---

### Review · Reviewer_Tuv2 · 2025-03-08

**Summary Of Contributions:**

The paper studies the spacial distribution of logits and how it changes during training, for In-distribution (ID) and out-of-distribution (OOD) data. Specifically, the paper shows that the logits of In-distribution data moves to large positive values, whereas logits of OOD data remains centered around 0. The paper studies this phenomenon for different ID and OOD datasets, different model architectures and other ablations over the architecture choice.

**Audience:**

Yes

**Broader Impact Concerns:**

I have no such concerns.

**Claims And Evidence:**

No

**Requested Changes:**

Please address the weaknesses and questions I mention above.

**Strengths And Weaknesses:**

# Strengths

1. The paper is well written and easy to follow.

2. I particularly liked the demonstration that ID-out (ID examples where model has bad probability predictions) and OOD examples have similar logit distribution. To be honest, OOD detection is a malformed problem, the more important/realistic problem is that of selective classification with OOD detection [5]. Ideally, one does not care about OOD detection particularly. What we care about is not making wrong predictions, and that means filtering both OOD data, and ID data where the model’s prediction is wrong. This paper’s study hints that they behave somewhat similarly, so there can possibly be a unified method to deal with both types of problematic data at once.

3. The paper does thorough ablations on multiple axes and reports them nicely.

# Weaknesses

My main concern for this paper is it implicitly (I did not find this assumption stressed anywhere in the paper) considers only easy OOD cases, i.e., cases when OOD data is clearly different from ID data. This means that it ignores the truly interesting OOD cases, called hard-OOD or near-OOD in recent literature, and cannot be used to understand those sets of problems. My specific concerns are listed below.

(**This particular claim is not correct**)

> Nevertheless, these approaches lack foresight into the configuration of OOD and ID data within the latent space, instead making an implicit assumption regarding their inherent separation.

Not all methods assume this. There is a test-time training line of literature that tries to enforce the separation of ID and OOD examples using the test set/some other dataset with OOD examples in the mix, see [1], [2].

(**DLs maintaining low dot-product for OOD data is unclear**)

> To clarify the configurations of ID and OOD logits, we highlight the reliance of DL models on convolutions and matrix multiplications, which are composed of dot-product computations between the model weights and the input data. During training, the objective is to maximize the dot-product values between the model weights and the ID training data. Concurrently, the DL model maintains low dot-product values for the OOD data.

This requires more assumptions. I suspect it heavily depends on the actual ID and OOD distributions in question (eg., if ID distribution is classifying huskies vs cats, and OOD distribution includes wolves, I would be curious to see if this assumption holds). The authors need to make their assumptions more concrete before making this claim.

(**This claim is also too strong**)

> Considering that OOD and ID data derive from fundamentally different distributions, they inherently exhibit a certain level of statistical independence.

In effect, the authors are assuming that the covariance between ID and OOD data is 0. This is too strong of an assumption (also not mentioned explicitly in the paper), and it also makes the rest of the paper weak.

# Questions

I have the following questions I hope the authors would try to address:

1. (**Figure 2**): are these logits calculated on held-out/validation examples or training examples?

2. Could the authors clarify the role of ID-in and ID-out more carefully? Are the model’s predictions on ID-out examples wrong? Or are they mostly correct? What is the fraction of ID examples that are ID-in vs ID-out?

3. I have a hard time understanding/believing Fig 2. It seems that one should be able to choose a threshold on logit values, and very cleanly separate CIFAR-100 (ID) and CIFAR-10 (OOD) images. This is unclear, since prior work has always classified this as a very hard problem [3, 4]. My personal experience says this is one of the hardest OOD detection dataset pairs among the toy datasets (CIFAR-10, CIFAR-100, SVHN) etc. Prior work, such as Table 8 of [1], shows that CIFAR-100 vs CIFAR-10 is a hard OOD detection problem and classifiers based on thresholding logits should perform poorly. Could the authors explain this discrepancy?

# References

[1] Conservative Prediction via Data-Driven Confidence Minimization, https://arxiv.org/abs/2306.04974

[2] Training OOD Detectors in their Natural Habitats, https://arxiv.org/abs/2202.03299

[3] Exploring the Limits of Out-of-Distribution Detection, https://arxiv.org/abs/2106.03004

[4] A Simple Fix to Mahalanobis Distance for Improving Near-OOD Detection, https://arxiv.org/abs/2106.09022

[5] Plugin estimators for selective classification with out-of-distribution detection, https://arxiv.org/abs/2301.12386

---

> ### Author Response · Authors · 2025-03-21
> **Comments regarding additional and concrete assumptions regarding OOD data**
>
> Dear Reviewer,
> Thank you for your comment on the assumptions. Please accept our apologizies for the very late reply
>
> >**Comment**:**(DLs maintaining low dot-product for OOD data is unclear)**
>  >>To clarify the configurations of ID and OOD logits, we highlight the reliance of DL models on convolutions and matrix multiplications, which are composed of dot-product computations between the model weights and the input data. During training, the objective is to maximize the dot-product values between the model weights and the ID training data. Concurrently, the DL model maintains low dot-product values for the OOD data.
>
> >This requires more assumptions. I suspect it heavily depends on the actual ID and OOD distributions in question (eg., if ID distribution is classifying huskies vs cats, and OOD distribution includes wolves, I would be curious to see if this assumption holds). The authors need to make their assumptions more concrete before making this claim.
>
>
> **We added a separate section in the method section to lay down more concretely only the assumptions about OODs and IDs in relation to the DL model**
>
> _In exploring ID and OOD data, it is crucial to delineate their distinctions in relation to DL classifiers._
>
> _The training dataset is regarded as the optimal empirical representation of ID data.
> Within this dataset, ID data tend to aggregate into class-specific clusters based on discriminative features corresponding to each class.
> The exact parameterization of this feature space remains unknown; however, the annotated empirical ID dataset serves as a practical surrogate.
> Assuming, these discriminative features constitute a multimodal distribution where each mode corresponds to one of the target classes.
> For ID data classification, DL models are engineered to exploit these discriminative features distribution and the annotated labels to effectively map the ID data into respective class-specific clusters within the logit vector space._
>
> _Whenever we encounter data whose features deviate significantly from this defined distribution of discriminative ID features, they are considered OOD.
> The degree to which the features of OOD data diverge from this distribution determines their shift with the ID features.
> Consequently, the more distant the OOD features are from the ID distribution, the lesser their resemblance to the ID discriminative features, rendering them less perceptible from the DL model._
>
> _More concretely, a DL classifier trained to differentiate cats from dogs will encounter difficulties with photos of horses and wolves. However, because wolves' features are more similar to dogs, wolves represent closer OOD data compared to horses, which are farther from both trained categories._
>
> **Utilizing the above assumptions, we altered the manuscripts indicated as follows**
>
> _Assuming that DL models operate as spatial-invariance pattern-matching mechanisms operating over the distribution of discriminative features, these models produce high positive logits when the data features exhibit strong coherence with the feature representations parameterized during training.
> To do so_, DL models rely on convolutions and matrix multiplications, which are composed of dot-product computations between the model weights and the input data.
> During training, the objective is to maximize the dot-product values between the model weights and the ID training data.
>
> _Furthermore, we assume that the feature representations of OOD data reside in regions significantly distant from the distribution of discriminative features associated with ID data.
> Consequently, these OOD features exhibit a distributional shift relative to the feature representation parameterized by the DL model._
> As a result, the dot-products for the OOD data with the model weights remain low, leading to their logits being centrally concentrated both before and after training.
> Simultaneously, the logits associated with ID data are driven towards higher positive values.
>
>
> **We updated this in the main text and hope we addressed your concerns. We look forward to your feedback.**
>
> **Thank you for your time.**

---

> ### Author Response · Authors · 2025-03-21
> **Comment regarding the statistical independence.**
>
> Dear Reviewer,
>
> Please apologize for the late reply, and thank you for your very insightful comment regarding the **certain level** of statistical independence.
>
> Considering this is a concern from the **Reveiwer iBLy** as well, we altered the assumption of **statistical independence** to **distributional shift of the discriminative features** to be coherent with the assumptions we added in the Method section.
>
>
> >**Comment:(This claim is also too strong)**
> >>Considering that OOD and ID data derive from fundamentally different distributions, they inherently exhibit a certain level of statistical independence.
>
> >In effect, the authors are assuming that the covariance between ID and OOD data is 0. This is too strong of an assumption (also not mentioned explicitly in the paper), and it also makes the rest of the paper weak.
>
> Accordingly, we altered the main text of Corrollari 1 as well as the indicated text as follows:
>
> _Since the discriminative features of_ OOD and ID data derive from fundamentally different distributions, they inherently exhibit a certain level of _distributional shift_.

---

> ### Author Response · Authors · 2025-03-21
> **Answer to the questions**
>
> >**Q1**:(**Figure 2**): are these logits calculated on held-out/validation examples or training examples?
>
> The logits for both **Figure 1** and **Figure 2** are during training on only the **training set only**.
> While the **rest of the figures** are post-train on the **test set only**.
>
> >**Q2**: Could the authors clarify the role of ID-in and ID-out more carefully? Are the model’s predictions on ID-out examples wrong? Or are they mostly correct? What is the fraction of ID examples that are ID-in vs ID-out?
>
> **All logits discussed in this paper are derived from ID data, which yields accurate model predictions**.
> Moreover, within this correctly classified ID data, the ID-in logits correspond to the logit cell corresponding to the correct class, while the ID-out logits correspond to the remaining cells.
> For instance, consider a 10-class classifier with 10 logit cells: the ID-in logit is from the first output cell for test ID data belonging to class one, whereas the rest of the consecutive cells are ID-out logits.
> For this reason we have included in the appendix one KDE plot for each class as they are obtained from separate logit cells.
>
> _We will update their definition in the text with a more concrete example to clarify it._
>
>
> >**Q3**: I have a hard time understanding/believing Fig 2. It seems that one should be able to choose a threshold on logit values, and very cleanly separate CIFAR-100 (ID) and CIFAR-10 (OOD) images. This is unclear, since prior work has always classified this as a very hard problem [3, 4]. My personal experience says this is one of the hardest OOD detection dataset pairs among the toy datasets (CIFAR-10, CIFAR-100, SVHN) etc. Prior work, such as Table 8 of [1], shows that CIFAR-100 vs CIFAR-10 is a hard OOD detection problem and classifiers based on thresholding logits should perform poorly. Could the authors explain this discrepancy?
>
> Indeed, CIFAR-100 (ID) and CIFAR-10 (OOD) is the hardest OOD detection tasks to our knowledge when utilizing benchmark image datasets.
> The reason is twofold: a) CIFAR-100 is very hard to classify, and b) their discriminative features are very close to the CIFAR-10.
> Now, to be more illustrative, in Figure 2, we utilized only the correctly classified train examples from CIFAR-100, which is around 70%, and utilized the entire CIFAR-10 as OOD.
> The **overlap** would have been much **more profound** if we had utilized **all test data** from CIFAR-100 and not just the correctly classified, but it would have been less illustrative.
>
> We hope to answer your questions and are looking forward to more feedback.

---

> ### Author Response · Authors · 2025-03-21
> **Comment regarding the Abstract**
>
> We thank the reviewer for the comment and for the references.
>
> >**Comment**:(**This particular claim is not correct**)
> >>Nevertheless, these approaches lack foresight into the configuration of OOD and ID data within the latent space, instead making an implicit assumption regarding their inherent separation.
>
> >Not all methods assume this. There is a test-time training line of literature that tries to enforce the separation of ID and OOD examples using the test set/some other dataset with OOD examples in the mix,
>
> We have updated this part in the abstract as follows:
>
> Nevertheless, these approaches lack foresight into the configuration of OOD and ID data within the latent space.
> _Instead, they make an implicit assumption about their inherent separation or force a separation post-training by utilizing selected OOD data._

---

> > ### Comment · Reviewer_Tuv2 · 2025-04-02
> >
> > I thank the authors for their thoughtful rebuttal! These have mostly answered my concerns.

---

### Review · Reviewer_gz9G · 2025-03-11

**Summary Of Contributions:**

This paper conducts a thorough analysis of the logit embedding landscape, revealing that both ID and OOD data exhibit a distinct trend. The proposed study is motivated by the mathematical property of neural networks. The empirical results demonstrate that leveraging the logit embedding space can be a prospective way of detecting OOD data.

**Audience:**

Yes

**Broader Impact Concerns:**

I do not see broader impact concern with this paper.

**Claims And Evidence:**

Yes

**Requested Changes:**

1. The paper can be strengthened by giving a concrete way of using the logit embedding space for detecting OOD data. The effectiveness of the method can be demonstrated by comparing the results with traditional methods for OOD detection on typical benchmark datasets for this task.
2. It would be great if the authors could clearly state the insights derived from the logit value plot. Currently, there are many plots of this kind in the main paper and Appendix, but it is unclear how to interpret them as the main results.

**Strengths And Weaknesses:**

Strengths:
1. The proposed study is motivated by the mathematical property of neural networks. The empirical results show that ID and OOD data behave differently in the logit embedding space.


Weaknesses:
1. The contribution of the current paper is more like a finding around ID and OOD data when embedded by deep neural networks. It is unclear how this finding would benefit OOD detection or the understanding/improvement of neural networks. Even though the paper argues that the observed patterns indicate the potential for new approaches that leverage ID-out logit as proxies for OOD instances, it is unclear how this would performance on benchmark datasets for the OOD detection task or in practice.
2. While the paper criticizes that most OOD detection methods result in complicated and hard-to-validate scoring techniques, I think this is maybe because OOD detection usually requires deriving a score for each sample and setting a threshold to distinguish OOD samples. If the finding of this paper is used to design a method for OOD detection, it is very likely that we need to derive a score based on the propert of logit embedding of the sample.

---

> ### Author Response · Authors · 2025-03-21
> **Comments regarding impact of the observation and concrete way of utilizing it**
>
> Dear Reviewer,
>
> Thank you for your thoughtful comment regarding the implications of our work. Please apologize for our late reply
>
> >**Comment**: The contribution of the current paper is more like a finding around ID and OOD data when embedded by deep neural networks. It is unclear how this finding would benefit OOD detection or the understanding/improvement of neural networks.
>
> >**Comment**: The paper can be strengthened by giving a concrete way of using the logit embedding space for detecting OOD data.
>
> a) **First implication**:
>
> This work's core contribution is demonstrating that by modeling the expected distribution of OOD and ID logits, we can construct a binary classifier capable of distinguishing between the two without relying on thresholding techniques that categorically label non-ID data as OOD.
> Specifically, by using an exact likelihood estimator, we define distinct regions corresponding to high-density areas of OOD and ID logits.
> This allows us to differentiate between OOD and ID data by comparing their maximum likelihood values from the two estimators, bypassing the need for arbitrary thresholds.
> This approach not only simplifies the detection process, but we expect to enhance its robustness.
>
> b) **Second implication**:
> Additionally, the near-orthogonal structure of ID clusters in the logit space can be leveraged to enable precise likelihood estimation using a single unimodal density estimator amortized along each axis.
> This eliminates the necessity for per-class density estimators, which are commonly employed by other methods.
> Per-class estimators often require more complex models and larger datasets and are more susceptible to errors.
> For example, even a small inaccuracy in estimating the likelihood for a single class can lead to significant errors in overall likelihood estimation, ultimately degrading OOD detection performance.
>
> > **Comment**: The effectiveness of the method can be demonstrated by comparing the results with traditional methods for OOD detection on typical benchmark datasets for this task.
>
> We are currently working on a very novel OOD detection separate work built on utilizing a) and b), and we will submit it once we have completed it.
>
> We hope this clarifies the implications of our work and its relevance to existing logit-based OOD detection methods. Please let us know if you have further questions or require additional details.

---

> ### Author Response · Authors · 2025-03-21
> **Comments on the continuation of the implications**
>
> Dear Reviewer,
> Thank you for your comment on the implication of the work. Please accept our apologies for this late reply.
>
> >**Comment**:While the paper criticizes that most OOD detection methods result in complicated and hard-to-validate scoring techniques, I think this is maybe because OOD detection usually requires deriving a score for each sample and setting a threshold to distinguish OOD samples. If the finding of this paper is used to design a method for OOD detection, it is very likely that we need to derive a score based on the property of logit embedding of the sample.
>
> Considering that we anticipate the placement of the OOD logits, we can now utilize this fact to map their region using density estimators for the exact likelihood estimator.
> All previous works have already showcased the effectiveness of utilizing the density mapping of ID logits; however, since they lack an exact likelihood estimator from the OOD region, they used a threshold on the ID likelihoods.
> Thus, once we have the likelihood estimator for both OOD and ID regions, one can differentiate between the two by simply comparing their values without the need for the threshold
>
> **We will expand this explain further in the main text**
>
> >**Comment**: Even though the paper argues that the observed patterns indicate the potential for new approaches that leverage ID-out logit as proxies for OOD instances, it is unclear how this would performance on benchmark datasets for the OOD detection task or in practice.
>
>
> Since, in practice, we do not have hold of OOD logits, we propose to mitigate by either:
> a) using ID-out as their proxies
> b) producing OOD logits by training the DL model first using half of the classes and the other half to produce OOD logits. Then, train the DL model in all classes to get the ID logits.
>
> **Both approaches to obtain proxies for the OOD logits are being investigated as follow-up future work** .
>
>
> We hope to answer your questions and are looking forward to your feedback.

---

> ### Author Response · Authors · 2025-03-24
> **Comment on the insight from the logit plots.**
>
> We sincerely appreciate the reviewer’s valuable feedback.
> Upon further consideration, we have expanded our discussion to provide additional clarification regarding the plots featured in both the main text and the appendix.
> >**Comment**: It would be great if the authors could clearly state the insights derived from the logit value plot.
> Currently, there are many plots of this kind in the main paper and Appendix, but it is unclear how to interpret them as the main results.
>
> To enhance clarity, we have included the following explanatory text:
>
> _To empirically validate the persistence of this configuration, we conduct a series of comprehensive experiments across diverse settings. Furthermore, to rigorously analyze the distributional properties of ID and OOD logits, we employ KDE to visualize their densities. This unified visualization framework enables a direct comparison of ID-in (the maximum logit values for correctly classified ID samples), ID-out (the remaining logit values for correctly classified ID samples), and OOD. By overlaying their KDE curves, we assess the divergence between ID-in and OOD, the separation between D-in and ID-out, and the degree of overlap between OOD and ID-out distributions._

---

### Decision · Action_Editor_uNQk · 2025-04-27

**Recommendation:** Reject

**Comment:**

This paper compares the dynamics of the model logits of in- and -out-of-distribution data in a deep learning model during model training.
 The reviews were below borderline with two reviewers leaning reject and one reviewer leaning accept.  One reviewer found that the claims made in the paper weren't sufficiently backed by evidence.    While the the reviewers found the paper well written and insightful, they seemed to have significant trouble agreeing with the underlying assumptions and claims of the authors.  In particular, the reviewers seemed to struggle to accept the notion that ID and OOD data are statistically independent and separable in the logit space.  They found that this included only "easy" OOD cases, which are somewhat already covered in the existing literature.  In addition, the reviewers seemed to feel that the paper fell short of convincing them that this would lead to any practical methods for doing OOD detection.  It would seem that this work could be made much stronger if the authors provided more justification for the underlying assumptions - maybe some experiments to show some circumstances that these hold.  In addition, the work would be more convincing if the authors demonstrated a practical method for performing OOD detection based on the analysis.

Unfortunately, it seems that the work falls just below the bar for acceptance.  However, I would encourage the authors to take the reviews into account and revise the manuscript.  As this involves possibly some significant additional empirical work, this would require a resubmission.

**Audience:**

There is certainly significant interest in out-of-distribution detection and it remains and active and important subfield of ML.

**Claims And Evidence:**

Two reviewers found that the claims were justified by the evidence in the paper.  However, one reviewer disagreed and felt that the authors did not sufficiently justify their approach.  Specifically, that reviewer argues that the notion that one can separate in and out-of-distribution data from just the model logits is strongly disputed in the literature.  Thus they argue that the authors need to do more to demonstrate that this is a reasonable setup.

**Resubmission Of Major Revision:**

The authors may consider submitting a major revision at a later time.